# Epidemic spreading under mutually independent intra- and inter-host pathogen evolution

Xiyun Zhang [1] ✉, Zhongyuan Ruan[2], Muhua Zheng [3], Jie Zhou[4], Stefano Boccaletti [5,6,7,11] & Baruch Barzel [8,9,10,11]

The dynamics of epidemic spreading is often reduced to the single control parameter $R_0$ (reproduction-rate), whose value, above or below unity, determines the state of the contagion. If, however, the pathogen evolves as it spreads, $R_0$ may change over time, potentially leading to a mutation-driven spread, in which an initially sub-pandemic pathogen undergoes a breakthrough mutation. To predict the boundaries of this pandemic phase, we introduce here a modeling framework to couple the inter-host network spreading patterns with the intra-host evolutionary dynamics. We find that even in the extreme case when these two process are driven by mutually independent selection forces, mutations can still fundamentally alter the pandemic phase-diagram. The pandemic transitions, we show, are now shaped, not just by $R_0$, but also by the balance between the epidemic and the evolutionary timescales. If mutations are too slow, the pathogen prevalence decays prior to the appearance of a critical mutation. On the other hand, if mutations are too rapid, the pathogen evolution becomes volatile and, once again, it fails to spread. Between these two extremes, however, we identify a broad range of conditions in which an initially sub-pandemic pathogen can breakthrough to gain widespread prevalence.

Epidemic dynamics are driven by the interplay of two processes, occurring at fundamentally different scales. First, at the individual level, once infected, the contracted pathogen undergoes replication, and potential mutation, within a host's body. Then, once reaching a sufficient load, transmission between hosts helps the pathogen penetrate the social network. These two processes, each characterized by its own intrinsic timescales and relevant parameters, are often modeled independently: the in-host dynamics is captured by a random process of mutation and selection[1–4]; the inter-host propagation is tracked using compartmental dynamics, such as the susceptible-infected-recovered (SIR) model[5,6], or its more elaborate variants[7]. While the former is inherently stochastic, most analytical advances on the latter[5] are achieved via deterministic methods, i.e. ordinary differential equations (ODE). This combination often prohibits systematic analytical advances[8], as we lack a unified toolbox by which to treat both the inter-host ODEs and the stochastic intra-host dynamics.

Here, we show that we can reduce the intra-host evolutionary dynamics into an effective random walk, biased or unbiased, in the

[1]Department of Physics, Jinan University, Guangzhou, Guangdong 510632, China. [2]Institute of Cyberspace Security, Zhejiang University of Technology, Hangzhou, Zhejiang 310023, China. [3]School of Physics and Electronic Engineering, Jiangsu University, Zhenjiang, Jiangsu 212013, China. [4]School of Physics and Electronic Science, East China Normal University, Shanghai 200241, China. [5]CNR - Institute of Complex Systems, Via Madonna del Piano 10, I-50019 Sesto Fiorentino, Italy. [6]Moscow Institute of Physics and Technology (National Research University), 9 Institutskiy per., Dolgoprudny, Moscow Region 141701, Russian Federation. [7]Universidad Rey Juan Carlos, Calle Tulipán s/n, 28933 Móstoles, Madrid, Spain. [8]Department of Mathematics, Bar-Ilan University, Ramat-Gan 5290002, Israel. [9]Gonda Multidisciplinary Brain Research Center, Bar-Ilan University, Ramat-Gan 5290002, Israel. [10]Network Science Institute, Northeastern University, Boston, MA 02115, USA. [11]These authors contributed equally: Stefano Boccaletti, Baruch Barzel. ✉e-mail: xiyunzhang@jnu.edu.cn

transmissibility fitness space. This allows us to construct a tractable analytical framework by which to predict the mutation/selection dynamics of a spreading pathogen. In this framework, mutations occur during the intra-host replication process[1–4]. Selection for transmissibility, on the other hand, is driven by the inter-host contagion dynamics, as fitter strains proliferate more rapidly and take over the available susceptible population. Together, these two processes may lead to a *mutation-driven phase*, whose boundaries and transition dynamics we pursue below. In this phase an initially non-pandemic pathogen breaks through to reach an evolved pandemic state.

We find that besides the classic epidemiological parameters, i.e. infection/recovery rates, two additional components factor in the observed mutation rate $\sigma$, quantifying the intra-host evolutionary timescales, and the fraction of infected individuals $\eta(t)$, which determines the likelihood of a critical mutation to occur within the relevant time-frame. Therefore, as opposed to classic pandemic transitions, which depend solely on the epidemiological parameters[9–13], here the time-dependent prevalence of the pathogen $\eta(t)$ and its replication stability, both have direct impact on its anticipated spread.

### Evolving pathogens and network contagion

Consider a pathogen spreading across a social network $A_{ij}$. Once contracted by a specific host $i$, it begins to replicate within $i$'s body, until, after an average of $\rho$ replication cycles, one (or more) of its offspring is transferred via infection to one of $i$'s network neighbors $j$. Throughout this in-host replication, the pathogen may undergo mutations, and hence $j$ may potentially be infected by a different strain than the one originally contracted by $i$. To track this process[14–20] we consider the different strains $\mu$ and the probabilities $M_{\mu\nu}$ that upon replication a strain $\mu$ will mutate into an offspring $\nu$ (Fig. 1a–c).

Each strain $\mu$ is characterized by two distinct fitness parameters. The *intra-host fitness* $\varphi_{\mu}(i)$ quantifies its replication rate within the host $i$. This parameter is affected by $i$'s internal biological environment, e.g., $i$'s immune response, body temperature and so on. Strains with higher $\varphi_{\mu}(i)$ multiply more efficiently within the host's body, and hence gain higher prevalence with each replication cycle. Complementing $\varphi_{\mu}(i)$ is the *inter-host fitness* $\psi_{\mu}$, which characterizes the pathogen's capacity to spread *between* hosts. This parameter, independent of $i$, increases if $\mu$ has phenotypical traits that enhance its tranmissibility, for example, an extended survival time outside the host's body[21], or a longer pre-symptomatic stage[22,23], both of which offer more opportunities for an $i \to j$ transmission.

Within the host, this results in a mutation and selection process that favors high $\varphi_{\mu}(i)$, as, indeed, fitter strains benefit from an increased intra-host growth rate. Consequently, after an average of $\rho$ in-host replication cycles, $i$'s pathogen population reaches a unique *multistrain* $\mathcal{Z}_i$[20], capturing a pathogen composition with a fraction $Z_{\mu}(i)$ of the strain $\mu$ (Fig. 1b). This composition is determined by the interplay between the pathogen's intrinsic mutation rates ($M_{\mu\nu}$), and its intra-host fitness landscape ($\varphi_{\mu}(i)$). The multistrain $\mathcal{Z}_i$ is characterized by the fitness distribution

$$f_i(\varphi,\psi) = \sum_{\substack{\varphi_{\mu}(i)=\varphi \\ \psi_{\mu}=\psi}} Z_{\mu}(i), \tag{1}$$

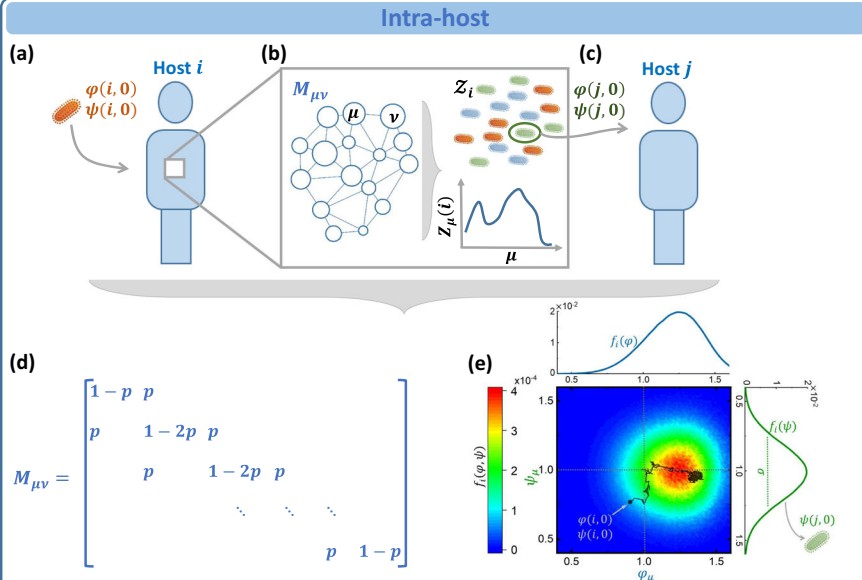

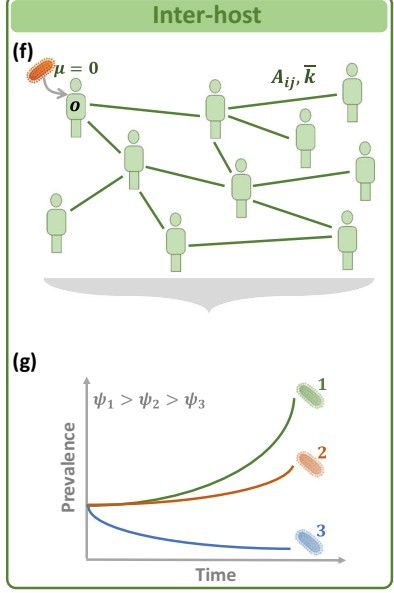

**Fig. 1 | The interplay between evolutionary and epidemiological dynamics.**
**a** Host $i$ contracts a pathogen (orange) with initial fitness $\varphi(i, 0)$, $\psi(i, 0)$. **b** Within $i$ the pathogen replicates, undergoing mutation and selection. The potential transitions between strains $\mu \to \nu$ is governed by $M_{\mu\nu}$. Strains with higher intra-host fitness ($\varphi_{\mu}(i)$) replicate at a higher rate. After $\rho$ replication cycles the pathogen population within $i$ takes the form of a multistrain $\mathcal{Z}_i$, in which there is a fraction $Z_{\mu}(i)$ of the strain $\mu$. **c** Upon transmission $i \to j$, a single (or few) individual pathogens are sampled from $\mathcal{Z}_i$, instigating again, $j$'s intra-host dynamics, this time staring from initial fitness $\varphi(j, 0)$, $\psi(j, 0)$. **d** The transition matrix $M_{\mu\nu}$. Mutations enable transition with probability $0 \le p \le 1$ between adjacent strains. The larger is $p$, the less stable is the pathogen. **e** The fitness distribution $f_i(\varphi, \psi)$ in (1) following $\rho = 25,000$ replication/mutation cycles initiated at an arbitrary $\varphi(i, 0)$, $\psi(i, 0)$ (grey dot). Each lineage captures a random walk in fitness space (black path), resulting in the density function $f_i(\varphi, \psi)$, describing $i$'s multistrain $\mathcal{Z}_i$. The intra-host fitness distribution $f_i(\varphi)$ (top, blue) is skewed in favor of fitter strains, which replicate more efficiently within $i$. The inter-host distribution, $f_i(\psi)$ (right, green), on the other hand follows a zero-mean normal distribution, as, indeed, there is no intra-host selection for high $\psi_{\mu}$. During infection the fitness of the transmitted pathogen $\psi(j, 0)$ (green) is extracted from $f_i(\psi)$. The variance $\sigma^2$ determines the size of the evolutionary gap $\Delta\psi$ of Eq. (10) between infection and transmission. **f** The inter-host dynamics is captured by network epidemic spreading, starting from an infection at node $o$ by the wild-type $\mu = 0$. As the pathogen propagates along the network $A_{ij}$ it undergoes random shifts in its observed inter-host fitness $\psi_{\mu}$. **g** The network spreading dynamics give rise to selection for higher $\psi_{\mu}$. Here, strain 1 (green), the fittest of the three, is more transmissible, and therefore, exhibits an exponentially growing dominance over the observed pathogen population.

capturing the probability that a randomly selected pathogen from $\mathcal{Z}_i$ has intra and inter-host fitness $\varphi$ and $\psi$, respectively. Upon transmission $i \to j$, a single pathogen is sampled from (1), initiating $j$'s intra-host evolutionary dynamics with (Fig. 1c)

$$\varphi(j,0) \sim f_i(\varphi) \qquad \psi(j,0) \sim f_i(\psi). \qquad (2)$$

Here $\varphi(j,0)$ and $\psi(j,0)$ denote the fitness parameters of $j$'s originally contracted pathogen, and $f_i(\varphi) = \sum_\psi = f_i(\varphi,\psi), f_i(\psi) = \sum_\varphi = f_i(\varphi,\psi)$ represent the marginal probabilities of (1). Hence, Eq. (2) describes a randomly selected pathogen from $\mathcal{Z}_i$ that is transmitted from $i$ to $j$, with initial fitness values $\varphi(j,0)$ and $\psi(j,0)$. From this starting point $j$'s own intra-host dynamics commences, resulting in a potential secondary transmission, this time extracted from $\mathcal{Z}_j$.

Taken together, as the process unfolds - from $i$'s exposure, through its intra-host replication, mutation, selection, and eventually, transmission to $j$ - we observe potential shifts in both the intra and the inter-host fitness of the pathogen population, such that

$$\varphi(j,0) = \varphi(i,0) + \Delta\varphi \qquad (3)$$

$$\psi(j,0) = \psi(i,0) + \Delta\psi. \qquad (4)$$

The evolutionary steps $\Delta\varphi, \Delta\psi$ are governed by the intra-host mutation/selection processes, yet they impact the inter-host spreading dynamics through $\psi_\mu$. On the other had, as fitter strains (higher $\psi_\mu$) spread more efficiently, the inter-host contagion governs the $\psi$-selection process. Below we analyze the resulting interplay between these intra and inter-host patterns of pathogen spread (Fig. 1d).

## Results

To observe the spreading patterns under pathogen evolution we consider the susceptible-infected-recovered[5] (SIR) dynamics on a random network $A_{ij}$. In the classic SIR formulation the spreading dynamics are governed by three parameters: the recovery rate $\alpha$, the infection rate $\beta$, and $A_{ij}$'s average degree $\bar{k}$. Together, they provide $R_0 = \bar{k}\beta/\alpha$, the *reproduction number*, which determines the state of the system[24–28] - pandemic ($R_0 \geq 1$) or infection-free ($R_0 < 1$).

In the presence of mutations, however, these parameter may change over the course of the spread, with each strain $\mu$ characterized by its own specific $\alpha_\mu, \beta_\mu$, and hence its unique reproduction number

$$R_\mu = \psi_\mu R_0. \qquad (5)$$

Here $\psi_\mu$, the inter-host fitness of the $\mu$-strain, is defined as

$$\psi_\mu = \frac{\beta_\mu/\beta_0}{\alpha_\mu/\alpha_0} \qquad (6)$$

where $\alpha_0, \beta_0$ are the recovery/infection rates of the wild-type. Therefore, the wild-type $\mu = 0$ has $\psi_0 = 1$, and consequently a reproduction number of precisely $R_0$. As the spread progresses, mutations may push $\psi_\mu$ below or above this value, by introducing strains with varying $\alpha_\mu, \beta_\mu$. The higher is $\psi_\mu$ the greater is $R_\mu$, and hence the fitter is the strain for inter-host transmission (Fig. 1g).

We can now model the SIR dynamics by tracking the infection, recovery and mutation processes. For the process of infection we write (Fig. 1a–c)

$$I_i + S_j \xrightarrow{\beta(j,0)} I_i + I_j \qquad (7)$$

$$\psi(j,0) = \max(\psi(i,0) + \Delta\psi, 0), \qquad (8)$$

in which a susceptible ($S$) individual $j$ contracts the pathogen from their infected ($I$) neighbor $i$, leading to both $i$ and $j$ becoming infected. The rate of the process in (7), $\beta(j,0)$, represents the infection rate of the transmitted strain, i.e. the selected pathogen from $\mathcal{Z}_i$, which will now begin to replicate and potentially mutate within $j$. The spreading fitness of $j$'s contracted strain, $\psi(j,0)$ in (8), is extracted from Eq. (4), with the max function conditioning that $\psi(j,0) \geq 0$, i.e. avoiding negative fitness. Next we consider the process of recovery

$$I_j \xrightarrow{\alpha_j} R_j, \qquad (9)$$

where an infected host transitions to the recovered ($R$) state. Here $\alpha_j$ represents the effective recovery rate of $j$'s multistrain $\mathcal{Z}_j$, whose composition results from $j$'s intra-host mutation/selection dynamics. The initial outbreak occurs at a randomly selected node $o$, with $\psi(o,0) = 1$, the wild-type spreading fitness (Fig. 1d), from which point fitness is gained/lost via (8) upon each instance of infection.

### Evolutionary dynamics

To complete the processes (7–9) we now evaluate the magnitude of the evolutionary *jumps* $\Delta\psi$ at each infection cycle. We therefore, in Fig. 1e, f, model a single instance of intra-host evolutionary dynamics, from initial exposure to transmission. Assigning $\mu = 0$ to represent the wild-type, we index all pathogen strains based on genetic similarity, such that $\mu = 1$ is closest to the wild-type, and as $\mu$ is increased we reach more highly mutated variants. Prohibiting direct mutation between highly distant strains, the transition matrix $M_{\mu\nu} \to 0$ in case $\mu$ and $\nu$ are far apart. Here we only allow transitions between directly neighboring strains, taking $M_{\mu\nu} = p$ in case $\nu = \mu \pm 1, M_{\mu\nu} = 1 - 2p$ if $\mu = \nu$ and $M_{\mu\nu} = 0$ otherwise (Fig. 1e). The parameter $p$, therefore, characterizes the stability of the pathogen, quantifying the expected frequency of mutations, where $p \to 0$ indicates the limit of no evolution at all.

The fitness parameters for each strain are extracted from a pair of normal distribution $\varphi_\mu(i) \sim \mathcal{N}(1,\sigma_\varphi^2)$ and $\psi_\mu \sim \mathcal{N}(1,\sigma_\psi^2)$, such that every strain is independently assigned an intra and inter-host fitness. This independence captures the fact that the intra-host environment is fundamentally distinct from that of the inter-host transmission, and hence the phenotypic traits enhancing $\varphi_\mu(i)$ and $\psi_\mu$ are, by and large, uncorrelated. In Supplementary Section 4 we also examine the case where the two *are* related.

To observe the evolutionary dynamics, at $t = 0$ we infect $i$ with a specific strain ($\varphi(i,0), \psi(i,0)$, Fig. 1f, grey dot). As the pathogen multiplies within $i$, fitter strains are selected and the composition of the pathogen population shifts in favor of strains with greater $\varphi_\mu(i)$. In this process of mutation/replication, the composition of $\psi_\mu$ changes as well. After $\rho$ reproduction cycles, we extract the two fitness distributions of Eq. (1) as obtained from the resulting multistrain $\mathcal{Z}_i$: As expected, we find that $f_i(\varphi)$, the intra-host fitness, follows a shifted normal distribution, with an average fitness greater than the initial $\varphi(i,0)$ (Fig. 1f, blue). This is a direct consequence of the intra-host selection process, which favors the rapidly replicating variants (large $\varphi_\mu(i)$).

The inter-host fitness, on the other hand, lacking selection within the host, drifts up and down at random (grey path), resulting in $f_i(\psi) \sim \mathcal{N}(\psi(i,0),\sigma^2)$, a zero-mean normal distribution with variance $\sigma^2$. Upon transmission to $j$, $\psi(j,0)$ in (8) is extracted precisely from this resulting normal distribution (Fig. 1f, green). Mathematically, this process is mapped to an effective random walk beginning at the initial starting point $\psi(i,0)$, and taking random steps of size $\sim \sigma$.

This captures our first key observation, that the detailed mutation/selection process taking place within the host can be reduced to an effective random walk in $\psi$-space. We can now use Eq. (4) to write

$\Delta\psi = \psi(j,0) - \psi(i,0)$, which using the above observation, that $\psi(j,0) \sim \mathcal{N}(\psi(i,0), \sigma^2)$, provides us with

$$\Delta\psi \sim \mathcal{N}(0, \sigma^2) \tag{10}$$

a zero-mean normal distribution, whose variance depends on the *intra*-host evolutionary dynamics. Hence, the intra-host mutation/selection process condenses into the single parameter $\sigma$ in (10) that helps characterize the observed *inter*-host evolutionary dynamics. In the limit where $\sigma = 0$, we have $\Delta\psi = 0$ in (8) at all times, mutations are suppressed, and Eqs. (7–9) converge to the classic SIR model, with a stable $R_0$. In contrast, as $\sigma$ is increased, significant $\psi$-mutations become more frequent and the inter-host fitness rapidly evolves. We therefore, below, vary $\sigma$ to control the *mutation rate* of the spreading pathogens.

Taken together, our modeling framework accounts for the dynamics of infection and recovery (SIR) under the effect of pathogen mutation. As the spread progresses, pathogens evolve via Eq. (8), blindly altering their inter-host fitness at random, as, indeed, the intra-host selection has no bearing on $\psi_\mu$. The inter-host dynamics, however, will favor positive $\psi$-mutations, in which $\Delta\psi > 0$. This is because such mutations lead to greater $R_\mu$ in (5), and hence to higher transmissibility. This, in turn, translates to higher prevalence, allowing fitter strains to eventually dominate the population (Fig. 1g).

### Critical mutation

Consider an outbreak of a wild-type with $R_0 < 1$, i.e. below the epidemic threshold. This can be either due to the pathogen's initial sub-pandemic parameters, or a result of mitigation, e.g., social distancing to reduce $\beta_0$. In the classic SIR formulation, such pathogen will fail to penetrate the network. However, in the presence of mutations ($\sigma > 0$ in (10)) the pathogen may potentially undergo selection, reach $R_\mu > 1$, and from that time onward begin to proliferate. This represents a *critical mutation*, which, using (5), translates to

$$\psi_c = \frac{1}{R_0}, \tag{11}$$

the *critical fitness* that, once crossed, may lead an initially non-pandemic pathogen to become pandemic. The smaller is $R_0$ the higher is $\psi_c$, as, indeed, weakly transmissible pathogens require a longer evolutionary path to turn pandemic. Next, we analyze the spreading patterns of our evolving pathogens, seeking the conditions for the appearance of such a critical mutation.

### Phase-diagram of evolving pathogens

To examine the behavior of (7–10) we constructed an Erdős-Rényi (ER) network $A_{ij}$ with $N = 5000$ nodes and $\bar{k} = 15$, providing a testing ground upon which we incorporate a series of epidemic scenarios (Fig. 2). Each scenario is characterized by a different selection of our model's three epidemiological parameters: $\alpha_0, \beta_0$, which determine the wild-type's reproduction $R_0$, and $\sigma$, which controls the effective inter-host mutation rate. We then follow the spread by measuring the prevalence $\eta(t)$, which monitors the fraction of infected individuals vs. time ($0 \le \eta(t) \le 1$). We also track the pathogen's evolution via the population averaged inter-host fitness $\bar{\psi}(t) = (1/N)\sum_{i=1}^{N} \psi(i,0)$, i.e. the average fitness over all transmitted pathogens. We observe several potential spreading patterns:

**Pandemic (Fig. 2a, b red).** In our first scenario we set $\alpha_0 = 0.1$, $\beta_0 = 8 \times 10^{-3}$ and $\sigma = 2 \times 10^{-2}$. This captures a pandemic wild-type, which, using $\bar{k} = 15$, has $R_0 = 1.2 > 1$, namely it can spread even without mutation. Indeed $\eta(t)$ rapidly climbs to gain macroscopic coverage,

congruent with the prediction of the classic SIR model, only this time instead of a single infection wave, we observe a secondary peak, due to the introduction of fitter variants, that help increase $\bar{\psi}(t)$. Such successive waves of infection, driven by pathogen evolution, have, in fact, being observed in the spread of SARS-CoV-2 [29–34].

**Mutation-driven (Fig. 2c, d green).** Next we reduce the wild-type infection rate to $\beta_0 = 1.67 \times 10^{-3}$, an initial reproduction of $R_0 = 0.25 < 1$. This describes a pathogen whose transmissibility is significantly below the epidemic threshold, and therefore, following the initial outbreak we observe a decline in $\eta(t)$, which by $t \sim 30$ almost approaches zero, as the disease seems to be tapering off (inset). In this scenario, however, we set a faster mutation rate $\sigma = 2$. As a result, despite the initial remission, at around $t \sim 5$, the pathogen undergoes a critical mutation as $\bar{\psi}(t)$ crosses the critical $\psi_c = 1/R_0 = 4$ (grey dashed line) and transitions into the pandemic regime. Consequently, $\eta(t)$ changes course, the disease reemerges and the mutated pathogens successfully spreads.

**Infection-free (Fig. 2e, f blue).** We now remain in the sub-pandemic range, with $R_0 = 0.25$, but with a much slower mutation rate, set again to $\sigma = 2 \times 10^{-2}$. As above, $\eta(t)$ declines, however the pathogen evolution is now too slow, and cannot reach critical fitness on time. Therefore, the disease fails to penetrate the network, lacking the opportunity for the emergence of a critical mutation.

Hence, the dynamics of the spread are driven by three parameters: the initial epidemiological characteristics of the wild-type, $\alpha_0$ and $\beta_0$, which determine $R_0$, and the mutation rate $\sigma$, which governs the timescale for the appearance of the critical $\psi_c$. Therefore, below, to determine the conditions for a mutation-driven contagion, we investigate the balance between the decay in $\eta(t)$ vs. the gradual increase in $\bar{\psi}(t)$.

### The mutation-driven phase

To understand the dynamics of the evolving pathogen model, we show in Supplementary Section 1.1 that at the initial stages of the spread, the prevalence $\eta(t)$ follows

$$\eta(t) = \eta(0)e^{\xi(t)}. \tag{12}$$

The time-dependent exponential rate $\xi(t)$ is determined by the epidemiological/mutation rates via

$$\xi(t) = -\alpha_0(1 - R_0)t + \frac{1}{2}\sigma^2\alpha_0^2 R_0^2 t^3, \tag{13}$$

whose two terms characterize the pre-mutated vs. post-mutated spread of the pathogen. The first term, linear in $t$, represents the initial stages of the spread, which are determined by the wild-type parameters, $\alpha_0, R_0$. For $R_0 < 1$ this describes an exponential decay, *a là* the SIR dynamics in the sub-pandemic regime. At later times, however, as $t^3$ becomes large, the second term begins to dominate, and the exponential decay is replaced by a rapid proliferation, now driven by the mutation rate $\sigma$. The transition between these two behaviors–decay vs. proliferation–occurs at $\tau_c = \sqrt{2(1-R_0)/3\alpha_0\sigma^2 R_0^2}$, which provides the anticipated timescale for the appearance of the critical mutation $\psi_c$ in (11).

This analysis portrays the mutation-driven contagion as a balance between two competing timescales: on the one hand the exponential decay of the sub-pandemic pathogen, and on the other hand the evolutionary timescale $\tau_c$ for the appearance of the critical mutation. For the evolution to *win* this race the pathogen must not vanish before

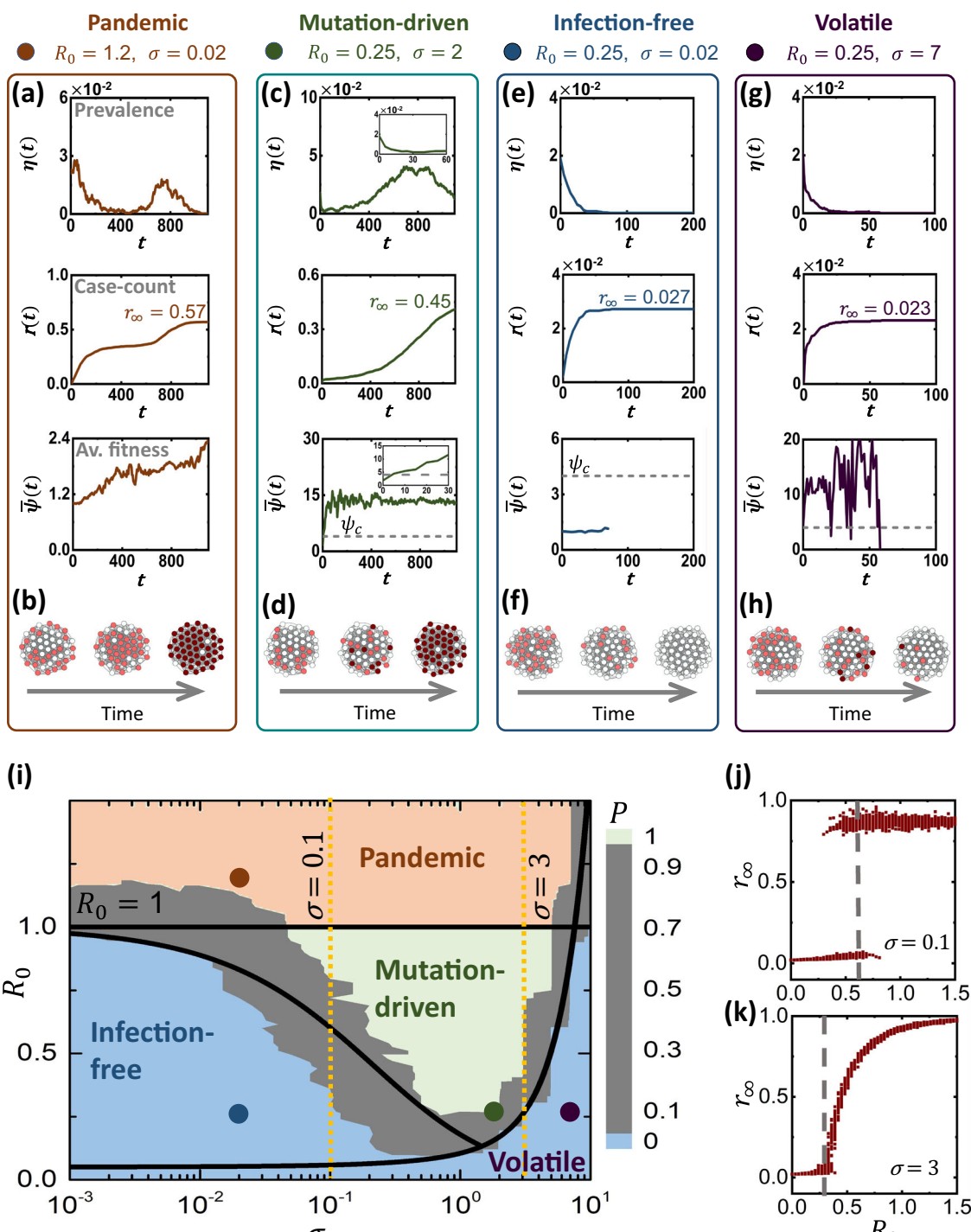

$t = \tau_c$. This imposes the condition that (Fig. 3a–c)

$$\eta(\tau_c) > \frac{1}{N}, \tag{14}$$

ensuring that at $\tau_c$ there are still one or more individuals hosting the pathogen. Indeed, $\eta(\tau_c) \le 1/N$ indicates that *on average*, at $t = \tau_c$ less than a single individual is left in the infected pool. Under this condition, the critical mutation is too late, the spread has already tapered off, and the exponential growth driven by the positive term in (13) is averted.

Taking $\eta(\tau_c)$ from (12), we can now use (14) to express the boundary of the mutation-driven phase, predicting the *critical mutation rate* as (Supplementary Section 1.1)

$$\sigma_c \propto \left( \frac{\sqrt{\alpha_0 (1 - R_0)^3}}{2 R_0} \right) \frac{1}{\ln(\mathcal{I}_0)}, \tag{15}$$

where $\mathcal{I}_0 = N\eta(t = 0)$ is the number of individuals infected at $t = 0$. Equation 15 describes the minimal mutation rate required for the wild-

**Fig. 2 | The pandemic phase-diagram. a** *Pandemic.* For $R_0 > 1$ we observe the classic pandemic phase. Prevalence $\eta(t)$ vs. $t$ (top) exhibits several waves, as more transmissible mutants emerge, and the average fitness $\overline{\psi}(t)$ (bottom) increases. **b** We observe two waves, first the wild-type (light-red) and then the fitter mutant variants (dark-red). **c** *Mutation-driven.* Setting $R_0 < 1$, but increasing $\sigma$ to 2, $\overline{\psi}(t)$ (bottom) reaches critical fitness within a short time (inset), and consequently $\eta(t)$ (top) and $r_\infty$ (middle) turn pandemic. **d** Sporadic instances of high fitness strains (dark-red nodes), take over the majority of the population. **e** *Infection-free.* Under slower mutations ($\sigma = 0.02$), $R_0$ remains sub-pandemic, $\eta(t)$ vanishes (top), $r_\infty \ll 1$ (middle) and $\overline{\psi}(t)$ remains almost constant (bottom). Hence, the pathogen fails to reach critical fitness $\psi_c$ (grey dashed-line). **f** The unfit pathogen (light red) tapers off without significant fitness gains. **g** *Volatile.* At $\sigma = 7$, $\overline{\psi}(t)$ fluctuates erratically (bottom), failing to lock-in the fitter strains. Consequently, $\eta(t)$ and $r_\infty$ fail to reach

measurable levels (top, middle). **h** High-fitness strains appear (dark red nodes), but are lost to the volatile mutations. **i** $\sigma$, $R_0$ phase-diagram. In addition to the classic pandemic-phase ($R_0 \geq 1$, red), we observe our three predicted phases: infection-free (blue) for $R_0 < 1$, small $\sigma$; volatile (blue) under large $\sigma$; and in between - mutation-driven (green), in which breakthrough mutations are almost guaranteed ($P \to 1$). The gaps between phases (grey) feature sharp transitions from $P \to 0$ to $P \to 1$, which are well-approximated by our theoretical predictions (solid black lines). The examples in panels (**a**–**h**) are shown as red, blue, green and purple dots. **j** $r_\infty$ vs. $R_0$ under $\sigma = 0.1$ (yellow dashed-line in panel (i), left). We observe a first-order pandemic transition, in which $r_\infty$ abruptly jumps from ~0 to ~1. The theoretically predicted transition is also shown (grey dashed-line). **k** The transition from volatile to mutation-driven (yellow dashed-line in panel (**i**), right) is continuous (see Methods Section for technical details).

---

type to evolve into a potentially pandemic strain. For $R_0 = 1$ it predicts $\sigma_c = 0$, as such pathogen can indeed spread even without mutation. However, as $R_0$ is decreased, the pathogen prevalence rapidly declines, and hence it must evolve at an accelerated rate to reach critical fitness on time. This is expressed in (15) by an increased $\sigma_c$, which approaches infinity as $R_0 \to 0$.

Note that the condition in (14), that at $t = \tau_c$ there is a *single* infected individual, is not necessarily stringent, and hence, should be taken more loosely as $\eta(\tau_c)$ is *of the order of* $1/N$. Indeed, even if the pathogen reaches its critical mutation when there are two or three infected individuals, due to the stochasticity of the spreading dynamics under such small numbers, pathogen extinction continues to be a probable scenario. Therefore $\sigma_c$ in (15) is only predicted upto a multiplicative constant of order unity, as signified by the $\propto$ sign, i.e. $\sigma_c$ *scales as*, not *equals to* the expression on the r.h.s. We also note that, for simplicity the derivation leading to Eq. (15) in Supplementary Section 1.1 is conducted under $\beta$-mutations, namely setting $\alpha_\mu = \alpha_0$ and allowing only the infection rate to evolve. In Supplementary Section 1.3 we further generalize the analysis, using a combination of analytical and numerical tools, to cover also $\alpha$-mutations.

To test prediction (15) we simulate in Fig. 2i an array of 1050 realizations of Eqs. (7–10), representing different epidemiological scenarios. We varied $R_0$ from 0 to 1.5, i.e. from non-transmissible to pandemic, and scanned a spectrum of mutation rates from $\sigma = 10^{-3}$ to $\sigma = 10$, spanning four orders of magnitude. Simulating each scenario 50 times we observe the probability $P$ for the disease to gain coverage. This is done by tracking the total fraction of recovered individuals $r_\infty = \int_0^\infty \eta(t)\, dt$ and counting the realizations in which $r_\infty \to 0$ vs. those where $r_\infty > 0$ (Supplementary Section 5.1). As predicted, we find that the pandemic state, classically observed only at $R_0 \geq 1$, now extends to lower $R_0$ in the presence of sufficiently rapid mutations. This gives rise to the mutation-driven phase (green), in which an initially decaying contagion suddenly turns pandemic. The boundaries of this phase (grey zone) are, indeed, well-approximated by our theoretically predicted (15), as depicted by the black solid line separating the infection-free from the mutation-driven phases.

### Explosive transition

Unlike the classic pandemic threshold, the mutation-driven phase shift is of explosive nature[35,36], capturing a first-order phase-transition (Fig. 2j). This is because once a critical mutation occurs, the pathogen gains prevalence, providing opportunities for additional mutations that push $\overline{\psi}(t)$ ever higher. This creates a positive feedback, which, in turn, leads to a discontinuous *jump* in $r_\infty$: below the boundary of (15) the total fraction of infected individuals is $r_\infty \to 0$, yet once we cross this boundary, it abruptly rises to $r_\infty \lesssim 1$.

Such discontinuous transition illustrates the risks of *soft mitigation*. Indeed, from social distancing to vaccination, our natural response in the face of a pandemic pathogen is to drive $R_0$ below the critical threshold, be it $R_0 \leq 1$ absent mutations ($\sigma = 0$), or lower in case of an evolving pathogen ($\sigma > 0$), as predicted in (15). To minimize the

ensuing socioeconomic disruption, it is tempting to tune our response, as much as we can, to push $R_0$ just below this threshold, reaching a state in which the system fluctuates around criticality[37–39]. Under a continuous transition, as classically observed for most epidemiological models, such minimal response may suffice. However, here, the abrupt first-order transition portrays a system that is highly sensitive around the phase boundary. Hence, minor excursions above the threshold can potentially lead to a sudden jump in prevalence, as they unleash a potential sequence of breakthrough mutations. We, therefore, must remain at a safe distance from criticality when responding to an evolving pathogen.

### Time sensitivity

Equation (15) shows that $\sigma_c$ depends not only on the epidemiological characteristics of the pathogen ($\alpha_0, R_0$), but also on the initial condition, here captured by the number of infected individuals $\mathcal{I}_0 = \eta(t = 0)N$. If $\mathcal{I}_0$ is large the critical rate $\sigma_c$ becomes lower, in effect expanding the bounds of the mutation-driven phase. To understand this consider the evolutionary paths followed by all pathogens as they spread. These paths represent random trajectories in *fitness space*, starting from $\psi_0 = 1$, the wild-type, and with a small probability crossing the critical fitness $\psi_c$. The more such attempts are made, the higher the chances that at least one of these paths will be successful. Therefore, a higher initial prevalence $\mathcal{I}_0$ of the pathogen increases the probability $P$ for the appearance of a critical mutation, enabling a mutation-driven phase even under a relatively small $\sigma$. In simple words, even rare mutations may occur if the initial pathogen pool ($\mathcal{I}_0$) is large enough[40–43]. Indeed, in Fig. 3d we find that the phase boundary shifts towards lower $\sigma_c$ as the initial prevalence is increased (grey shaded lines). Hence, the greater is $\mathcal{I}_0$, the broader are the conditions that enable mutation-driven contagion.

This dependence on $\mathcal{I}_0$ indicates that the risk of a critical mutation increases in case the pathogen is already widespread. For example, consider a pandemic pathogen ($R_0 > 1$) that begins to spread, exhibiting an exponentially increasing $\eta(t)$. At time $t = t_R$ we instigate our response, aiming to suppress $R_0$, e.g., via social distancing. In case of a late response ($t_R$ large), our mitigation encounters an already prevalent pathogen, with a large $\mathcal{I}_0$, and hence an increased probability to undergo a critical mutation. Therefore, early response is key, aiming to eliminate the pathogen while its population is still small, when it is still outside the bounds of the mutation-driven phase.

We emphasize, that also without mutations, the advantages of responding early are well known[44–47], primarily to help us avoid a prolonged pandemic state. The crucial difference is, that in these classic formulations pushing $R_0$ below criticality guarantees the desired transition to the infection-free phase. The only question being the temporal trajectory of this transition—prolonged or rapid. Here, however, responding late, may catch the pathogen when it has already entered a different phase, i.e. mutation-driven, and hence it is not just a matter of *how long* it will take for the pandemic to decay, but rather if it will decay at all. Indeed, our dynamics lead to a unique phenomenon

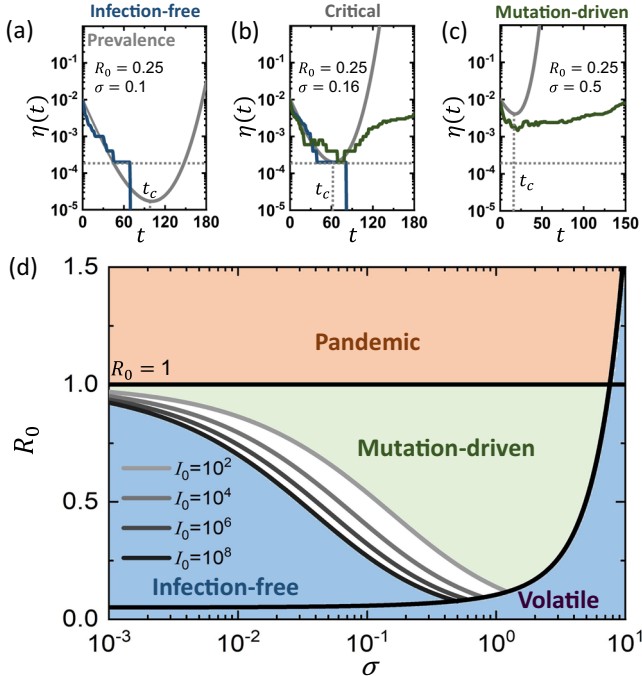

**Fig. 3 | The transition to the mutation-driven phase.** For mutation-driven contagion a critical mutation must arise before the pathogen is eliminated, namely before $\eta(t)$ crosses $1/N$ (horizontal grey dashed lines). This represents the *unit-line*, a state in which a single infected individual remains among the $N$ node population. **a** $\eta(t)$ vs. $t$ (grey solid line) as obtained from Eq. (12) in the infection-free phase ($R_0 = 0.25$, $\sigma = 0.1$). The critical mutation occurs at the minimum point ($t_c$), which is below the unit line. Therefore the pathogen is eliminated prior to the appearance of the critical mutation. Indeed, the stochastic simulation (blue solid line) approaches zero prevalence, never reaching the exponentially growing branch of $\eta(t)$, which arises at $t > t_c$. **b** Setting $\sigma = 0.16$ the system is at criticality. $\eta(t_c)$ is adjacent to the unit line, and hence we observe critical behavior: some realizations decay (blue), whereas others successfully mutate (green). At criticality, as the number of infected individuals reaches ~1, the dynamics become stochastic, and hence the outcome differs across realizations. The long term behavior of the mutation-driven branch (green) diverges from the theoretically predicted grey line, as it reaches saturation. Indeed, Eq. (12) is only designed to capture the initial stages of the spread, and disregards the exhaustion of the susceptible population that occurs at large $t$. **c** Under $\sigma = 0.5$, the system is in the mutation-driven phase, $\eta(t_c)$ is sufficiently above the unit line and the critical mutation is reached with probability $P \rightarrow 1$. **d** The phase boundary in Eq. (15) depends on the initial size of the infected population $\mathcal{I}_0$. Here we show this boundary for $\mathcal{I}_0 = 10^2, \ldots, 10^8$ (grey solid lines), finding that the larger is the infected population ($\mathcal{I}_0$), the broader is the coverage of the mutation-driven phase.

where the phase boundary itself, as expressed in (15), shifts with time, due to the accumulation of infected individuals, which increases $\mathcal{I}_0$. Therefore, the longer we wait the more likely we are to experience, at some point later in the pathogen's spreading trajectory, a game-changing mutation. In simple terms - the risk is not just a prolonged decay, but an actual reemergence of the pandemic state.

To observe this, in Fig. 4a we simulated the spread of a pandemic pathogen with $R_0 = 1.2$ (red). To counter the spread, we first employ our mitigation at $t_R = 20$ days, reducing $R_0$ by a factor of two to $R_R = 0.6$ (blue). The result is a sharp turning point in the spread, as the pathogen begins to exponentially decay, eventually eradicated within several weeks. We then examine an alternative scenario of late mitigation, this time setting $t_R = 75$ days (green). At first, the mitigation seems to be working, as $\eta(t)$ begins to rapidly decline. However, as we have now *caught* the pathogen at a higher prevalence, our mitigation is

insufficient, as $R_R = 0.6$ is now within the bounds of the mutation-driven phase. Indeed, the phase boundary has shifted with $t$, and what was a sufficient response at $t_R = 20$ is no longer enough now that $t_R = 75$. Therefore, after an initial remission, we observe a second wave of infection, due to breakthrough mutations.

Our late response, we emphasize, did not simply lead to a prolonged pandemic state, but rather to a reemergence of the pandemic. This second wave, we emphasize, occurs at $t \gg t_R$, however its seeds were already planted at the time of our initial delayed response. This delay caused the pathogen to cross the boundary into the mutation-driven phase, sealing our fate to a potential future second wave. In Fig. 4b we examine this more systematically, plotting the probability $P$ to reach critical mutation vs. our response time $t_R$, as obtained for a range of mutation rates $\sigma$. As predicted, we find that the risk for mutation driven contagion consistently increases as we delay our response.

## Hysteresis

This dependence of the phase boundary on $\mathcal{I}_0$ indicates that the transition point changes if we approach it from the infection-free state (small $\mathcal{I}_0$) or from the pandemic state (large $\mathcal{I}_0$). This captures a hysteresis phenomenon, in which the critical transition depends on the direction towards which we push $R_0$: starting from low $R_0$ we observe the transition at $R_{c,1}$; however, reversing the path, pushing $R_0$ from high to low we approach the transition from a high prevalence state, and therefore only reach the infection-free phase at $R_{c,2} < R_{c,1}$. This can be observed in case the pathogen spreads via the susceptible-infected-susceptible (SIS) dynamics, where as opposed to SIR, the pandemic fixed-point has a constant fraction of infected individuals, i.e. $\eta(t \rightarrow \infty) > 0$ (see Supplementary Section 2).

## The volatile phase

The analysis above indicates that rapid intra-host mutations, i.e. large $\sigma$, support a pathogen's ability to spread. This is thanks to the combination of fast mutation and inter-host selection, that leads to critical fitness within a sufficiently short timescale. There is, however, an upper limit to $\sigma$, beyond which mutation-driven spread cannot be sustained. Indeed, when $\sigma$ is too large the pathogen fitness $\overline{\psi}(t)$ becomes unstable. As a result, while the pathogen rapidly reaches critical fitness, the random nature of its frequent mutations, renders it unable to uphold this fitness–resulting in an irregular $\overline{\psi}(t)$, that fluctuates above and below the critical $\psi_c$ (Fig. 2g, h). The result is a second transition from mutation-driven contagion (Fig. 2i, green) to the volatile phase (blue), in which, once again, the pathogen fails to spread. The difference is that while in the infection-free phase ($\sigma$ small) contagion is avoided because mutations are too slow, here the pathogen extinction occurs because they were too rapid.

To gain deeper insight into the volatile phase, consider the natural selection process, here driven by the inter-host reproduction benefit of the fitter strains. This process is not instantaneous, and requires several reproduction instances, i.e. generations, to gain a sufficient spreading advantage. With $\sigma$ too high, natural selection is confounded, the pathogen shows no consistent gains in fitness and, as Fig. 2g indicates, $\eta(t)$ decays exponentially to zero. In Supplementary Section 1.2 we use a timescale analysis, similar to the one leading to Eq. (15), to show that the volatile phase occurs when $\sigma$ exceeds

$$\sigma_c \propto \sqrt{\frac{\alpha_0}{3}} \frac{(\psi_{max} R_0 - 1)^{\frac{3}{2}}}{R_0}. \tag{16}$$

Here $\psi_{max} = \max_{\mu}\{\psi_\mu\}$ represents the maximal inter-host fitness over all potential strains of the pathogen. This prediction is, indeed, confirmed by our simulated phase diagram in Fig. 2i (black solid line).

While related, the volatile phase is distinct from the classic error-catastrophe, that renders a pathogen non-viable due to high error

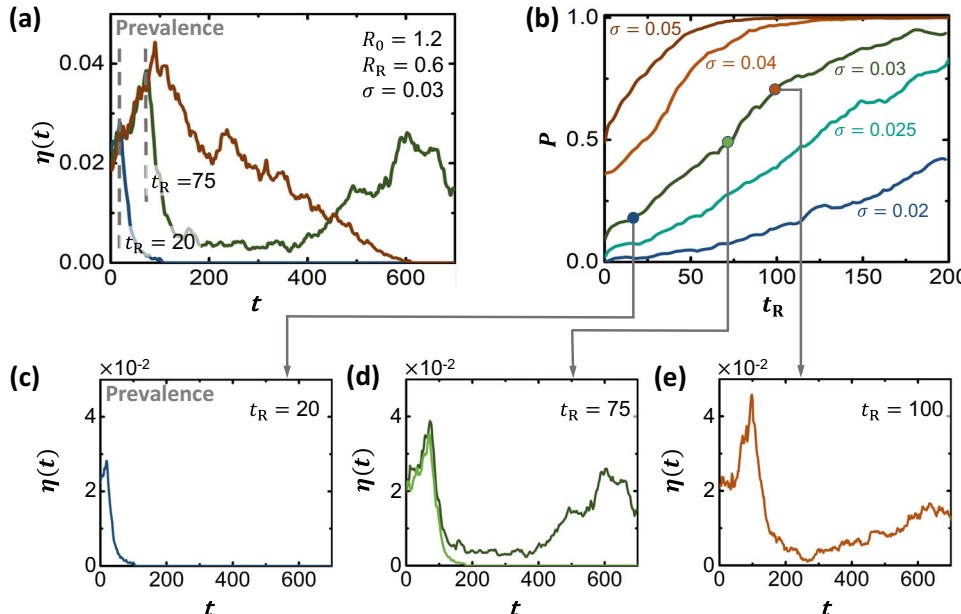

**Fig. 4 | Timing our mitigation. a** The prevalence $\eta(t)$ vs. $t$ without mitigation (red), under early mitigation (blue, $t_R = 20$) and under late mitigation (green, $t_R = 75$). Starting from $R_0 = 1.2$, $\sigma = 0.03$, the mitigation reduces $R_0$ to $R_R = 0.6$, a factor of one half. While early mitigation ensures the elimination of the pathogen (blue), under late mitigation (green), the initial decline is eventually reversed due to the appearance of a critical mutation. Note that for evolving pathogens the late mitigation does not simply prolong the pandemic, but rather, as the green curve clearly shows, causes it to enter the mutation-driven phase. Consequently, we risk a

second wave that may occur much later than $t_R$, but whose seeds were sown during our initial delayed response. **b** The probability $P$ to reach critical fitness ($\psi_c$, Eq. (11)) vs. $t_R$, as obtained for different values of $\sigma$. The later we instigate our mitigation, the higher the risk for a breakthrough mutation. **c** Setting $t_R = 20$, we have $P \to 0$, and hence a successful mitigation. **d** At $t_R = 75$ we predict $P \approx 0.5$, and hence in some realizations the pathogen is eliminated (light green), while in others, it reemerges in the form of a second pandemic wave (dark green). **e** Further delaying our response to $t_R = 100$ we enter $P \lesssim 1$, predicting an almost inevitable mitigation failure.

rates during replication[48]. Indeed, the error-catastrophe captures a state in which mutations are frequent enough to disrupt the inter-generational preservation of genetic information. Such conditions prohibit the pathogen from producing enough viable offspring, resulting in failure to sustain its population. Here, however, the spreading bottleneck is different: the pathogen *is* able to successfully replicate, but it fails to lock-in its fitness gains. Indeed, by the time natural selection, here governed by the inter-host spreading dynamics, affords the fitter strains a sufficient spreading advantage, mutations have already driven the pathogen, or better yet–its offspring, into a different area in $\psi$-space. Hence, the pathogen extinction is not due to replication failure, indeed–it replicates viably within its host, but rather to the short memory of advantageous mutations, vs. the relatively longer natural selection timescales[49–53]. Consequently, absent selection, mutations drive the pathogen erratically across fitness-space, a volatile dynamics, in which the critical mutation, despite being reached, is short-lived, overrun by new mutations, before the fitter strain can gain sufficient ground.

Our phase-diagram illustrates the different forces governing the spread of pathogens in the presence of mutations. While spread is prohibited classically for $R_0 < 1$, here we observe a mutation-driven phase, in which the disease can successfully permeate despite having an initially low reproduction rate. The conditions for this phase require a balance between three separate timescales: (i) The time for the initial outbreak $\eta(0)$ to reach near-zero prevalence $\tau_1$; (ii) The time for the pathogen to evolve beyond critical fitness $\tau_c$; (iii) The time for the natural selection to lock-in the fitter mutations $\tau_2$. Pathogens with small $R_0$, we find, can still spread provided that

$$\tau_1 > \tau_c > \tau_2, \tag{17}$$

illustrating the window of mutation-driven pandemic spread. The l.h.s. of (17) ensures that the pathogen can reach critical fitness before reaching near-zero prevalence. This gives rise to the first transition of Eq. (15), between the infection-free and the mutation-driven phases. The r.h.s. of (17) is responsible for the second transition, from mutation-driven to *volatile*. It ensures that fitter pathogens do not undergo additional mutation before they proliferate via natural selection. Therefore, we observe a Goldilocks zone, in which the mutation rate $\sigma$ is just right: on the one hand, enabling unfit pathogens to cross the Rubicon towards pandemicity, but on the other hand, avoiding aimless capricious mutations.

## Reinfection

Our discussion up to this point focused on mutations that randomly increase $R_0$ by impacting $\alpha_0$ or $\beta_0$, namely the epidemiological parameters. We now complete our analysis considering a different breakthrough mechanism of reinfection via immunity evasion. Here, a mutant variant, distinct enough from the wild-type, can reinfect an already recovered individual. This mechanism is, of course, well-studied[54–59], however, our analysis framework can help us advance by observing its outcomes under a range of variants $\mu = 0, 1, 2, \ldots$.

To examine this in a realistic setting we consider the spread of SARS-CoV-2. We therefore collected data on the COVID-19 infection cycle[60–70] (Fig. 5a): upon infection, individuals enter a pre-symptomatic state, which lasts, on average 5 days. During this period, typically within 2–4 days they begin to shed the virus and infect their network contacts (PS, purple). This continues until the onset of mild ($I_M$), severe ($I_S$) or critical ($I_C$) symptoms, at which point they mostly enter isolation and cease to spread the virus. (Of course, individuals may not comply, and hence in Supplementary Section 5.3 we examine the case where

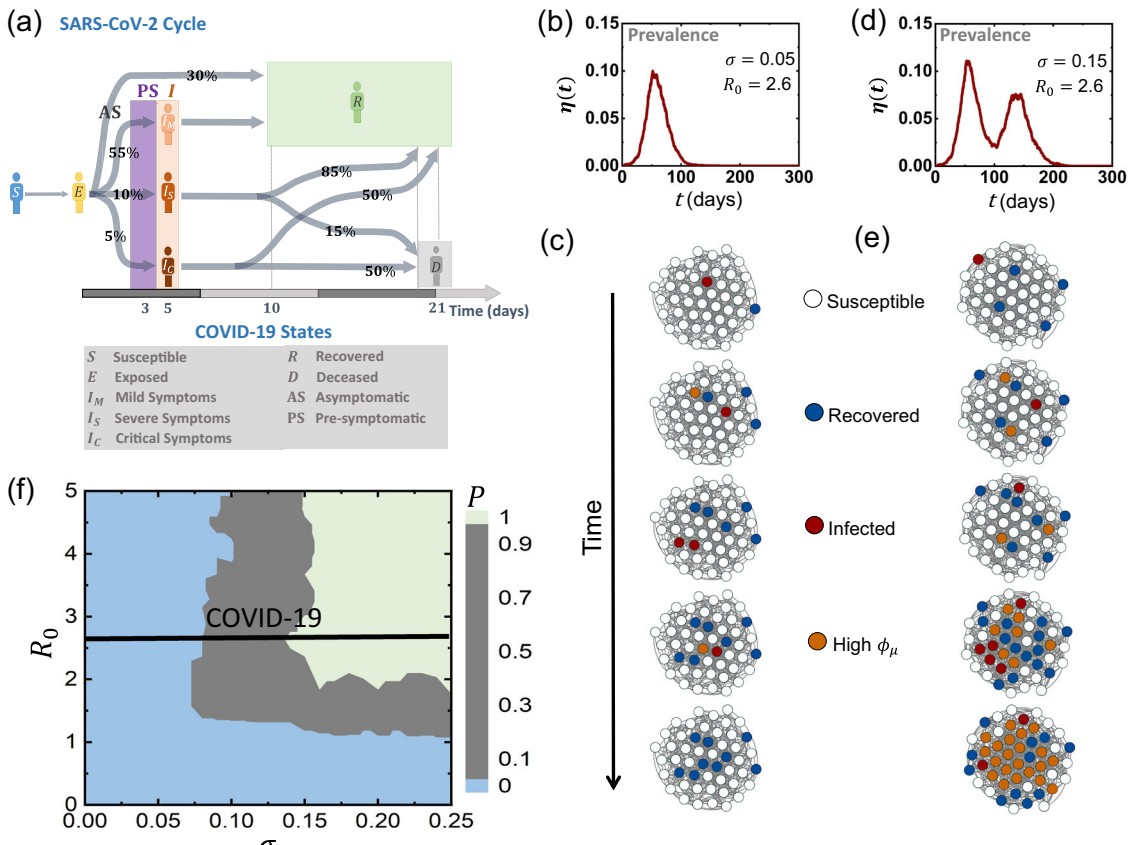

**Fig. 5 | Reemergence due to immunity evasion. a** The SARS-CoV-2 disease cycle. Upon exposure ($E$, yellow) individuals enter a pre-symptomatic phase (PS, purple), from which they later develop mild ($I_M$), severe ($I_S$) or critical ($I_C$) symptoms, determining the duration of their infected phase and their probability to recover ($R$, green) or decease ($D$, grey). The probability of all transitions appears along the arrows, and the typical time-line is shown at the bottom of the cycle. 30% of exposed individuals show no detectable symptoms at all (AS). **b, c** Under small $\sigma$ we observe a single infection wave, which tapers off as the susceptible population is exhausted. This captures the classic pandemic curve, with a single wave of infection. **d, e** Increasing $\sigma$ we now observe a second infection wave, in which a variant with sufficiently high $\phi_\mu$ (orange nodes) is able to reinfect the recovered (blue) individuals. **f** The probability $P$ for reemergence (double peaked $\eta(t)$) in function of $R_0$ and $\sigma$. For SARS-CoV-2, $R_0$ is estimated at 2.6 (black solid line); Supplementary Section 5.3.

~30% of the mildly symptomatic ($I_M$) continue to interact despite their symptoms.) In addition, a fraction (~30%) of infected individuals never go on to develop noticeable symptoms (AS, top arrow), and hence they continue to spread the virus until their full recovery ($R$), typically within ~7 days.

To evaluate the (wild-type) infection rate $\beta_0$ we used empirical data on the observed spread in 12 different countries[29] (Supplementary Section 5.3). Focusing on the early stages of the contagion, prior to the instigation of lock-downs or other mitigation schemes, we find that $\beta_0 = 5 \times 10^{-2}$ days$^{-1}$ best fits the observed spreading dynamics. This corresponds to a reproduction rate of $R_0 \approx 2.6$, congruent with existing[60,71,72] valuations of $R_0$ under COVID-19.

For the evolutionary dynamics, we consider, again, intra-host mutations, governed by $M_{\mu\nu}$ and selected via each host's idiosyncratic fitness landscape $\varphi_\mu(i)$. The result, as discussed above, is an effective random walk, starting from the wild-type $\mu = 0$, and accumulating increasing levels of genetic variation along the $\mu$-axis. Each variant $\mu$ is assigned a *reinfection fitness* $\phi_\mu$, capturing its probability to penetrate the wild-type induced immunity. The greater is $\mu$, the higher the genetic distance from the wild-type, and hence also the greater is $\phi_\mu$[16,73–75]. In our simulations we use $\phi_\mu = z^h/(z_r^h + z^h)$, where $z = \mu/\mu_{max}$ is the normalized distance ($0 \leq z \leq 1$) from the wild-type, and $z_r$ is a parameter governing the typical distance required for $\mu$ to reinfect a wild-type recovered individual. For $z \ll z_r$ we have $\phi_\mu \to 0$, i.e. no reinfection, and if $z \gg z_r$ we have $\phi_\mu \to 1$, a variant fully resistant to the wild-type antibodies.

Similarly to our previous analysis, also here, the emergence of a breakthrough mutation is driven by the timescales of the inter/intra-host dynamics. If mutations are slow compared to the spreading dynamics ($\sigma$ small) the outbreak decays before the critical mutation $z > z_r$ is reached, and we observe a single wave of infection (Fig. 5b, c). If, however, mutations are rapid ($\sigma$ large), $z$ grows fast enough to reach $z_r$ on time, and we risk a second wave with relatively high probability (Fig. 5c, d). This gives rise to the $R_0, \sigma$ risk-map, capturing the probability $P$ of a second wave, given the wild-type epidemiological parameters (encapsulated within $R_0$) and the pathogen replication stability (encapsulated within $\sigma$); Fig. 5f.

Our sigmoidal $\phi_\mu$ allows us to account for the gradual accumulation of reinfection fitness, as governed by the saturation parameter $h$. Under large $h$, $\phi_\mu$ jumps abruptly from 0 to 1 at $z = z_r$, mapping to existing frameworks in which one considers a discrete set of (typically two) variants[76–80]. Tuning $h$, however, or considering more complex $\phi_\mu$, we can take advantage of our analytical framework that helps capture the evolutionary dynamics across a continuum of variants.

## Prospects and limitations
Throughout our presentation we we have made several simplifying assumptions, selected primarily for methodical reasons, which can be relaxed to introduce more realism, where relevant. For example, our Gaussian fitness distributions $\mathcal{N}(1, \sigma_\varphi^2)$ and $\mathcal{N}(1, \sigma_\psi^2)$ (Fig. 1e) may be replaced with more complex fitness landscapes. This will impact the patterns of the observed inter-host fitness jumps, and hence replace

the simplistic zero-mean random walk of Eq. (10) by a more specified exploration of the inter-host fitness space. Another potential generalization, relevant in real-world applications, considers the fact that individuals may experience different disease cycles. For instance, in COVID-19, most individuals exhibit a limited infection time-window, while few sustain the virus for extended periods. These latter individuals, hosting the virus for many replication cycles, are the main contributors to its genetic variability[81–86]. This heterogeneity across hosts translates to a distributed $\sigma$, i.e. $\sigma_i$, a potential complication that was not considered in our implementation. Finally, in our last example we focus solely on reinfection vs. the wild-type $\mu = 0$, and hence we observed–at most–two infection waves. More generally, one may consider more complex fitness functions $\phi_{\mu\nu}$, designed to capture the probability of $\nu$ to reinfect an individual recovered from $\mu$. This, however, may require complex book-keeping of infections, which, if carried out in all detail, i.e. per each pair of $\mu, \nu$, may limit the feasibility of our framework.

## Discussion

The phase diagram of epidemic spreading is a crucial tool for forecasting and mitigating pandemic risks. First, it identifies the relevant control parameters, such as $\alpha_0, \beta_0$ and $\bar{k}$, whose values determine $R_0$. Then it predicts the phase boundaries that help us assess the state of the system–infection-free or pandemic. Finally, it provides guidelines for our response, from social distancing to reduce $\bar{k}$, through therapeutic treatment to increase $\alpha_0$ or mask wearing to suppress $\beta_0$.

Here we have shown that mutations can fundamentally change this phase diagram. Instead of just one dimension, the relevant phase space is now driven by two control parameters: $R_0$, the wild-type reproduction number, which encapsulates the inter-host epidemiological characteristics of the pathogen, and $\sigma$ the observed mutation rate, which is determined by the intra-host evolutionary dynamics. This $(\sigma, R_0)$ phase space gives rise to a mutation-driven phase, whose boundaries we predict here, observing several characteristics that stand in contrast to most known pandemic transitions. Most notably, the abrupt first-order transition, that depicts extreme sensitivity in the vicinity of the phase boundary, and the dependence of the transition point on the current prevalence $\eta(t)$, which predicts that similar responsive actions may lead to different outcomes at different stages of the spread[44]. As we are now experiencing, first hand, how novel mutations are driving the proliferation of SARS-CoV-2, we hope that these qualitative predictions will guide our response.

While the classic control parameter $R_0$ is well-understood, our analysis introduces an additional relevant parameter $\sigma$, designed to capture the observed inter-host mutation rate. This parameter emerges from the microscopic model of the intra-host replication/mutation dynamics. It is, therefore driven by the intra-host fitness landscape $\varphi_\mu(i), \psi_\mu$, its patterns of mutation transitions $M_{\mu\nu}, p$ and its replication cycle $\rho$. In Supplementary Section 5.2 we analyze the impact of each of these microscopic model characteristics on the observed mutation rate $\sigma$.

Our analysis assumed that the intra and inter host fitness parameters, $\varphi_\mu(i)$ and $\psi_\mu$, are independent of each other. Indeed, the two environments, that of the in-host replication vs. that governing host-to-host transmission, are, by and large, unrelated, and hence being fit for one has no bearing on the pathogen's fitness for the other[75,87]. One exception, is under drug-based mitigation. Drugs are designed to disrupt the intra-host replication, and by that increase the recovery rate $\alpha_0$ (reducing $R_0$), thus suppressing the inter-host transmission. This naturally creates a positive correlation between $\varphi_\mu(i)$ and $\psi_\mu$, since variants with higher drug resistance will not only be more fit for intra-host replication, but also have a lower $\bar{\alpha}_0$, and hence higher transmissibility. The observed inter-host evolutionary dynamics will no longer follow a random walk in fitness space, as in Eq. (4), but rather show a systematic bias towards higher fitness (Supplementary Section 4). With an array of new drugs, some at final stages of approval[88], being currently considered for the treatment of COVID-19, we believe that this analysis may offer key insights on the potential arms-race between our treatment and SARS-CoV-2's evolution [89].

## Methods

### Simulations

All simulations were done on a random network of $N = 5000$ nodes and $\bar{k} = 15$. The disease parameters were set to $\alpha_0 = 0.1$ and the infection rate was set variably to $\beta_0 = \alpha_0 R_0/\bar{k}$, to obtain the different values of $R_0$. The mutation rate $\sigma$ is specified in the relevant figure panels panel. In the results presented in the main text we employ $\alpha$-mutations, and set $\beta = \beta_0$ unchanged. This is complemented in Supplementary Section 3, where we examine $\beta$-mutations under fixed $\alpha = \alpha_0$. In each scenario we set the initial condition to $\eta(t = 0) = 0.02$.

### Phase-diagram

To construct the phase-diagrams in Figs. 2 and 3 we varied $R_0 \in (0, 1.5)$ and $\sigma \in (10^{-3}, 10)$, amounting to a total of 1050 distinct epidemiological scenarios, as characterized by $\alpha_0, \beta_0$ and $\sigma$. For each of these 1050 scenarios we ran 50 independent stochastic realizations, and measured the fraction $P$ that resulted in $r_\infty > \epsilon$; setting $\epsilon = 0.2$. If the majority of realizations under a specific $R_0, \sigma$ failed to spread ($r_\infty < \epsilon$) we have $P \to 0$, and yet if most reached pandemic levels ($r_\infty > \epsilon$) then $P \to 1$.

### Reporting summary

Further information on research design is available in the Nature Research Reporting Summary linked to this article.

## Data availability

Data for analyzing and extracting the COVID-19 parameters is available at https://github.com/heibaihe/EpidemicSpreadingUnderMutation.

## Code availability

All code to study, reproduce and improve the results shown here is freely available at https://github.com/heibaihe/EpidemicSpreading UnderMutation; https://doi.org/10.5281/zenodo.7092756.

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

## Acknowledgements

X.Z. thanks Dr. Xiaobo Chen, Dr. Tingting Shi, Dr. Xing Lu and Prof. Weirong Zhong for their fruitful discussions and support of the numerical simulations. This work was supported by the NNSF of China (grant No. 12105117; X.Z., and 12005079; M.Z.), the Fundamental Research Funds for Central Universities (grant No. 21621007; X.Z.), the Guangdong Basic and Applied Basic Research Foundation (grant No. 2022A1515010523; X.Z.), the Science and Technology Planning Project of Guangzhou (grant No. 202201010360; X.Z.), the funding for Scientific Research Startup of Jiangsu University (grant No. 4111710001; M.Z.), Jiangsu Specially-Appointed Professor Program (M.Z.), the Israel Science Foundation (grant No. 499/19; B.B.), the bi-national Israel-China ISF-NSFC joint research program (grant No. 3552/21; B.B.) and the Bar-Ilan University Data Science Institute grant for COVID-19 related research (B.B.).

## Author contributions

All authors jointly designed the research. X.Z., B.B. and S.B. conducted the mathematical modeling and analysis; X.Z. and Z.R. performed the numerical simulations. X.Z., B.B. and S.B. were the lead writers of the paper.

## Competing interests

The authors declare no competing interests.

## Additional information

**Correspondence and requests** for materials should be addressed to Xiyun Zhang.

