## [Peer Review File · Nature Communications]

REVIEWER COMMENTS

Reviewer #1 (Remarks to the Author):

The authors discuss the effect of virus mutations and their feedback effect on the epidemic dynamics. The subject could be of interest. However it is well known that mutations and virus propagation can interfere in a non trivial way and there are papers discussing and modeling this effect. See: e.g. (some of the paper are related with recent COVID outbreak other older describes e.g. the mutation of influenza):

Fudolig M, Howard R (2020) The local stability of a modified multi-strain SIR model for emerging viral strains. PLoS ONE 15(12). Rüdiger, S., Plietzsch, A., Sagués, F. et al. Epidemics with mutating infectivity on small-world networks. Sci Rep 10, 5919 (2020). Gubar E, Taynitskiy V, Zhu Q. Optimal Control of Heterogeneous Mutating Viruses. Games. 2018; 9(4), 103. Simon A. Rella, Yuliya A. Kulikova, Emmanouil T. Dermitzakis, Fyodor A. Kondrashov
<https://www.medrxiv.org/content/10.1101/2021.02.08.21251383v1>

The fact that mutations can remove the effects of non pharmaceutical (social distancing) and pharmaceutical interventions (vaccinations), favoring the emergence of new more infective strains is a well known fact. In this perspective a purely theoretical paper could be of interest for the broad audience of Nature Communication if it highlights properties of the infection process which have never been noticed before; in my opinion the paper describes within the considered model some interesting effects which are expected to characterize systems with virus mutations. Therefore, the paper in the present form is not suitable for publication in Nature Communications.

Moreover there are other points that should be improved before publication:

- The first part of the paper focus on the SIS model however a realistic description of an epidemic spreading is given by SIR model or its variants (e.g. the model in Fig4a). The authors should discuss which of the results in the section "Phase-diagram of evolving pathogens" could be valid even for SIR model. This is not really trivial since the effect of mutations is relevant during the whole dynamical evolution where the differences between the two models are important. Indeed only close to the threshold the models should behave in a similar way; since the mean field equations displays the same linearized behavior for $S=1$.
- The hysteresis should be discussed in a more clear way. For any I_0 if you start at very small R_0 , in the lethargic phase, the systems should go into the absorbing state where no infected nodes are present. Therefore, the system remains in the state with no infection even when R_0 becomes larger than the transition value and for arbitrary large value of R_0 . Clearly this happens if the variation of R_0 is slow enough. While for fast variations even in the lethargic phase the number of infected agents may not vanishes so that for larger R_0 the system may enter in a active phase. Therefore some discussion on the speed of the variation of R_0 should be introduced. Moreover, the presence of hysteresis should be one of the aspect that is very affected by the difference between SIS and SIR models.

- I cant find the value of the initial conditions $\rho(0)$ or I_0 in Fig1. This is a very important parameter.
- In the SIR model the importance of early adoption of containment measures is well known fact even in absence of mutations: Phys. Rev. E 102, 020301 (2020)

Reviewer #2 (Remarks to the Author):

In the current study Zhang et al. present a simple, but elegant model of pathogen transmission and concurrent evolution. Through a combination of theory and simulations they find (admittedly unsurprising and known) that vaccination can lead to vaccine escape mutants and that rapid vaccination plus NPIs can limit escape mutants. They also find regimes where R_0 can be under 1 but with a high enough mutation rate there can still be sustained transmission (which is a really interesting result), ditto for the hysteresis results (these are very important results). This is a nice piece with some ... let's say interesting ... choices of language. There are very few evolution-transmission models and this one is particularly good. I have only a few comments/questions that should be easy for the authors to address.

Comments/Questions

- First, how did this collaboration come about? The institution list reads like a United Nations of science: China, Italy, Russia, Spain, and Israel? Kudos on the collaboration!
- The language is colloquial in places for a scientific paper. Some examples: "drug-persistence" should be "drug-resistance", I believe. P3: referring to $R_0 < 1$ as the "healthy" state seems odd to me, not sure of the replacement, though, maybe increasing and decreasing states? P3: "unmutated" is called "wild-type". I'm not a fan of the use of "lethargic" for the low-mutation, $R_0 < 1$ state (best not to anthropomorphize the virus). Maybe "shrinking"? While I like "Goldilocks zone" it probably isn't appropriate in this setting.
- P6: I wouldn't say that R_0 of 1.5 is "highly contagious". Measles, with $R_0 \sim 18$, is "highly contagious".
- P9: it's a strong assumption to assume that all symptomatic individuals isolate and cease to transmit that needs better justification. The 30% asymptomatic needs a citation, I think it's more like 50%.
- Supplement Figure 2: I really like how the authors estimate α from the data, but two things, one, I assume the authors mean cumulative cases not prevalence; and two, why just take the average? It would be nice to also do the low and high estimates.

Reviewer #3 (Remarks to the Author):

Zhang et al. firstly present a simple SIS model for pathogen spreading on a static contact network (Erdős-Rényi), which is extended for a stochastic term ("evolution") that changes the models' parameters randomly. They analyze the model's behavior with regards to its infinite time behavior (pandemic will establish, vanish). They then extend the model by introducing several 'infected compartments' to apply the model to COVID and make several claims regarding mitigation and vaccination strategies. The latter model is however insufficiently described in technical details.

The authors are quite obviously not very well acquainted with the field of evolutionary dynamics, which in itself is ok, but there is a complete disregard of what has already been achieved in the field in terms of extremely profound mathematical theory. This weakens the paper considerably, firstly, because some assumptions made about how pathogens evolve are fundamentally wrong and secondly the field of evolutionary dynamics has already progressed way further than where this manuscript stands (examples will follow). Overall the manuscript, in my view, lacks a remarkable innovation, is overall not very well researched (literature; what have others done?) and the conclusions are pretty trivial (one would not need a model for that, but could always come up with one that yields the presented conclusions). The used terminology is problematic. Unfortunately, all in all I have to recommend rejection.

Specific comments:

-The paper lacks references, as well as critical discussion with regards to works in the field of evolutionary dynamics. I suggest to consult the works of Martin Novak, Manfred Eigen and Peter Schuster as a STARTING POINT and put less focus on works within the network inspired physics community. Along these lines, the authors may want to differentiate between intra-host evolution and an observed 'evolution' at the population level (inter-patient). For viruses, to correctly model their evolution at the population level, this is an extremely important issue.

Introduction:

1. In the first paragraph, the authors mix up evolution of species (vast time-scales) and evolution of viruses. Also, to be exact: It is not the prevalence that causes mutations, but the virus replication within the host(s) (more replication -> more mutations may happen).
2. Refs. 1-6 in the Introduction (page 1) do not support the claims made by the authors when they cite them. Same with Refs 7-13.

3. Page 2: I find many incorrect statements and presumptions here:

-Ref 28-32: Again, it may be of advantage to move out of the 'spreading phenomena on networks' bubble.

-3rd paragraph: the understanding of "responsive evolution" is entirely wrong. For example: A pathogen does not become more transmissible because social distancing is enforced. It always tends to become more transmissible (or persistent), if it can. Also, the authors may want to consult some relevant works regarding evolutionary escape from biomedical intervention (e.g. PMID: 14728779 and citing references)

-4th paragraph: This paragraph lacks detail. Again: it would be helpful to differentiate between intra-host evolution and observable evolution at the level of the population (between individuals that are infected). At the intra-host level, I would consider evolution as a random walk in fitness space. At the inter-host level things are a bit more complicated, but inherently depend on within-host viral dynamics and e.g. things like quasispecies dynamics and "transmission bottlenecks".

-last paragraph: There is a lack of understanding of how vaccines work and how resistance against vaccines can develop. Foremost, vaccines induce a spectrum of antibodies, similar to poly-pharmacology with a spectrum of targets. Thus, there is not 'one mutation that induces resistance', but many have to co-occur for vaccines to lose efficacy. Secondly, resistance emergence occurs while virus replicates within an infected individual. If a vaccine is applied and it protects, there is no virus replication, because the person does not get infected in the first place, hence resistance may not emerge. In a non-vaccinated individual there would not be selective pressure for resistance to emerge (COVID exception below). If a vaccine gets applied after infection, resistance may eventually get selected. This paper <http://dx.doi.org/10.1098/rspb.2016.2562> gives a good overview.

4. However, and most importantly: In Covid a natural infection and vaccines induce the same spectrum of immune responses/antibodies. Thus, in a population that has experiences infection, a virus gets a transmission advantage if it escapes natural (and therefore also vaccine-induced) immune responses. The authors' model completely fails to capture this phenomena.

Results:

5. In my opinion, an SIS model is completely the wrong approach to model transmission dynamics and pathogen-evolution, w.r.t. SARS-CoV-2. After an infection, the infected person is (at least for some time) not-susceptible to infection anymore (reconvalescent). My suspicion for the authors choosing this model is to avoid an exhaustion of susceptibles. However, there are better ways to do that.

6. The evolutionary model (a zero-centered Gaussian shift in fitness space) denotes an incorrect model of viral evolution, at least at the population level. There is a nice theoretical paper by Martin Novak, illustrating that 'downwards' moves in fitness space are extremely unlikely.

7. Again, the authors' model does not capture SARS-CoV-2's evolutionary dynamics, which is largely driven by the fact that a mutation has to become selected within the infected host, before transmission.

8. Terminology: Much of the putatively “surprising” conclusions from the model are because of poor terminology. If you correctly define

a. $R(t_0)$; the reproduction number at the onset of infection (or of the wildtype)

b. $R_i(t)$ reproduction number of mutant i at t

c. $\langle R(t) \rangle$ the average reproduction number of the population at t

You recover all well-known phenomena.

Moreover, some other terminology is quite sloppy.

9. The fact that the probability to evolve resistance/fitness depends on the population size is not fundamentally new. You should however, be a bit more precise: It depends on the number of virus replication events: many individuals being infected, or few individuals being infected for extended durations. From this it is fairly straight-forward to derive mitigation recommendations.

10. Page 7: I do not understand where the claim “Most mitigation strategies seek a minimal approach, aiming to drive R_0 just below unity” stems from. In my understanding there is a difference of what we seek to do and what we can do.

11. Volatile phase: This is not new: I strongly suggest that the authors consult the works that Manfred Eigen on quasispecies theory and error catastrophe, etc... (!developed in the 1970ies!).

12. Page 9 first paragraph: Refs 51-53 are quite unsuited here. There are good models of intra-host SARS-CoV-2 dynamics.

13. COVID model: It is unclear whether the recovered re-enter the susceptible population, and if not if there are any effects of susceptible exhaustion.

Small things:

There is inconsistent use of parameter names, page 9, middle, vaccination rate.

To all Referees

We wish to thank all Referees for their thorough and deep reading of our paper. We were especially excited to read Referee 2's assessment of our results as *really interesting* and *important*, and our modeling framework as *particularly good*. At the same time, we were also intrigued by the critique of all Referees, which prompted us to thoroughly revise our presentation, linking it to the existing *philosophy* of evolutionary dynamics, and highlighting our paper's novelty within this context.

We have now addressed all the comments made by the Referees, resulting in what we believe is a significantly improved paper. Before we begin with our detailed response, let us first summarize the *big picture* of our revisions:

Modeling framework. Following Referee 3's comments we have fundamentally reformulated our modeling of the evolutionary dynamics to bring it on par with the existing knowledge of the field.

Presentation. The reports helped us elucidate our *specific* novelty within the state-of-the-art, which is three-faceted: *Methodological* – showing how to treat the interplay between the epidemic and the evolutionary dynamics. *Fundamental* – predicting the precise pandemic phase-diagram in the presence of pathogen evolution. *Applicable* – Using our framework to analyze the potential for breakthrough mutations under COVID-19.

Breadth. Encouraged by the Referee's appreciation of our unique predictions, such as the hysteresis and the mutation-driven pandemic transition, we have expanded, in the present submission, our set of applications. Hence, our paper now includes different disease models (SIS, SIR, COVID-19), coupled with different forms of evolutionary dynamics, *e.g.*, our newly added biased random walk, as prompted by Referees 2 and 3.

Practical relevance. We deepened our discussion on mitigation guidelines, obtaining specific quantitative results. For example, we obtain the risk for a critical *game-changing* mutation in function of our response delay time (see, for example, newly added **Fig. 4**).

Taken together, we are quite certain that the Referees will find our paper much more focused, attuned to our current understanding of evolutionary processes, and strengthened in terms of its quantitative results. Please find below our detailed response to all of the Referees' comments.

Thanks,
The authors

Reviewer #1

1. Comment

The authors discuss the effect of virus mutations and their feedback effect on the epidemic dynamics. The subject could be of interest. However it is well known that mutations and virus propagation can interfere in a non trivial way and there are papers discussing and modeling this effect. See: e.g. (some of the paper are related with recent COVID outbreak other older describes e.g. the mutation of influenza): Fudolig M, Howard R (2020) The local stability of a modified multi-strain SIR model for emerging viral strains. PLoS ONE 15(12). Rüdiger, S., Plietzsch, A., Sagués, F. et al. Epidemics with mutating infectivity on small-world networks. Sci Rep 10, 5919 (2020). Gubar E, Taynitskiy V, Zhu Q. Optimal Control of Heterogeneous Mutating Viruses. Games. 2018; 9(4), 103. Simon A. Rella, Yuliya A. Kulikova, Emmanouil T. Dermitzakis, Fyodor A. Kondrashov

<https://www.medrxiv.org/content/10.1101/2021.02.08.21251383v1>

The fact that mutations can remove the effects of non pharmaceutical (social distancing) and pharmaceutical interventions (vaccinations), favoring the emergence of new more infective strains is a well known fact. In this perspective a purely theoretical paper could be of interest for the broad audience of Nature Communication if it highlights properties of the infection process which have never been noticed before; in my opinion the paper describes within the considered model some interesting effects which are expected to characterize systems with virus mutations. Therefore, the paper in the present form is not suitable for publication in Nature Communications.

Response

We wish to thank the Referee for this summary of our work, and especially for helping us place it in the broader context of related contributions. Specifically, we appreciate the inclusion of these highly relevant references, which are, we note, all extremely recent – in our view indicating the timeliness of the subject matter.

This way or another, we agree, there are, indeed, several previous investigations into the impact of mutations on pathogen spreading, especially in the context of viral diseases, due to the relatively high mutation rate of viruses. We have therefore, in our revised submission, made special effort to elucidate our precise novelty within this landscape.

Let us summarize the main points, that we hope are now better articulated in our current submission:

The challenge. As the Referee notes, it is well-known, and – in fact – quite intuitive, the mutations can help a sub-pandemic pathogen to break through and gain prevalence. The precise conditions for this to happen, and the dynamics by which such a process unfolds, are however, not fully mapped. The challenge is, in our understanding, that the two governing processes that control these dynamics are of highly distinct mathematical nature. On the one hand, the epidemic spreading is described by compartmental models – SIS, SIR or more elaborate disease cycles, e.g., our COVID-19 model. These can potentially offer analytical advances through differential equations that capture a deterministic

modeling (*e.g.*, current Supplementary Eqs. (1.1) and (1.2)). On the other hand, the evolutionary dynamics are inherently stochastic and cannot be easily coupled with the ordinary differential equations tracking the pandemic spread.

One approach to overcome this difficulty is to consider a discrete,¹ and most crucially, limited, set of mutant variants. Often this reduces to a binary model,² in which the pathogen may transition between just two distinct mutant strains, or, in more complicated applications, having larger, albeit still limited, number of competing strains. This is sufficient if, for example, one wishes to model vaccine evasion,³ where, indeed, it makes sense to reduce the complexity of the evolutionary dynamics into a two-state alternation, *e.g.*, between a wild-type and an immunity evading strain (although, even here, one may consider a continuum of immunity evasion, as we now do in our revised discussion on COVID-19).

In our corrected submission, we now condensed our discussion of our main results, as we detail below:

Our novelty 1 – methodological. The above discrete treatment misses the gradual and continuous nature of pathogen evolution as a series of small random steps in an, effectively, continuum fitness space.⁴ To treat this *our formulation* allows us to translate the internal replication/mutation dynamics occurring within the host’s body into an effective random walk in the inter-host transmissibility fitness domain. This allows us to couple the epidemic differential equations with the stochastic evolutionary dynamics, as outlined in Eqs. (7) – (10).

The result is a quite generic formulation which accepts any desired disease cycle (SIS, SIR, COVID-19) and systematically extracts its interplay with any assigned fitness distribution (*e.g.*, zero or non-zero mean Gaussians). For the more fundamental disease models, SIS and SIR, our methodology enables us to obtain analytical results, providing a direct prediction on the resulting pandemic phase-diagram (see below). Our methodology, therefore, resides in a mathematical *sweet-spot*: detailed enough to capture the evolution/spreading dynamics, yet simple enough to allow analytical advances.

An analogous, albeit not identical, modeling approach has been recently considered in Ref. 4, which we now reference in the appropriate location in our revised paper. We emphasize, however, that while the mathematical tools are, indeed, similar in certain aspects, the main result we present is unrelated to that of Ref. 4. There, the goal is to understand the role of long-range network links. In contrast, our work is not necessarily focused on the role of the network structure, but rather seeks to expose the fundamental pandemic phase-diagram of evolving pathogens.

This brings us to our second outcome, which is achieved using, precisely the above

¹ E. Gubar, V. Taynitskiy & Q. Zhu. Optimal Control of Heterogeneous Mutating Viruses. *Games* **9**, 103 (2018).

² S.A. Rella, Y.A. Kulikova *et al.* Rates of SARS-CoV-2 transmission and vaccination impact the fate of vaccine-resistant strains. *Scientific Reports* **11**, 15729 (2021).

³ M. Fudolig & R. Howard. The local stability of a modified multi-strain SIR model for emerging viral strains. *PLoS ONE* **15**(12): e0243408 (2020).

⁴ S. Rüdiger, A. Plietzsch, F. Sagués, I.M. Sokolov & J. Kurths. Epidemics with mutating infectivity on small-world networks. *Scientific Reports* **10**, 5919 (2020).

methodology:

Our novelty 2 – quantitative predictions. The Referee, correctly notes that the fact that mutations can help a pathogen overcome herd-immunity is well-known,^{2,3} and – in fact – quite intuitive. However, such intuition remains *qualitative*, and cannot offer the precise conditions for such mutation-driven contagion to occur. This requires a *quantitative* analysis, that allows to observe the actual points of transition, and most importantly – an *analytical framework*, that can help reduce the system’s complexity into a manageable set of relevant parameters and critical points.

Our contribution offers precisely that. Starting from a detailed description of the intra-host evolutionary processes and the inter-host transmission dynamics, our analysis helps reduce the system’s microscopic details in to its two relevant parameters, which *de-facto*, determine the state of the pandemic. The first, R_0 , captures the classic epidemic reproduction rate – indeed, a well-known determinant of epidemic dynamics. The second, σ , quantifies the pathogen stability through its observed inter-host mutation rate. Our analytical results help us extract the competing timescales of the pandemic/evolutionary dynamics, providing *quantitative testable* predictions on the resulting pandemic phase-diagram:

- We identify the precise conditions for an initially sub-pandemic pathogen ($R_0 < 1$) to reach a critical mutation and turn pandemic (Eq. (15)).
- We uncover the discontinuous (*explosive*) nature of the resulting pandemic transition (**Fig. 2j**), which is not only a rarity in epidemic models, but also has crucial implications on forecasting and mitigation (newly added **Fig. 4**).
- Finally, we predict a second transition, back to the healthy state, occurring when mutations are too rapid, and the pathogen fails to sustain its evolutionary gains (Eq. (16)).

All of these analytical predictions arise from the complex interplay between the intra-host evolutionary dynamics, which determine the mutation rate (σ), and the inter-host transmission dynamics (R_0), which govern the observed natural selection timescales. Our derivations allow to reduce this complexity into a manageable analytically tractable modeling framework, that provides direct prediction on the critical transitions in the system’s effective two-dimensional phase-space.

Broader picture. To put this in perspective, consider the classic SIS or SIR models without mutations. It is, indeed, *intuitive* that the more contagious is the pathogen, the higher is its predicted prevalence. Still, such intuition, even if backed by direct observations, lacks *quantitative* insight. In contrast, the analytical treatment of these basic models predicts the system’s precise phase-diagram, exposing that: (i) The relevant control parameter is R_0 ; (ii) The critical point is $R_0 = 1$; and (iii) The nature of the transition is second order, *i.e.* continuous. We are certain the Referee would agree that such *quantitative* predictions are not extraneous, despite the fact that the *qualitative* behavior of the system could, in principle, be intuitively *guessed*.

In a similar fashion here, our work constructs the expanded pandemic phase-diagram,

including the added component of pathogen evolution. Its main contribution is in identifying the relevant control parameters and the precise location and nature of the observed critical transitions, *à la* (i)-(iii) above. Most crucially, just like the pandemic transition at $R_0 = 1$ offers insights that extends well-beyond the stylized SIS/SIR models, we hope that our timescale analysis of Eq. (17), and its associated phase-diagram (**Figs. 2,3**), can offer similarly broad insight for mutation-driven contagion.

Revisions

We have now fundamentally reshaped our paper, following the comments of all three Referees. As the Referee will see, our modeling framework, as well as our analysis has been dramatically improved, and better attuned to the existing knowledge on evolutionary dynamics (see our response to **Comment 2** below). We have also revised the paper's introduction and presentation to reflect the above discussion, highlighting our contribution *in context* and, of course, adding the relevant citations, that seem to have been overlooked in round one.

2. Comment

The first part of the paper focus on the SIS model however a realistic description of an epidemic spreading is given by SIR model or its variants (e.g. the model in Fig4a). The authors should discuss which of the results in the section "Phase-diagram of evolving pathogens" could be valid even for SIR model. This is not really trivial since the effect of mutations is relevant during the whole dynamical evolution where the differences between the two models are important. Indeed only close to the threshold the models should behave in a similar way; since the mean field equations displays the same linearized behavior for $S=1$.

Response

We fully agree with the Referee – SIS and SIR *can* behave potentially differently under pathogen evolution. Indeed, SIS reaches a long-term pandemic state, while SIR peaks and then decays to a zero-prevalence state. This offers a limited timeframe for mutations to build up in SIR as opposed to the extended pandemic state of SIS. We, therefore, agree that diverting the discussion to SIR is more relevant: both in terms of epidemiological realism, and in terms of the meaningfulness of our theoretical discussion. *Hence, following this comment, we have now redone our analytical and numerical analyses on the SIR spreading dynamics.* We continue to show results on SIS in the newly added **Supplementary Section 2**, as indeed, the SIS model offers several unique features that are absent in SIR, for example, the hysteresis phenomenon (see **Comment 3**).

We also wish to use this opportunity to detail other changes we have made to our modeling framework, following the comments of the other Referees, that we believe strengthen our analysis significantly.

Evolutionary dynamics. Revising our paper, we have realized that our previous treatment of the mutation process overlooked some of the specific details of real-world pathogen evolution. We, therefore, now begin with a much more detailed model of the

intra-host replication/mutation dynamics:

Upon exposure, the pathogen begins to replicate within its host i , mutating between strains μ and ν as dictated by the transition matrix $M_{\mu\nu}$. These strains compete within the host, governed by their i -dependent intra-host fitness $\varphi_\mu(i)$, in which fitter strains, that is better adapted to the host environment, proliferate more rapidly. After a series of ρ replications the pathogen population within the host reaches a host-specific distribution of strains. This captures the host's *multistrain* Z_i , a combination of mutant strains determined by the host-environment ($\varphi_\mu(i)$), the pathogen replication dynamics ($M_{\mu\nu}$) and the number of replications within an average infection cycle (ρ). Upon transmission to another host, a random sample from Z_i is transferred to the new host j , and the replication/mutation process is instigated once again.

In our analysis, we show how these detailed evolutionary dynamics reduce into an effective *observed inter-host fitness* ψ_μ , which performs a Gaussian random walk in the transmissibility fitness space (newly added **Fig. 1e**). This framework also allows us to consider the scenario where the intra and inter-host fitness gains are correlated, *i.e.* where $\varphi_\mu(i)$ and ψ_μ are interdependent, and as a result the mutation-driven random walk is biased towards higher (or lower) transmissibility (newly added **Supplementary Section 4**).

Epidemic dynamics. As a result of the incremental changes in ψ_μ , we observe a concurrent spread of different strains, all competing over the limited susceptible host population. More transmissible strains exhibit a spreading advantage, that allows them to proliferate more effectively and gain more prevalence, a contagion driven natural selection mechanism. While the evolutionary timescales are determined by the intra-host replication dynamics, the natural selection timescales are governed by the epidemic spreading process. These competing timescales lead to our observed *Goldilocks zone*: mutations must be rapid enough for the pathogen to achieve $R_0 \geq 1$ before it is eliminated, yet not *too* rapid, for the spread-driven selection to secure the proliferation of fitter strains (see **Box** in main text).

Together this modeling framework, we believe is: (i) More grounded in our current understanding of intra/inter-host evolutionary dynamics; (ii) Offers intuitive, and hence generalizable, insights on the factors driving the spread of evolving pathogens; (iii) Allows us to achieve analytical advances, leading to testable quantitative predictions, *e.g.*, Eqs. (15) and (16), capturing the critical transition points, **Figs. 2j** and **3d**, describing the nature of the observed transitions, and **Fig 4**, offering direct insights on mitigation timescales.

3. Comment

The hysteresis should be discussed in a more clear way. For any L_0 if you start at very small R_0 , in the lethargic phase, the systems should go into the absorbing state where no infected nodes are present. Therefore, the system remains in the state with no infection even when R_0 becomes larger than the transition value and for arbitrary large value of R_0 . Clearly this happens if the variation of R_0 is slow enough. While for fast variations even in the lethargic

phase the number of infected agents may not vanishes so that for larger R_0 the system may enter in a active phase. Therefore some discussion on the speed of the variation of R_0 should be introduced. Moreover, the presence of hysteresis should be one of the aspect that is very affected by the difference between SIS and SIR models.

Response

The Referee is *spot on*. The hysteresis is indeed only present in the SIS dynamics, due to the fact that the system reaches a continuous pandemic steady state. It, therefore, *remembers* its current R_0 through its present state. Indeed, if, *e.g.*, the initial $R_0 > 1$ the system reaches a state of non-zero prevalence η_∞ . Then when R_0 is reduced, the new dynamics ensues from this initial condition of high prevalence, *inheriting* the steady state of the previous condition. Under SIR, as the Referee correctly notes, the effect is lost, as now, the steady state is always $\eta_\infty = 0$, the system reaches herd-immunity, and hence, lacks the memory-effect that induces hysteresis. We also agree that the temporal dynamics of R_0 are important. In our original analysis we assumed extremely gradual changes in R_0 allowing the system to always reach steady state between R_0 updates.

While this discussion is, indeed, worthwhile, it is sidelined in our current version of the paper, in which, following the *Referee's* advice, we switched our original focus on SIS with our current SIR-based modeling. Wishing to maintain the paper's focus and conciseness, and to avoid digression, we removed the discussion on SIS, and its ensuing hysteresis to **Supplementary Section 2**, where we now explain the precise conditions for this phenomenon to appear. We refer to this discussion in the appropriate location in the main-text.

We also wish to emphasize that while the hysteresis phenomenon is SIS-specific, the discontinuity of the mutation-driven pandemic transition remains relevant also in SIR (as well as in more elaborate compartmental models). Hence, this novel and – we think, surprising, characteristic of the transition – that it is of explosive nature – persists also under our current storyline.

We wish to thank the Referee for helping us elucidate these intricacies.

4. Comment

I cant find the value of the initial conditions $\rho(0)$ or I_0 in Fig1. This is a very important parameter.

Response

With the changes in the modeling framework (SIR, Inter/intra-host evolutionary dynamics etc.), the figures are now completely redone. Therefore, this specific comment is no longer relevant. Of course, in the present submission, we made sure to clearly highlight all the relevant parameters, including $\eta(0)$ (ρ is no longer our notation for prevalence), in the figure itself, its caption or the text, as appropriate. Specifically, see caption of **Fig. 2**, where we report all relevant details in a concentrated fashion.

5. Comment

In the SIR model the importance of early adoption of containment measures is well known fact even in absence of mutations: Phys. Rev. E 102, 020301 (2020)

Response

We agree – there are many reasons to respond sooner than later. Indeed, even without pathogen evolution, a late response may lead to a prolonged pandemic phase, as explained in the mentioned reference (which we now cite). Our analysis provides two additional considerations, that we believe are important, both theoretically and practically:

Theoretically. We offer a different mechanism by which late response is risky. This is rooted in the fact that as the pathogen gains prevalence it also incurs an increased risk of reaching critical fitness. Indeed, our phase boundary, predicted in Eq. (15), depends on the initial prevalence J_0 , and hence, if we allow this number to grow, we may find that our mitigation may have arrived too late, when the pathogen has already entered the mutation-driven pandemic phase.

In that sense, time plays here a distinct role, unobserved in the classic SIR dynamics. Indeed, in the *mutation-less* modeling, the pathogen's healthy/pandemic phase is independent of time, fully determined by R_0 . A late response may lead to a different temporal trajectory, but it continues to tackle the *same pathogen*, within the same pandemic phase. Here, however, the phase boundary itself depends not just on R_0 , but also on J_0 , the prevalence at the time of our response. And since this prevalence grows exponentially at the early stages of the spread, we find that the phase boundary itself shifts with time. Therefore, a late response, might, in fact, catch the pathogen when it is already at a different phase – crossing into the mutation-driven phase. Now, even if we push R_0 to be below unity, the pathogen already exhibits an increased risk of breaking through.

Such mechanism of time-dependence, we believe, is novel and potentially interesting, and certainly distinct from the effect brought forth in previous works.

Practically. The above distinction has also practical implications. Indeed, in the standard SIR, a late response may cause a prolonged mitigation, however, it is, after all, guaranteed to successfully mitigate. Namely, as long as one pushes $R_0 < 1$, the disease will eventually change course and taper off. Under mutations, if the late response occurs when the pathogen is already in the mutation-driven phase, then it will not just be prolonged, but rather – it will utterly *fail*. This is because if J_0 is large enough, the pathogen has a high probability of reaching critical fitness, the dominating strain will then have $R_0 > 1$, and the spread will reemerge in the form of a new pandemic wave (see, *e.g.*, newly added **Fig. 4**).

Revisions

We have now added a detailed discussion on these theoretical and practical aspects of the risk in delayed response. We also specifically discuss these risks in relation to the known drawbacks of late mitigation. We track the exponential rise in J_0 , and denote the specific point in time t_c in which the pathogen crosses into the mutation-driven phase. We show that responding before t_c leads to successful mitigation, however, later than that –

mitigation is overridden by mutations. Such effect, driven by mutations, cannot be observed in the classic SIR framework, and yet it is naturally predicted within our formulation.

We wish to thank the Referee for helping us elucidate our precise contribution and for pushing us to improve our modeling, the resulting practical implications and their connection to the state of the art. We feel that the discussion, indeed, guided us towards focusing on our framework's most relevant and intriguing outcomes.

Reviewer #2

1. Comment

In the current study Zhang et al. present a simple, but elegant model of pathogen transmission and concurrent evolution. Through a combination of theory and simulations they find (admittedly unsurprising and known) that vaccination can lead to vaccine escape mutants and that rapid vaccination plus NPIs can limit escape mutants. They also find regimes where R_0 can be under 1 but with a high enough mutation rate there can still be sustained transmission (which is a really interesting result), ditto for the hysteresis results (these are very important results). This is a nice piece with some ... let's say interesting ... choices of language. There are very few evolution-transmission models and this one is particularly good. I have only a few comments/questions that should be easy for the authors to address.

Response

We are grateful for this concise and appreciative summary of our paper – indeed, capturing all the main aspects of our contribution. Following this and the other reports we have now introduced several significant improvements that we believe further strengthen our paper's narrative. Therefore, before we proceed to our point-by-point response, let us first summarize the main improvements introduced in the current paper.

Modeling. We agree with the Referee that one of the main contributions in the present work is our proposed *evolution-transmission* modeling framework. On the one hand, it is designed to capture the incremental and random nature of the evolutionary dynamics, and on the other hand, it accounts for the interplay of these dynamics with the spreading process, which is governed by the compartmental disease models. The challenge is that while the former is inherently stochastic, the latter is typically analyzed via deterministic differential equations – a problematic combination that often prohibits analytical advances. In that sense, we believe that our paper has detected a *sweet-spot*, sufficiently detailed to capture the richness of the observed spreading patterns, and yet, simple enough to allow for tractable analytical derivations.

Having said that, reading the other reports, we realized that our treatment of the evolutionary dynamics may have been too simplistic, in some instances, not on par with the *philosophy* of the existing evolutionary models. We, therefore, now revised this component of our modeling, to better align it with the existing domain knowledge. In our revised paper, we show how our analytical reductions arise naturally from this, more detailed, modeling framework.

Our current modeling of the evolutionary dynamics. We now begin with a detailed model of the intra-host replication/mutation dynamics. In this framework, upon exposure of host i , the pathogen begins to replicate within i 's body, mutating between strains μ and ν as dictated by the transition matrix $M_{\mu\nu}$. Within the host, these strains compete, each replicating at a rate governed by its *intra-host* fitness $\varphi_\mu(i)$, such that fitter strains, *i.e.* better adapted to the host environment, proliferate more rapidly. After a series of ρ

replications the pathogen population within i reaches a host-specific distribution of strains. This captures i 's *multistrain* Z_i , a combination of mutant strains, whose composition is determined by the host-environment ($\varphi_\mu(i)$), the pathogen replication dynamics ($M_{\mu\nu}$) and the number of replications within an average infection cycle (ρ). Upon transmission to another host j , a random sample from i 's multistrain is transferred to the new host, and the replication/mutation process is instigated again, this time within j .

In our analysis, we show how these detailed evolutionary dynamics reduce into an effective observed *inter-host* fitness ψ_μ , which performs a Gaussian random walk in the transmissibility fitness space. This naturally translates to the continuum random walk of Eq. (10), in which all the microscopic intra-host dynamics condense into the single control parameter σ , capturing the observed inter-host mutation rate; see our newly added illustration in **Fig. 1e**.

From this point on, all our analytical and numerical results, from the predicted phase boundaries (Eqs. (15) and (16)) to the type of transitions (**Fig. 2j,k**), remain unchanged (*almost* unchanged, see below). *The main point is that now, these same results are much more firmly grounded in the existing views on evolutionary dynamics.*

Our current modeling of the disease dynamics. Following the advice of the other Referees, we now focus on the SIR model instead of SIS. SIR is not only more relevant epidemiologically, but also provides a more meaningful discussion in the context of mutations. This is because, as opposed to SIS, in which the system reaches a pandemic steady-state, with a stationary non-zero prevalence, in SIR the spreading dynamics decays, reaching elimination, as herd-immunity is accumulated. Therefore, while SIS allows, in principle, an unlimited timeframe for mutation, under SIR, we have a restricted time-window for the evolution to take place, offering a more realistic and, we think, also more interesting test case.

Hence, in the main text we now analyze SIR dynamics, leaving the SIS discussion to the Supplement (newly added **Supplementary Section 2**).

Results. Our main results pertain to the structure of the pandemic phase-diagram. Specifically, our observation of the *mutation-driven* phase in which a sub-pandemic pathogen is able to break through and spread. We also uncover the *volatile* phase, in which pathogens lose this ability due to their overly rapid mutations. Finally, we analyze the dynamics of the observed transitions, exposing the explosive (discontinuous) nature of the mutation-driven criticality – a rather uncommon phenomenon in the epidemic landscape. We were excited to see that the Referee found these observations to be *interesting* and *important*.

We also agree with the Referee that our vaccine evasion scenario is not as novel or surprising. In retrospect it was, in fact, a digression from the main storyline of the paper, whose prime focus was on the above phase-diagram. We, therefore, now removed this part of the discussion, to make the paper more concise, more focused, and, most importantly, tailored around its actual novelty. We wish to thank the Referee for prompting us to take this step.

Instead of the vaccine evasion scenario, we now discuss reinfection, in which mutations allow a virus to penetrate the wild-type's naturally induced immunity. This allows us to characterize the conditions for the reemergence of a second pandemic wave. Focusing this discussion on the COVID-19 disease cycle we demonstrate two potential strengths of our proposed framework:

- (i) **Generalizability.** While our analytical observations are achieved around SIR and SIS, their insights range beyond these simplified models, and can be readily generalized to *any* compartmental model. For that matter, our application to COVID-19 is not only timely, but also a showcase of our framework's potential broader insights.
- (ii) **Realism.** While immunity evasion (just like vaccine evasion) is, in and of itself, not novel, our framework allows us to model it in, what we believe, is a highly meaningful fashion. Indeed, previous analyses employed a binary approach, in which a single mutation transforms the wild-type into a breakthrough strain that can reinfect the recovered population. In reality, however, the evolutionary dynamics is continuous, not binary. As mutations build up, the evolved strains become more and more genetically distinct from the wild-type, and hence acquire a gradually increasing probability ϕ_μ to evade its immunity. Such step-by-step evolutionary dynamics are, generally, difficult to model, yet they fit quite naturally into our dynamic formulation, which is specifically designed to track the continuous accumulation of fitness.

Therefore, our current COVID-19 demonstration is not simply *another* application, but rather a specific exposition, tailored to support our main scientific narrative.

Hysteresis. The hysteresis phenomenon, which the Referee deems *very important*, is still part of our results, however, we must update that it is now longer in the spotlight. The reason is that this observation is specific to the SIS model, and does not translate to SIR, which is now the prime model upon which we demonstrate our analysis. Still, we agree that it is an interesting finding, with important implications, and therefore we discuss it explicitly in the main text, and refer the reader to the detailed analysis that now appears in **Supplementary Section 2**. To be honest, we were hesitant to move such an interesting result to the supplement, however, we felt that including the full SIS analysis in the paper *in addition* to SIR, would overburden the presentation and risk the paper losing focus.

The SIR model, we emphasize, continues to exhibit the discontinuous (*explosive*) pandemic transition, which we discuss in detail. Such first-order like transition, now shown in **Fig. 2j**, is not a common sight in epidemic models, and has crucial implications for forecasting and mitigation, as we discuss in the current version of the manuscript.

Terminology. Rereading our original submission, we understand what the Referee means by our loose, at times colloquial, language (echoed also in **Comment 3** below). We have now rewritten our paper quite extensively and used this opportunity to clean up our terminology: (i) Better aligning it with the existing terms; (ii) Reserving a more professional form. We think that now, following this rewrite, the paper reads significantly better and wish to thank the Referee for pushing us to improve our presentation style.

2. Comment

First, how did this collaboration come about? The institution list reads like a United Nations of science: China, Italy, Russia, Spain, and Israel? Kudos on the collaboration!

Response

Global challenges call for global efforts.

This was, indeed, a fun, heavily zoom-supported, multi-lingual, and cross-cultural collaboration. Hopefully, if no new variant comes about to ruin our plans, once our paper is accepted (🙏), our multi-national team will have the opportunity to raise a toast face to face in some neutral destination (travel bans notwithstanding).

3. Comment

The language is colloquial in places for a scientific paper. Some examples: “drug-persistence” should be “drug-resistance”, I believe. P3: referring to $R_0 < 1$ as the “healthy” state seems odd to me, not sure of the replacement, though, maybe increasing and decreasing states? P3: “unmutated” is called “wild-type”. I’m not a fan of the use of “lethargic” for the low-mutation, $R_0 < 1$ state (best not to anthropomorphize the virus). Maybe “shrinking”? While I like “Goldilocks zone” it probably isn’t appropriate in this setting.

Response

We agree. We have now made an effort to remove these inaccuracies. Specifically, we now use *Infection-free* to replace the *Healthy* state, and avoid the term *Lethargic* altogether. We have also converged to more formal evolutionary dynamics terminology, e.g., wild-type, multistrain etc.

We do, however, stand our grounds regarding the *Goldilocks zone*. Indeed, it is informal, however, the metaphorical meaning, we believe, is quite clear. Moreover, its position in the paper, i.e. the grey Box, is meant as an informal, intuitive, summary of our findings, hence we think it is appropriate. We hope the Referee will concur.

4. Comment

I wouldn’t say that R_0 of 1.5 is “highly contagious”. Measles, with $R_0 \sim 18$, is “highly contagious”.

Response

We agree.

5. Comment

It’s a strong assumption to assume that all symptomatic individuals isolate and cease to transmit that needs better justification. The 30% asymptomatic needs a citation, I think it’s

more like 50%.

Response

The full compliance with the isolation policy is obviously over-optimistic, and was implemented for simplicity only. In reality, we completely agree, that people cut corners. This is mainly relevant for the mildly symptomatic (I_M), since the severe (I_S) and critically (I_C) ill are hospitalized or ventilated and cannot break isolation. We, therefore, now repeated, in **Supplementary Section 5.3**, our COVID-19 simulations under a more realistic assumption of 70% compliance, *i.e.* 30% of the mildly symptomatic violate the isolation. Of course, these distinctions are secondary to the main message, and apart from minor corrections, the results, in essence, remain unchanged (see **Supplementary Fig. 7**).

Regarding the fraction of asymptomatic individuals (I_{AS}), the literature is split on the numbers, which, reviewing the (many) papers on the matter, were found to range from⁵ $\sim 20\%$ to⁶ $\sim 40\%$. Our suggested cycle took the average estimate of 30%. These details, have, of course, no significant bearing on our reported results, which can be easily adapted to different rates/transition probabilities of any other pathogen/variant.

6. Comment

Supplement Figure 2: I really like how the authors estimate α from the data, but two things, one, I assume the authors mean cumulative cases not prevalence; and two, why just take the average? It would be nice to also do the low and high estimates.

Response

The Referee is correct, we measure the cumulative cases. Since we focus on the early stages of the spread, the majority of patients have not yet recovered, and hence this is quite close to the prevalence. Regardless, we have now corrected this in our description, and wish to thank the Referee for pointing this out.

We also agree that it makes sense to cover the entire range of β estimates. The crucial point is that in our analysis we do just that, and in fact much more. Indeed, the phase-diagram of **Fig. 5** (and **Supplementary Fig. 7**) runs over a range of R_0 values from 0 to 5, whereas SARS-CoV-2 is estimated, in our data to be within $R_0 = 2.6 \pm 0.3$. Hence, in effect, we cover not just the upper/lower bounds, but also a significant margin beyond that. In our current analysis of the COVID data we now explain this point and also discuss the margins in our empirically extracted SARS-CoV-2 infection rate (**Supplementary Section 5.3**).

We once again wish to thank the Referee for this constructive and insightful report.

⁵ Ferretti, L. *et al.* Quantifying SARS-CoV-2 transmission suggests epidemic control with digital contact tracing. *Science* **368**, 6936 (2020).

⁶ Tao, Y. *et al.* High incidence of asymptomatic SARS-CoV-2 infection, Chongqing, China. *medRxiv* 20037259 (2020).

Reviewer #3

1. Comment

Zhang et al. firstly present a simple SIS model for pathogen spreading on a static contact network (Erdős-Rényi), which is extended for a stochastic term (“evolution”) that changes the models’ parameters randomly. They analyze the model’s behavior with regards to its infinite time behavior (pandemic will establish, vanish). They then extend the model by introducing several ‘infected compartments’ to apply the model to COVID and make several claims regarding mitigation and vaccination strategies. The latter model is however insufficiently described in technical details.

Response

We wish to thank the Referee for this summary of our work, as well as for his/her comments below. While critical, we found this report to be extremely helpful in reshaping our presentation, placing it within the context and terminology of the state of the art, and, most importantly, elucidating our precise novelty within this broader context.

As the Referee will see below, this report prompted us to move out of our network science *comfort zone*, and reformulate our findings to better adhere to the language and *philosophy* of evolutionary dynamics. It also helped us better define *our* specific contribution within this context. We now understand that our novelty is *not* in introducing a new modeling framework, but rather in deriving (analytically) the pandemic phase-diagram that results from pathogen evolution, *within our current understanding of evolutionary/epidemic dynamics*. Namely, we do not seek to add a new model to the pool, but rather to expose the outcomes of our existing modeling frameworks, when pathogen evolution is coupled with an ongoing inter-host spread.

This leads us to focus our revised paper on our two main contributions:

Methodological. The two processes we consider can each be analyzed on their own: Epidemic spreading on networks via compartmental models, which are often tracked using deterministic differential equations;⁷ evolutionary processes via replication-mutation dynamics, which, by their nature, require stochastic, often discrete, mathematical tools. The challenge is that these two modeling frameworks, differential equations and stochastic analyses, cannot be easily coupled, and hence the *interplay* between the spreading and the evolutionary dynamics remains uncharted. A common bypass is to discuss a limited space of mutations, comprising just few competing strains. Yet this overlooks the gradual and continuous nature of real-world evolutionary processes.

To address this we show that we can reduce the complexity of the two dynamic processes into two *relevant parameters*, the classic R_0 , characterizing the epidemic spread, and the observed inter-host mutation rate σ , that arises from the intra-host evolutionary dynamics. This allows us to couple the disease model with the evolutionary processes via Eqs. (7) – (10), offering an analytical framework that can help us systematically map the

⁷ R. Pastor-Satorras, C. Castellano, P. Van Mieghem & A. Vespignani. Epidemic processes in complex networks. *Rev. Mod. Phys.* **87**, 925 (2015).

state of the system in the R_0, σ phase-space. This leads us to our main contribution:

Predicting the pandemic phase-diagram. We use our formulation to identify the precise critical points of transitions under a concurrently spreading and evolving pathogen. Specifically:

1. We uncover the conditions within the R_0, σ space for an initially sub-pandemic pathogen ($R_0 < 1$) to reach a critical mutation and turn pandemic (Eq. (15)).
2. We further show that the transition in 1 is discontinuous (*explosive*, **Fig. 2j**), which is not only a rarity in epidemic models, but also has crucial implications on prediction/mitigation (**Fig. 4**).
3. Under SIS, this transition is further characterized by hysteresis, which is, again, an uncommon, yet highly pertinent phenomenon in the context of epidemic transitions (**Supplementary Section 2**).
4. Finally, we predict a second transition, back to the infection-free state, occurring when mutations are too rapid, and the pathogen fails to sustain its evolutionary gains (Eq. (16)). See our response to **Comment 16**, where we discuss the distinction between this and the classic *error catastrophe*.
5. Along the way, we extract the relevant timescales that govern the interplay between the evolutionary and the spreading dynamics.

Phenomena 1-4, we explain below, are absent in the current canonical models of epidemic spreading, and yet are highly relevant for prediction/mitigation. Item 5 allows to reduce the complexity of the spreading dynamics into a set of manageable parameters, allowing robust predictions, despite the detailed and stochastic nature of the spreading/evolutionary processes.

We wish to emphasize that our analysis and its mathematical results focus on our theoretical understanding of epidemic dynamics. It is, therefore, natural to rely on simplified descriptions such as the SIS or SIR compartmental models, which allow analytical advances. Still, fundamental aspects, such as the nature of the observed phase-transitions (items 1-3) or the macroscopic control parameters that drive them (R_0, σ) often extend, or at least provide crucial insights, significantly beyond their underlying mathematical simplifications and approximations.

For example, in epidemic spreading, the reproduction rate R_0 , the main control parameter for the pandemic transition, was originally derived in the context of the SIS model – perhaps the simplest compartmental model for epidemics. Yet, it remains relevant even for more complicated epidemic dynamics. Similarly, the phase-diagram itself, and specifically the nature of the pandemic transition (second-order phase transition), are also quite general, extending significantly beyond the analytical bounds of the SIS model, and, in fact, currently informing us on the dynamics of the much more complex COVID-19.

This outlines precisely the track we now take in our revised submission: we derive our theoretical results based on our tractable mathematical framework, but then show numerically, that their insight holds also to more detailed and elaborate, and hence,

empirically relevant, models.

On that note, we now made sure not to overstate the reach of our findings, keeping the language and the focus of our paper on its theoretical novelty, in the realm of dynamical systems, phase-transitions and epidemic spreading, but avoiding claims of novelty pertaining to the models of evolutionary dynamics themselves. Instead, we made sure to place our evolutionary model within its proper context, on track with the state of the art.

All our revisions are detailed below in our point-by-point response. We hope that the Referee will agree that our revised version is now better grounded in the existing knowledge base of evolutionary dynamics, and that our findings are appropriately described within this context – much thanks to this detailed and in-depth report.

Revisions

We have now fundamentally reshaped our paper, following the comments of all three Referees. As the Referee will see, our modeling framework, as well as our analysis have been dramatically improved, and better attuned to the existing knowledge on evolutionary dynamics. We have also revised our introduction and general presentation to reflect the above discussion on the paper's main focus, highlighting our contribution *in context* and, of course, adding the relevant citations, that were overlooked in round one.

2. Comment

The authors are quite obviously not very well acquainted with the field of evolutionary dynamics, which in itself is ok, but there is a complete disregard of what has already been achieved in the field in terms of extremely profound mathematical theory. This weakens the paper considerably, firstly, because some assumptions made about how pathogens evolve are fundamentally wrong and secondly the field of evolutionary dynamics has already progressed way further than where this manuscript stands (examples will follow). Overall the manuscript, in my view, lacks a remarkable innovation, is overall not very well researched (literature; what have others done?) and the conclusions are pretty trivial (one would not need a model for that, but could always come up with one that yields the presented conclusions). The used terminology is problematic. Unfortunately, all in all I have to recommend rejection.

Response

This comment has led us to fundamentally revisit our modeling assumptions and their presentation. Indeed, as the Referee correctly identifies, we are newcomers to the field of evolutionary dynamics, bringing our expertise from network science, nonlinear dynamics and epidemic spreading. Hence, we have now thoroughly reviewed the relevant literature, to help us reformulate our modeling framework and bridge the gap between our mathematical findings and the current understanding of pathogen evolution. Let us below explain in detail our current formulation.

Our modeling framework and its relation to evolutionary dynamics

Intra/inter-host evolution. We consider intra-host evolutionary dynamics. Upon

exposure of host i , the pathogen begins to replicate within i 's body, mutating between strains μ and ν as dictated by the transition matrix $M_{\mu\nu}$. These strains compete within the host, as governed by their i -dependent intra-host fitness $\varphi_\mu(i)$. Fitter strains, better adapted to the host environment, proliferate more rapidly. Therefore, after a series of ρ replications the pathogen population within the host reaches a host-specific distribution of strains Z_i , which is biased in favor of strains with large $\varphi_\mu(i)$. This captures host i 's *multistrain*, a combination of mutant strains that is determined by the host-environment ($\varphi_\mu(i)$), the pathogen replication dynamics ($M_{\mu\nu}$) and the number of replications within a typical infection cycle (ρ). Upon transmission to another host j , a random sample from Z_i is transferred to the new host, and the replication/mutation process is instigated again, this time within j 's intra-host environment.

In our analysis, we show how these detailed evolutionary dynamics reduce into an observed inter-host fitness ψ_μ , which performs an effective random walk in the transmissibility fitness space (newly added **Fig. 1e**). This arises quite naturally from the fact that the intra-host fitness landscape is, by and large, independent of the inter-host fitness. Therefore, while Z_i is biased towards higher intra-host fitness ($\varphi_\mu(i)$), it is, in effect, neutral towards inter-host fitness (ψ_μ), leading to incremental, typically random, shifts $\Delta\psi_\mu$, upon each instance of infection. The distribution of $\Delta\psi_\mu$, and specifically whether it is a zero-mean Gaussian or not, is discussed in our response to **Comment 11**.

Of course, this framework naturally generalizes to the case where the intra and inter-host fitness gains *are* correlated, *i.e.* where $\varphi_\mu(i)$ and ψ_μ are interdependent, and as a result the observed inter-host evolutionary jumps $\Delta\psi_\mu$ are biased towards higher (lower) transmissibility. We now investigate this scenario in **Supplementary Section 4**.

Most crucially, now that we have adopted this detailed description of the evolutionary process, we can systematically investigate how the microscopic intra-host evolutionary dynamics ($\varphi_\mu(i), M_{\mu\nu}, \rho$) translate to our macroscopically relevant parameter σ , the observed inter-host mutation rate. We discuss this in our revised formulation, and in the newly added **Supplementary Section 5.2**. We find that of all parameters, the factor that plays the main role in determining σ is $\varphi_\mu(i)$, namely the broadness of the intra-host fitness landscape. This is quite expected, since φ -heterogeneity leads to significant evolutionary shifts during the intra-host replication process. These, in turn, translate to potentially measurable changes $\Delta\psi$ between the incoming strain infecting i and the outgoing strain transmitted to j .

Epidemic dynamics. As a result of the incremental changes in ψ_μ , we observe a concurrent spread of different strains, all competing over the limited susceptible host population. More transmissible strains (larger ψ_μ) exhibit a spreading advantage, that allows them to proliferate more effectively and gain more prevalence. This captures the selection process that occurs in the *inter-host* fitness landscape. Hence, while the evolutionary timescales are determined by the intra-host replication/mutation process, the timescales of the natural selection mechanism are governed by the inter-host spreading dynamics. This leads to our observed *Goldilocks* zone: mutations must be rapid enough for the pathogen to achieve $R_0 \geq 1$ before it is eliminated, yet not *too* rapid, to allow sufficient time for the selection to secure the proliferation of fitter strains (see **Box**

in main text).

Taken together, apart from specific updates, we now arrive at similar results as those reported in our original submission. However, we believe that now they are much more firmly connected to our current understanding of evolutionary processes. We, therefore, wish to thank the Referee for prompting us to include these improvements.

Our novelty

In the comment the Referee considers our conclusions *trivial* – a harsh, yet subjective, critique, that we wish to contend. We believe this critique is mainly rooted in the previous presentation of our work, which emphasized our *evolutionary modeling*, rather than our *mathematical analysis* and its conclusions on the emergent *pandemic phase-diagram*. As we explain above, we agree that the model we consider for the pathogen evolution is – in and of itself – not our main contribution, and, in fact, maps directly to the existing mechanisms broadly considered in the literature. This is, therefore, not *the* focus of our work.

What we *do* focus on are the *outcomes* of these models – specifically, what patterns of epidemic spread emerge when evolutionary mechanisms are incorporated into the SIS/SIR spreading process. This combination of intra-host evolutionary dynamics with inter-host network transmission, is quite elusive in two challenging ways:

1. **Analytical challenge.** The two processes, evolution and spread, are closely intertwined, yet mathematically incompatible. The evolutionary process requires stochastic modeling, otherwise mutations simply average out to zero. On the other hand, to predict and understand the spreading dynamics one must employ a mean field approach, which relies on deterministic continuous differential equations. As a result, we can readily *simulate* the two coupled processes, using stochastic numerical tools *a la* Monte-Carlo simulations. However, we face crucial limitations when seeking *analytical* insight, *e.g.*, predicting the precise conditions for the mutation-driven phase.

A common simplification to allow mathematical tractability is to assume a limited discrete set of potential strains^{8,9} (typically two competing variants). This, however, overlooks the potentially continuous fitness space, in which significant fitness gains are often a result of a gradual accumulation of many incremental mutations.¹⁰

2. **Phase space.** Once we establish a tractable mathematical formulation the challenge is to solve it, and expose the different states and transitions of the system. As is often the case with analytical solutions, they can only be obtained relying on approximate descriptions, which must then be validated against detailed (and fully stochastic) numerical simulations.

⁸ E. Gubar, V. Taynitskiy & Q. Zhu. Optimal Control of Heterogeneous Mutating Viruses. *Games* **9**, 103 (2018).

⁹ S.A. Rella, Y.A. Kulikova *et al.* Rates of SARS-CoV-2 transmission and vaccination impact the fate of vaccine-resistant strains. *Scientific Reports* **11**, 15729 (2021).

¹⁰ S. Rüdiger, A. Plietzsch, F. Sagués, I.M. Sokolov & J. Kurths. Epidemics with mutating infectivity on small-world networks. *Scientific Reports* **10**, 5919 (2020).

Our paper offers crucial novelty on both fronts. To address 1, we show how to reduce the microscopic complexity into a low-dimensional macroscopic description, driven by two relevant parameters: R_0 , characterizing the spreading dynamics, and σ , quantifying the observed inter-host mutation rate. With this simplification we can address item 2, and obtain the complete pandemic phase-diagram.

Outlook. Before we unpack the specific findings that we obtain, let us first describe the broader picture of our methodological contribution. Beyond SIS, or SIR, our proposed framework can help systematically couple any compartmental disease model with any form of fitness distribution. For example, in our current submission we apply it to SIS, SIR and COVID-19, capturing three relevant, yet highly distinctive, epidemiological models. In terms of the fitness landscape, we consider both zero and non-zero mean Gaussian fitness distributions (Eq. (10), **Supplementary Section 4**), and also a sigmoidal fitness function, which is relevant in the context of our COVID-19 application.

What we observe

Sub-pandemic spread. In its classic formulation, epidemic spreading, whether modeled using the simple SIS/SIR or via more elaborate descriptions, is driven by a single parameter R_0 . For $R_0 > 1$ we observe an active pandemic, while for $R_0 \leq 1$, the system resorts to the infection-free state. The presence of mutations introduces a more complex state space, no longer determined by the pathogen's R_0 , but rather by R_0 's balance against the pathogen's mutation rate σ . In this two-dimensional phase-space we identify relevant conditions under which a pathogen can break through despite having initially $R_0 < 1$. Unlike the classic scenarios of an *escape variant*, in which the mutant strain evolves to evade a treatment or a vaccine, here the wild-type is non-pandemic from the *get-go*. Still, if mutating rapidly enough, we show, it can potentially change course, and transition, from exponential decay to exponential proliferation.

Nature of the pandemic transition. In addition to the shift in the *location* of the transition (from $R_0 = 1$ to $R_0 < 1$, as per Eq. (15)), the *nature* of the transition also changes in the face of mutations. Instead of a second-order phase transition, as observed in almost all standard epidemic spreading models, we show that mutations lead to a first-order transition. In simple terms, the transition from the infection-free to the pandemic states is not just *sudden*, occurring precisely at the critical R_0 (**Fig. R1a**), but also *abrupt*, jumping discontinuously from zero to high prevalence (**Fig. R1b**).

This discontinuity is not just a theoretical novelty, but also has crucial practical implications. For example, consider mitigation via social distancing. It is based on adapting our social behavior in order to push R_0 below the pandemic threshold. In the current pandemic, many societies converged to an R_0 which waggles around criticality,^{11,12} managing to sustain a social routine that, most of the time, is just below the pandemic threshold R_c , and at times, fluctuates above it. Under second-order transition, such minor fluctuations in R_0 have no major effect, thanks to the continuous, and hence gradual, nature of the pandemic transition. The abrupt first-order transition, on the other hand, indicates that around the critical point, the system is by far more sensitive: small changes to R_0 may result in a dramatic *jump* in the resulting long term prevalence.

Hysteresis. The final outcome of our analysis, observed specifically under SIS dynamics, is that the critical point depends on the *direction* of the transition (**Fig. R1c**). Approached from the infection-free state (left to right) the transition takes place at R_{High} , yet in the opposite direction it only occurs at R_{Low} . This, once again has important practical implications, distinguishing between preemption and mitigation. Indeed, preemptive steps aim to prevent the disease before it gains prevalence, therefore it suffices to push R_0 below R_{High} . Mitigation, however, attacks the disease when it has already gained ground, hence requiring stricter measures, *i.e.* $R_0 < R_{\text{Low}}$.

Figure R1. Different types of pandemic transitions. (a) The classic pandemic transition is critical but continuous. (b) Under SIR with mutations, we observe a discontinuous (explosive) transition, in which as cross criticality the prevalence jumps abruptly to $\eta \sim 1$. (c) In SIS, we also predict a hysteresis phenomenon: increasing R_0 follows the bottom path, whereas decreasing R_0 from an initially pandemic state, for example, when attempting to eliminate an already widespread pathogen, we must follow the top path.

We wish to emphasize that such discontinuous transitions or hysteresis phenomena, are unobserved in the classic models of epidemic spreading. They represent a fundamental shift from the current picture of a continuous, single point second-order transition. *Hence, while such findings are, perhaps, not at the heart of evolutionary dynamics, they are bound to interest researchers of epidemic dynamics.*

Quantitative predictions. As is often the case in analytical predictions, the conclusions, in hindsight may seem self-explanatory. Our case is no different – one can, indeed, reason out an intuitive explanation for all of the above findings, as in fact, we do in the current version of the paper (see, *e.g.*, grey **Box** in main text). This is likely what led the Referee to state that *one would not need a model* for our conclusions. Such intuitive insight, however, is not standalone, but rather a retrospection that follows from our analytical findings. Truly, we do not think that one can simply *guess*, based on intuition alone, that evolutionary dynamics lead to such fundamental changes in the pandemic phase-diagram,

¹¹ G. Lobinska, A. Pauzner, A. Traulsen, Y. Pilpel & M.A. Nowak. Evolution of resistance to COVID-19 vaccination with dynamic social distancing. *Nature Human Behaviour* **6**, 193 (2022).

¹² X. Zhang, G. Lobinska, M. Feldman, E. Dekel, M.A. Nowak, Y. Pilpel, Y. Pauzner, B. Barzel & A. Pauzner. A spatial vaccination strategy to reduce the risk of vaccine-resistant variants. [10.21203/rs.3.rs-969637/v1](https://doi.org/10.21203/rs.3.rs-969637/v1)

i.e. first-order, hysteresis etc.

Most importantly, intuitive insight is restricted to *qualitative* understanding, but prohibits *quantitative* predictions. In contrast, our modeling framework, its mathematical formulation and the ensuing analysis that follows – indeed, the main focus of our paper – offer specific *quantitative* observations • First, they help identify the relevant control parameters that drive the system’s transitions (R_0, σ). This helps capture the specific quantities of the evolutionary and spreading dynamics that determine the eventual pandemic state • Then, beyond that, our analysis predicts the precise phase boundaries, as appear, *e.g.*, in Eqs. (15) and (16). Hence, we do not simply *claim* that if evolution is fast enough the pathogen can reach critical fitness. Rather, we *calculate* precisely how fast is *fast enough* – offering *quantitative* testable predictions, which we indeed test in **Figs. 2-4**.

These *quantitative* predictions, we are certain, the Referee will agree go beyond *trivial conclusions* that can be achieved even *without a model*. In fact, they require not just a model, but also a concise mathematical formulation of that model (Eqs. (5) – (11)), and an analytical exposition to quantitatively solve it (Eqs. (12) – (16)). *These latter ingredients, i.e. the methodological framework and its resulting quantitative predictions, are precisely the crux of our current contribution.*

Broader picture. To put this in perspective, consider the classic SIS or SIR models without mutations. It is, indeed, *intuitive* that the more contagious is the pathogen, the higher is its predicted prevalence. Still, such intuition, even if backed by direct observations, lacks *quantitative* insight. In contrast, the analytical treatment of these basic models predicts the system’s precise phase-diagram, exposing that: (i) The relevant control parameter is R_0 ; (ii) The critical point is $R_0 = 1$; and (iii) The nature of the transition is second order, *i.e.* continuous. We are certain the Referee would agree that such *quantitative* predictions are not extraneous, despite the fact that the *qualitative* behavior of the system could, in principle, be intuitively *guessed*.

In a similar fashion here, our work constructs the expanded pandemic phase-diagram, including the added component of pathogen evolution. Its main contribution is in identifying the relevant control parameters and the precise location and nature of the observed critical transitions, *a là* (i)-(iii) above. Most crucially, just like the pandemic transition at $R_0 = 1$ offers insights that extends well-beyond the stylized SIS/SIR models, we hope that our timescale analysis of Eq. (17), and its associated phase-diagram (**Figs. 2,3**), can offer similarly broad insight for mutation-driven contagion.

Terminology

In his/her last remark the Referee raises the *problematic terminology* of our original submission. After consulting the other reports, we now understand that we were perhaps too *creative* in our choice of language. The revised paper now adheres more strictly to the formal terminology used in the field.

Revisions

We have now rewritten our paper quite extensively to express all of the above points: we connect our modeling of the evolutionary dynamics to the common view of intra-host

selection, and added references to exiting literature in all appropriate locations. We then proceed to derive our analytical phase-diagrams, making sure to highlight our main findings regarding the *qualitative* nature of the resulting transitions, and our specific *quantitative* predictions on the critical phase boundaries. Finally, our wording is now revised to reflect, as much as we could, the common terminology of the field.

3. Comment

The paper lacks references, as well as critical discussion with regards to works in the field of evolutionary dynamics. I suggest to consult the works of Martin Novak, Manfred Eigen and Peter Schuster as a STARTING POINT and put less focus on works within the network inspired physics community. Along these lines, the authors may want to differentiate between intra-host evolution and an observed 'evolution' at the population level (inter-patient). For viruses, to correctly model their evolution at the population level, this is an extremely important issue.

Response

We wish to thank the Referee for these references, which are indeed relevant to our current exposition. As explained in detail in our response to **Comment 2**, our work is focused on the interplay between the *intra-host* evolution^{13,14} and the network spreading dynamics, which together lead to the observed *inter-host* selection process. The work, and its references, must therefore balance both the evolutionary dynamics literature and *the network inspired physics community*. We agree that in its original form our paper may have leaned more towards the latter, and hope that the Referee will agree that it is now, after carefully reviewing the suggested references, more balanced.

Revisions

We now understand that our original narrative was detached from the literature and knowledge of evolutionary dynamics. Hence, we have now thoroughly rewritten our introduction to explain the relevance of our contribution along the lines of the above response. We also introduced our modeling framework in connection with the *intra-host* evolutionary dynamics, as suggested by the Referee, showing how it follows directly from our current understanding of pathogen evolution.

4. Comment

In the first paragraph, the authors mix up evolution of species (vast time-scales) and evolution of viruses. Also, to be exact: It is not the prevalence that causes mutations, but the virus replication within the host(s) (more replication -> more mutations may happen).

Response

¹³ Y. Iwasa, F. Michor & M.A. Nowak. Evolutionary dynamics of escape from biomedical intervention. *Proceedings of the Royal Society of London. Series B: Biological Sciences*, **270**, 2573-2578 (2003).

¹⁴ R.R. Regoes, M.A. Nowak & S. Bonhoeffer. Evolution of virulence in a heterogeneous host population. *Evolution*, **54**, 64-71 (2000).

We agree. This was careless wording on our end. We now corrected both instances. We note, however, that prevalence *does* play a role in the potential appearance of advantageous mutations. Consider the probability p for a certain mutation to appear within the host. This probability, as the Referee notes, is, indeed, determined by the intra-host replication. However, at the population level, the chances of realizing this probability p increases if more hosts are infected, as each may independently come to host this mutation.

Having said that, we agree that our original wording overlooked these nuances, and hence we have now rephrased our presentation accordingly.

5. Comment

Refs. 1-6 in the Introduction (page 1) do not support the claims made by the authors when they cite them. Same with Refs 7-13.

Response

As we detail above, our current introduction has been fundamentally revised, guided by the comments in this report. As a result, the narrative, motivations and the model description have all undergone a thorough rewrite, including the references to relevant papers. We believe that the Referee will find it now much better connected with the existing literature, and, most importantly, better focused and placed in the context of our current knowledge and understanding of evolutionary/spreading dynamics.

6. Comment

Page 2: I find many incorrect statements and presumptions here: Ref 28-32: Again, it may be of advantage to move out of the 'spreading phenomena on networks' bubble.

Reply:

Once again, in our current presentation we believe this critique is already addressed.

7. Comment

3rd paragraph: the understanding of "responsive evolution" is entirely wrong. For example: A pathogen does not become more transmissible because social distancing is enforced. It always tends to become more transmissible (or persistent), if it can. Also, the authors may want to consult some relevant works regarding evolutionary escape from biomedical intervention (e.g. PMID: 14728779 and citing references)

Response

This, again, represents loose writing on our end. Of course, pathogens do not *respond*. Yet, our actions, *e.g.*, social distancing, may potentially change the fitness landscape and impact the resulting selection pressure. In our rewrite we avoid such inaccuracies. We also wish to thank the Referee for the references to relevant publications. We have now

consulted these papers and referred to them where appropriate.

8. Comment

4th paragraph: This paragraph lacks detail. Again: it would be helpful to differentiate between intra-host evolution and observable evolution at the level of the population (between individuals that are infected). At the intra-host level, I would consider evolution as a random walk in fitness space. At the inter-host level things are a bit more complicated, but inherently depend on within-host viral dynamics and e.g. things like quasispecies dynamics and “transmission bottlenecks”.

Response

We wish to thank the Referee for this comment, which, once again, touches on the microscopic mechanisms of pathogen evolution, both intra and inter-host – prompting us to better formulate our modeling framework.

Intra-host. Within each host i , following ρ replication cycles, we observe a steady-state distribution $Z_\mu(i)$ of pathogen genotypes $\mu = 1, 2, \dots$. This captures i 's intra-host quasispecies, which emerges from i 's internal replication/mutation dynamics.

Transmission bottlenecks. We consider horizontal inter-host transmission, *i.e.* infection occurs between any two hosts, infected i and susceptible j , that share a network link. Upon infection, a small sample of pathogens extracted from $Z_\mu(i)$ is transferred to j , then begins to undergo replication within j , once again forming a distinct quasispecies $Z_\mu(j)$.

Selection. Regardless of the competition *within* each host, at the level of the population, the different strains compete for coverage, namely how well they undergo the horizontal transfer. This competition, we emphasize, is not between quasispecies, but rather between the strains themselves, due to the bottlenecks above, in which only one or few strains are transmitted during the $i \rightarrow j$ transfer. This competition is driven by the reproduction rate $R_\mu = \psi_\mu R_0$ of each variant μ , which determines its inter-host spreading efficiency.

Population level. The outcome at the population level is a set of concurrently spreading variants with potentially distributed R_μ values, that compete over the susceptible population. These variants are therefore subject to selection pressure for higher transmissibility.

Generalization. An important aspect of our formulation is that it can be readily generalized to treat other forms of evolutionary dynamics. For example, we can consider a biased random walk, in which fitness either tends to increase or decrease between infection instances. This can capture the scenario where the intra-host replication feeds also to the inter-host infection rate. For example, when rapid replication within-host also renders the pathogen more infectious. One can also substitute the Gaussian structure of the mutation steps $\Delta\psi$ by any other arbitrary distribution. Hence, our formalism can accommodate also non-random walks in the inter-host fitness space. To examine this, we

now added in **Supplementary Section 4** results obtained from a non-zero Gaussian mutation process. Additionally, in our analysis of COVID-19, we used a sigmoidal fitness function ϕ_μ , highlighting the potential flexibility of our methodology.

Revisions

Our revised model presentation, and its potential generalizations, have now been implemented in the paper's main text, accompanied by supporting numerical analysis in the newly added **Supplementary Sections 2-4**. We also added an illustrative figure (**Fig. 1**) to explain the link between our modeling framework and the intra/inter-host evolutionary dynamics.

9. Comment

Last paragraph: There is a lack of understanding of how vaccines work and how resistance against vaccines can develop. Foremost, vaccines induce a spectrum of antibodies, similar to poly-pharmacology with a spectrum of targets. Thus, there is not 'one mutation that induces resistance', but many have to co-occur for vaccines to lose efficacy. Secondly, resistance emergence occurs while virus replicates within an infected individual. If a vaccine is applied and it protects, there is no virus replication, because the person does not get infected in the first place, hence resistance may not emerge. In a non-vaccinated individual there would not be selective pressure for resistance to emerge (COVID exception below). If a vaccine gets applied after infection, resistance may eventually get selected. This paper <http://dx.doi.org/10.1098/rspb.2016.2562> gives a good overview.

However, and most importantly: In Covid a natural infection and vaccines induce the same spectrum of immune responses/antibodies. Thus, in a population that has experiences infection, a virus gets a transmission advantage if it escapes natural (and therefore also vaccine-induced) immune responses. The authors' model completely fails to capture this phenomenon.

Response

Following this and the other reports we realized that our original application, focused on vaccine resistance was lacking: (i) It has been thoroughly studied in the past, and it was unclear to us that our analysis offered sufficient novelty; (ii) As this comment notes, there were overlooked subtleties in our implementation. Specifically, as the Referee indicates, it is often not a binary distinction, in which a single mutation helps the pathogen penetrate the vaccine induced immunity. Rather, resistance builds up through a series of mutations, in which the pathogen gradually penetrates the spectrum of immune responses.

Together with the Referee's second observation about escaping natural immunity, we decided that, indeed, in the context of COVID-19, this may offer a much more relevant demonstration of our formulation. In fact, it touches precisely on the main advantage of our proposed framework – the fact that we do not have to assume a discrete (typically binary) set of variants, but rather can capture the effect of evolution's incremental accumulation of fitness.

With this in mind, we have now switched focus from vaccine resistance to the evasion of natural immunity, a phenomenon we have (since our original submission, prior to the appearance of the δ -variant) now witnessed occurring several times with respect to SARS-CoV-2.

Incremental gains in resistance.

The main point of our mathematical formulation is that we can treat the continuous nature of the evolutionary dynamics, as it

Fig. R2. Cross-infection fitness. ϕ_μ vs. μ can be (a) binary; (b) continuous; (c) characterized by few cross-infection boosts or (d) centered around specific genetic traits.

interplays with the inter-host spreading process. In simple terms, our methodological advance is focused on our ability to capture the gradual dynamics of evolution. We, therefore, now designed our COVID-19 demonstration to exploit precisely this aspect of our analysis. Specifically, in the context of reinfection, we assume that the probability to evade the wild-type induced immunity increases *continuously* as the pathogen becomes more genetically distant. We therefore now assign a reinfection fitness ϕ_μ that continuously rises as the variants drift away from the wild-type $\mu = 0$. This allows us to assess the probability of reemergence, *i.e.* a second infection wave, as we show in the newly added **Fig. 5**.

The strength of our formalism is that it can accept any function ϕ_μ , and can therefore be tailored to address all forms of accumulation of cross-infection capabilities. To capture this we set the variants μ according to their genetic distance from the wild-type, such that $\mu = 0$ is the wild-type, and as additional mutations accumulate μ increases. With this setup, the classic binary model, *i.e.* variants can either reinfect or not, is characterized by a step-function (**Fig. R2a**): all variants that cross a certain critical genetic distance are able to penetrate the wild-type induced immunity. Variants below this threshold cannot. More realistically, we can take a smooth sigmoidal ϕ_μ , in which, as variants become more genetically distinct from the wild-type ($\mu = 0$) they *gradually* increase their reinfection probability (**Fig. R2b**). Other plausible functions include ϕ_μ with several steps (**Fig. R2c**), each representing a *milestone* mutation that gains an additional evasion probability, or a multi-peaked function, in which specific variants (and their immediate genetic surrounding) acquire the traits that help them evade immunity (**Fig. R2d**). Hence, we can design ϕ_μ , quite generally, around the most biologically relevant fitness function.

Beyond SIR. In line with the current narrative of our paper, focusing the above discussion on the COVID-19 disease cycle helps us demonstrate two potential strengths of our proposed framework:

1. **Generalizability.** While our analytical observations are achieved around SIR and SIS, their insights range beyond these simplified models, and can be readily generalized to *any* compartmental model. For that matter, our application to COVID-19 is not only timely, but also showcases our framework's broader insights.
2. **Realism.** While immunity evasion (just like vaccine evasion) is, in and of itself, not novel, our framework allows us to model it in, what we believe, is a highly meaningful fashion. Instead of a binary approach, in which a single mutation transforms the wild-

type into a breakthrough strain that can reinfect the recovered population, we allow mutations to build up continuously. As the evolved strains become more and more genetically distinct from the wild-type, they acquire a gradually increasing probability ϕ_μ to evade its immunity. Such *step-by-step* evolutionary dynamics are, generally, difficult to model, yet they fit quite naturally into our dynamic formulation, which is specifically designed to enable a continuous accumulation of fitness.

Therefore, our current COVID-19 demonstration is not simply *another* application, but rather a specific exposition, tailored to support our scientific advance.

10. Comment

In my opinion, an SIS model is completely the wrong approach to model transmission dynamics and pathogen-evolution, w.r.t. SARS-CoV-2. After an infection, the infected person is (at least for some time) not-susceptible to infection anymore (reconvalescent). My suspicion for the authors choosing this model is to avoid an exhaustion of susceptibles. However, there are better ways to do that.

Response

We agree with the Referee that SIS is not the ideal setting on which to examine the role of mutations. Primarily because, under SIS the pandemic state (prevalence $\eta > 0$) remains stationary indefinitely. This allows for an unrestricted window of opportunity for mutations, as opposed to the limited time-window in, *e.g.*, SIR. As this point has also been raised by the other Referees, we now reformulated our paper around SIR dynamics, leaving the SIS analysis for the Supplementary Information (**Section 2**).

We also wish to emphasize that both in our present submission, as well as in our original manuscript, the simulation of COVID-19 was not done using SIS nor SIR. To realistically model the spread of SARS-CoV-2 we collected data on the actual COVID-19 disease cycle, which includes many more compartments, such as exposed (E) or asymptomatic (I_{AS}) individuals. We also used empirical data on the spread from a set of different countries to extract the observed (wild-type) infection rate β_0 . The precise disease cycle, its parameters and the relevant data analysis are detailed in **Fig. 5** and elaborated in **Supplementary Section 5.3**.

11. Comment

The evolutionary model (a zero-centered Gaussian shift in fitness space) denotes an incorrect model of viral evolution, at least at the population level. There is a nice theoretical paper by Martin Novak, illustrating that ‘downwards’ moves in fitness space are extremely unlikely.

Response

We wish to thank the Referee for the opportunity to clarify this issue, which touches on two points that, we agree, were not properly articulated in round 1:

First, we note that there is nothing *sacred* about the zero-mean Gaussian form of Eq. (10).

Other, more realistic distributions can be easily introduced into our formulation. In that sense, our assignment of $\Delta\psi \sim \mathcal{N}(0, \sigma^2)$ represents a *didactic choice*, to focus on a simple, non-specific, if you wish – *default* – distribution, on which to first present our application. Indeed, any other choice would require *specific* motivation. Hence, our goal is not to argue that *this* is the most appropriate distribution by which to model evolutionary dynamics, just that *this* is the most appropriate *demo* of our modeling framework.

Indeed, our *methodological* contribution, as explained in our response to **Comment 2**, is that we can couple *any* disease compartmental model with *any* form of gradual, continuous and random evolutionary process. This methodological framework, as we now emphasize, is quite flexible and can accept different choices for both components. Hence, the user, *not us*, decides on the epidemic model of their choice (SIS, SIR, COVID-19 etc.) and couples it with the fitness distribution they believe is most appropriate. We, by no means advocate that a specific choice, *e.g.*, our first application on SIR with zero-mean Gaussian $\Delta\psi$, is the *correct* choice. This all depends on the specific disease/evolutionary dynamics at hand.

We now emphasize this point in our discussion, explaining that our method is, in essence *agnostic* about the specific fitness distribution. We also go further and demonstrate our formulation of other fitness distributions. For example, in the newly added **Supplementary Section 4** we discuss non-zero mean Gaussian evolutionary steps $\mathcal{N}(\chi, \sigma^2)$, in which, on average, fitness gains are either positive ($\chi > 0$), as the Referee hints in their comment, or negative ($\chi < 0$), representing *downward* steps; the latter may arise in the presence of anti-correlations (see below). Furthermore, in **Supplementary Section 2** we recover the same results under SIS dynamics, demonstrating the flexibility to select the epidemic model of choice.

Finally, in our revised COVID-19 application we demonstrate both degrees of freedom: first, using an alternative, much more elaborate disease model, second designing the fitness distribution ϕ_μ via a sigmoidal function (see **Fig. R2**), which offers an alternative to the *default* Gaussian form of ψ_μ .

Are downward steps likely? Having said all the above, we also wish to emphasize that we are not convinced that zero-mean $\Delta\psi$ are as unrealistic as the comment suggests. The crucial point is, that, in our modeling the selection occurs *intra*-host, and hence the evolutionary process favors strains with increased intra-host fitness $\varphi_\mu(i)$. This, in many cases, has no bearing on the *inter*-host fitness ψ_μ , and hence $\Delta\psi$ may, indeed, be randomly shifting up or down. Of course, once transmission occurs, in the inter-host fitness landscape only positive $\Delta\psi$ are selected. But this process, driven by the inter-host dynamics, only kick-in after the initial intra-host replication, where there is no selection for higher ψ_μ .

In case the intra and inter-host fitness parameters, $\varphi_\mu(i)$ and ψ_μ , are interdependent, then we agree that $\Delta\psi$ is no longer a zero-mean random variable, as the intra-host selection also affects, indirectly, the inter-host fitness. However, here the φ, ψ correlations can be either positive *or* negative, and hence the typical $\Delta\psi$ can be both upward and downward. Such downward moves arising from potentially negative

correlations between competing traits has, indeed, been observed in the literature,^{15,16,17} and, as explained above, also captured in our newly added **Supplementary Section 4**, where we treat both positive and negative correlations between the fitness parameters.

To summarize, we do believe that zero-mean $\Delta\psi$ can capture realistic evolutionary dynamics under our intra/inter-host interplay. Yet, by no means does our contribution hinge on this assumption, as we can (and in fact do) generalize to other realistic forms of fitness gain distributions. Still, regardless of how likely or unlikely such zero-mean $\Delta\psi$ are, we hope the Referee will agree that, as our first demonstration, this choice represents the most natural methodological distribution upon which to apply our formalism.

12. Comment

Again, the authors' model does not capture SARS-CoV-2's evolutionary dynamics, which is largely driven by the fact that a mutation has to become selected within the infected host, before transmission.

Response

This touches again on our original description which did not properly address the distinction between intra and inter-host evolutionary dynamics. We believe our response to previous comments (**Comments 1 – 3** in particular), and our current model description in the revised submission address this point.

13. Comment

Terminology: Much of the putatively "surprising" conclusions from the model are because of poor terminology. If you correctly define

- a. $R(t_0)$; the reproduction number at the onset of infection (or of the wildtype)*
- b. $R_i(t)$ reproduction number of mutant i at t*
- c. $\langle R(t) \rangle$ the average reproduction number of the population at t*

You recover all well-known phenomena. Moreover, some other terminology is quite sloppy.

Response

Perhaps our original presentation was not clear enough. We hope that now, following our extensive rewrite, all of these issues will become more evident. We fully agree with the Referee's interpretation, and, in fact, the terms mentioned in the comment are defined precisely as appears therein. So – indeed – *no magic* is involved in our results.

Still, regarding novelty, we, once again, emphasize that our contribution focuses on two

¹⁵ C.T. Bergstrom, P. McElhany & L.A. Real. Transmission bottlenecks as determinants of virulence in rapidly evolving pathogens. *Proc. Natl. Acad. Sci. USA* **96**, 5095 (1999).

¹⁶ M.A. Nowak & K. Sigmund. Evolutionary dynamics of biological games. *Science* **303**, 793 (2004).

¹⁷ Bonhoeffer, S. and Nowak, M. A., Mutation and the evolution of virulence. *Proceedings of the Royal Society of London. Series B: Biological Sciences*, 258, 133-140 (1994).

crucial advances (see also our responses to **Comments 1,2**):

1. **Quantitative predictions.** Offering not just qualitative descriptions, but rather defining the relevant control parameters, their precise critical points of transition and the nature of each transition.
2. **Methodology.** Deriving a systematic formulation that allows to couple any compartmental model (SIS, SIR, COVID-19, etc.) with any form of continuous fitness distribution (Gaussian, Sigmoidal, etc.).

All of the above is now articulated more clearly in the revised paper, where we revisited our terminology, and describe our specific contributions within the existing state-of-the-art.

14. Comment

The fact that the probability to evolve resistance/fitness depends on the population size is not fundamentally new. You should however, be a bit more precise: It depends on the number of virus replication events: many individuals being infected, or few individuals being infected for extended durations. From this it is fairly straight-forward to derive mitigation recommendations.

Response

We agree. The probability for a given mutation, indeed, depends on the number of replication events. This, in turn, depends on both the number of hosts (\mathcal{J}_0) and on the typical number of replications within each host (ρ), as noted by the Referee. On this note, we wish to emphasize, again, that while the *qualitative* observation – that with many infections comes an increased risk of mutation – may seem straightforward, the *quantitative* outcomes, one cannot *guess* absent our analytical/numerical observations.

In the present context these observations are:

1. The critical transition along the R_0, σ phase-diagram, from an infection-free to a mutation-driven contagion, and, most crucially, *how* this transition dependence on \mathcal{J}_0 (we emphasize, *how* it depends, not simply the fact that it depends).
2. Using 1 to show how the phase boundary closes-in as we delay our response time, finding the probability of mitigation failure in function of t_R . Once again, we do not just proclaim that early response is important, but obtain the mathematical foundations of this time-sensitivity, helping precisely quantify this effect (see our newly added **Fig. 4**).
3. The hysteresis observed under SIS, which is a direct (and quite surprising) outcome of this \mathcal{J}_0 dependence. Hysteresis is an unobserved phenomenon in the epidemic spreading landscape, and yet – as we discuss in **Supplementary Section 2**, it has crucial implications on mitigation.

15. Comment

Page 7: I do not understand where the claim “Most mitigation strategies seek a minimal approach, aiming to drive R_0 just below unity” stems from. In my understanding there is a difference of what we seek to do and what we can do.

Response

What we meant in this statement is that our response to an epidemic, especially when based on social distancing, is expensive, both economically and psychologically. Therefore, when possible, societies will seek an optimal response, in which lock-downs, masking or social isolation will be employed as minimally as possible. Often, of course, we do not know how to (or simply cannot) precisely tune our response to achieve such optimality. Yet, still, quite generally we wish to avoid overreacting, due to the heavy socioeconomic price of most mitigation measures. We now changed the wording of these claims to make this point clearer. We also emphasize that this specific claim is by no means crucial for our overall message, and even if we cannot fine-tune our response, the mitigation guidelines that we put forth continue to be equally worthwhile.

Having said that, we wish to note that, there are, in fact, rather strong indications, that even though society cannot fully control R_0 , during COVID-19 many countries have naturally converged to $R_0 \approx 1$.^{18,19} This empirical observation is likely rooted in people’s natural response to the perceived prevalence of the pathogen, restricting their contacts when prevalence is high, then letting loose as numbers decline.²⁰ This leads to an effectively *minimal response*, sustaining an R_0 that fluctuates around unity.

Of course, while people respond to the infections around them, what they cannot perceive is the *invisibles* risks. Specifically, the risk of mutations, that may arise as a result of the prolonged pandemic state that is experienced under *soft mitigation*. This is precisely where our phase-diagrams can help, quantifying this invisible risk under any given combination of R_0 and σ .

16. Comment

Volatile phase: This is not new: I strongly suggest that the authors consult the works that Manfred Eigen on quasispecies theory and error catastrophe, etc... (!developed in the 1970ies!).

Response

We believe that we have been misunderstood on this point. As we understand, Manfred Eigen’s error catastrophe discusses the bounds on the rate of mutation, above which the organism fails to produce viable offspring. Such error threshold is not present in our

¹⁸ F. Arroyo-Marioli, F. Bullano, S. Kucinskis, & C. Rondón-Moreno. Tracking R of COVID-19: A new real-time estimation using the Kalman filter. *PLoS ONE* **16**, e0244474 (2021).

¹⁹ G. Lobinska, A. Puzner, A. Traulsen, Y. Pilpel & M.A. Nowak. Evolution of resistance to COVID-19 vaccination with dynamic social distancing. *Nature Human Behaviour* **6**, 193 (2022).

²⁰ J. S. Weitz, S. W. Park, C. Eksin & J. Dushoff. Awareness-driven behavior changes can shift the shape of epidemics away from peaks and toward plateaus, shoulders, and oscillations. *Proc. Natl. Acad. Sci. USA* **117**, 32764 (2020).

modeling framework, and hence even under extremely high σ , *our* pathogens can still successfully reproduce. Hence, our volatile phase is unrelated to Eigen's predicted reproduction failure. The phenomenon that *we* observe in our volatile phase is, indeed, not about the inability to reproduce, but rather about the failure of the selection process.

To understand this, consider a strain μ producing, due to mutation, a fitter strain ν , *i.e.* $\psi_\nu > \psi_\mu$. The fitter ν has an evolutionary advantage – it spreads more rapidly than μ in the inter-host epidemic dynamics. It will, therefore, benefit from selection in the inter-host fitness space, covering an exponentially larger portion of the susceptible host population. Such natural selection, however, is not instantaneous, and requires some time, τ_s , for ν to gain ground and overcome its predecessor's already acquired prevalence. The challenge is that, as it spreads and gains ground in the inter-host space, ν itself undergoes mutations, and within a timescale τ_m , its successors, replicating within other hosts, will mutate, and effectively *forget* their father's fitness gains. Hence, selection must be fast enough to lock-in the evolutionary gains before random mutations eliminate them. In simple terms, we need $\tau_s < \tau_m$.

The crucial point is that while selection (τ_s) is driven by the inter-host epidemic dynamics, mutations (τ_m) are governed by the intra-host replication process. Hence, these two timescales are almost independent, and the system may have rapid mutations (large σ) accompanied by potentially slow natural selection. Under these conditions, pathogens mutate randomly, and selection is insufficient to ensure an uphill trend in fitness space (ψ).

Hence, the volatile phase, as opposed to the classic error catastrophe, does not imply an inability to replicate. Rather it describes an insufficient selection rate to ensure that fitter strains gain dominance over the host population. Under these conditions, pathogens can indeed replicate, and, in fact, thanks to the volatility, they are likely to reach one or more instances of a breakthrough mutation. However, overly rapid mutations prohibit such breakthrough mutations from actually breaking through, as their inter-host offspring are too genetically diverse.

Revisions

We have now added a specific discussion, along the lines of the above response, to explicate this distinction.

17. Comment

Page 9 first paragraph: Refs 51-53 are quite unsuited here. There are good models of intra-host SARS-CoV-2 dynamics.

Response

This is no longer relevant considering the major revision to our paper's narrative and model description. We thank the Referee for pointing us in the direction of the relevant literature throughout this review.

18. Comment

COVID model: It is unclear whether the recovered re-enter the susceptible population, and if not if there are any effects of susceptible exhaustion.

Response

Recovered individuals (R) *cannot* be reinfected by the wild-type. This is now clearly explained, both in the text and in the accompanying illustration of **Fig. 5a**. We also further clarify this point in **Supplementary Section 5.3**, where we explicitly list all the transitions of the SARS-CoV-2 infection track. Of course, mutant strains μ *can* potentially reinfect a wild-type recovered host, with probability ϕ_μ , that increases as we drift away from $\mu = 0$. Indeed, this represents *the* subject matter of this part of our analysis.

19. Comment

Small things: There is inconsistent use of parameter names, page 9, middle, vaccination rate.

Response

Corrected. Thanks.

In summary, our paper has undergone a thorough revision, with special attention to aligning it with the existing literature, and emphasizing its true contribution within this context. We are quite certain that the Referee will find the current version significantly more grounded, and its novelty better articulated.

On a personal note, we found this report to be critical, yet – at the same time – highly insightful. We believe it truly helped us shape up our contribution.

REVIEWER COMMENTS

Reviewer #1 (Remarks to the Author):

The paper has been deeply modified with respect to the previous version and even the main topics seem to be quite different. While the previous version gives emphasis to the effect of mutation in the context of mitigation measures and vaccination and the possible presence of hysteresis. However, as already discussed, these potentially interesting topics are based on the use of a SIS model which is not a realistic description of the epidemic propagation even in an approximated way. These parts now have been moved to appendix, while the paper focuses on the more realistic SIR model and its phase diagram. In particular, it is shown that the epidemic may spread even if the initial R_0 is smaller than one, if the typical time of the mutations is shorter than the typical time of the extinction of the epidemics. In this perspective, I think that the result is not surprising, therefore, maybe it is not suitable to be published in Nature communication due to its limited impact.

Moreover, before publication even on a different journal, the author should answer the following important points:

- The paper has been deeply modified in order to take into account a much more complicated model of mutations dynamics. With these changes the presentation in my opinion turns out to be a bit convoluted, while the previous version was much more clear. In particular the authors introduce intra-host mutations, and it seems that if these mutations act independently on the intra-host and inter host-fitness the epidemic evolution turns out to be described by the diffusive process considered in the first version while if there are correlations among intra and inter host fitness a drift emerges in the diffusive dynamics. If this is a correct view of the new model this should be more clearly explained; in particular in the new version it is not clear how the epidemic parameters α and β are varied due to the fitness mutation and which are the characteristic time scales. In general, I think that maybe it is not so interesting to devote a lot of work to discuss the evolution in the mutation space if the overall effect is only a diffusion of a fitness which is a multiplicative factor of the epidemic parameters.

- In the supplementary section 1 the authors obtain the effect of fitness fluctuations on the boundary of the active phase in the phase diagram. In the calculation the fitness ψ multiplies $R_0 = \langle k \rangle \beta_0 / \alpha_0$ i.e. it seems that the fitness multiplies β_0 (notice that α_0 is present also in a different point of $\exp(-at)$ and it is not varied under fitness changes). Therefore, I expect that the expression for the boundaries of the active phase holds in the model where the fitness is coupled to the infection rate Eq. (3.1 suppl) while the author uses the expression (1.14 suppl) to study the case where fitness varies the recovery rate α .

- In the previous version, it was clear that the volatile phase emerges as consequence of the boundary ψ_{\max} in the fitness combined with a large fluctuation of the mutations. Here, it seems that it is generated by large fluctuations only which should be quite surprising.

Reviewer #3 (Remarks to the Author):

Overall, the authors did a very good job in revising and streamlining the manuscript, which is now much improved and clearer. Particularly, paragraph 1 “Evolving pathogens and network contagion” is appreciated. I have a few comments that the authors should address before publication.

1. Figure annotations: In the figures it may help if you just, instead of using the mathematical notations, write what the variables are, e.g. ‘average number infected’? or ‘fraction infected’ instead of $\eta(t)$, intra-host fitness $\phi(l,0)$.

2. Variables: With regards to $\eta(t)$ it is unclear what it means: In the Introduction both ‘number infected individuals’ and ‘prevalence’ are stated. I think you mean ‘fraction of the population infected’.....same with $r(t)$ actually. In Eq (14): Introduce N . I guess you mean total population, but it is unclear. I suggest the authors go through the manuscript rigorously to introduce all variables in concise language.

3. Also, I think eq. (14) should be ‘greater than’ instead of ‘greater equal then’. (there is still a considerable probability that the pathogen vanishes for $\eta(\tau) = 1$, i.e. starting from one infected individual..... Explanation: Note that you derive the critical values by analyzing the corresponding ODE system, which approximates the evolution of the [expected number] of infected individuals. The underlying stochastic system is however bimodal (a number of trajectories will go extinct, while others grow). Hence $d/dt E[\text{Infected}] = 0$ ($= d/dt \xi$ in your manuscript) can also relate to the case where as many trajectories go extinct as there are infected individuals in those trajectories that grow. E.g. check eq. (1.12) in the suppl. Text.

4. Eq. (15): do you actually mean ‘approximately’, instead of the used ‘tilde’ sign (\sim) which rather refers to ‘sampled from’.

5. The COVID example models a scenario, as if the virus would gradually accumulate mutations as it spreads from individual to individual; This is actually not the driving force in COVID evolution (see limitations)

6. Include a limitations (or 'limitations and outlook') section.

Limitations:

i. The COVID example models a scenario, as if the virus evolves by gradually accumulating mutations, which it does not (I may be wrong here and it would be great if the authors depicted the maximum inter-host (?reinfection?) fitness in the population for Fig. 5 b& d). There are COVID-relevant factors not considered here:

a. Variability (rather bimodality) in the number of replication events before onwards transmission.

- E.g. infection of a subpopulation of immune-compromised individuals which drive the emergence of highly evolved strains, due to long durations of viral shedding (= a large number of intra-patient replication). There are reports of individuals shedding infection-competent virus for > 6 month. Here is an example: PMID: 35120605. These individuals may be more common in regions with e.g. HIV-prevalence (= immunocompromised), like e.g. sub-Saharan Africa (HIV-prevalence up to 10%).

- In the "normal" case however, SARS-CoV-2 hardly changes between transmission (there are < 1 substitutions on average in the entire SARS-CoV-2 genome for cases that are linked via transmission, i.e. PMID: 35062291, 34181711, 34248221, 34387545, 33688063). The consequence would be that $\sigma \rightarrow 0$. The weakness of the utilized model is that σ has to be quite large in the model to reproduce a multi-wave, mutation-driven pandemic.

In summary: In a small fraction of individuals, there may be a large random walk in the inter-host fitness landscape, while for the majority this walk is quite small. Or more general: heterogeneity in pathogenesis can possibly lead to a mutation-driven contagion dynamic.

b. Zoonotic reservoirs beyond the control of a policy maker

ii. The mutation-fitness model (Gaussian) is of course oversimplified. In particular, the long tail towards increasing fitness values is quite unrealistic. Please state these limitations and discuss possibilities to improve the model.

Minor:

Figure 2 and text on page 5/6: maybe rearrange the panels or the text such that the order of figure panels that you discuss is 2a/b, then c/d, then e/f, instead of a/b, e/f and c/d as it is currently the case.

Reviewer #1

1. Comment

The paper has been deeply modified with respect the previous version and even the main topics seem to be quite different. While the previous version gives emphasis to the effect of mutation in the context of mitigation measures and vaccination and the possible presence of hysteresis. However, as already discussed, these potentially interesting topics are based on the use of a SIS model which is not a realistic description of the epidemic propagation even in an approximated way. These parts now have been moved to appendix, while the paper focuses on the more realistic SIR model and its phase diagram. In particular, it is shown that the epidemic may spread even if the initial R_0 is smaller than one, if the typical time of the mutations is shorter than the typical time of the extinction of the epidemics. In this perspective, I think that the result is not surprising, therefore, maybe it is not suitable to be published in Nature communication due to its limited impact.

Response

As the Referee notes, the paper has indeed been thoroughly revised, balancing between the comments made by all Referees. The transition from SIS to SIR was encouraged by all reports, and, as the comments notes, enhanced the relevance and realism of the paper. This transition, from a theoretical perspective had little impact, as the vast majority of observations on the pandemic phase-diagram carried over from SIS to SIR quite naturally. The main differences were, as the Referee correctly notes: (i) The absence of the hysteresis phenomenon, which was unique to SIS; (ii) The limited time for mutations to accumulate, due to the SIR dynamics, in which infections are only present within a bounded time-window.

We agree with the Referee that item (i) captures an intriguing phenomenon that strengthens the theoretical exposition and discussion of our model. Yet, on the other hand, item (ii), introduces a crucial realistic factor, that affects the mutation dynamics, and enhances the paper's relevance. Indeed – under SIS, once the disease is able to spread, there is unlimited time to continue and accumulate mutations for an ever-fitter pathogen. This is avoided in SIR, as there, even a pandemic pathogen, eventually exhausts the susceptible population and tapers off. Keeping both discussions, we felt, SIS *and* SIR, risks overburdening the paper and losing focus.

Therefore, in our revised version we arrived at, what we believe, is the right balance: placing our SIR analysis at the spotlight, then complementing it with our SIS analysis in the supplement. To make sure the interesting item (i) – hysteresis - is not lost on the casual reader, we include a dedicated paragraph in the main text explaining it and a reference to the relevant figure in the Supplementary Information. Following this comment – we now further expanded the main text discussion on the matter, to ensure the message shines through.

We also wish to emphasize, that apart from the hysteresis – all other interesting features of the phase-diagram remain relevant under SIR. This includes the analytically derived

phase-boundaries, the relevant control parameters (R_0, σ) and the nature of the different transitions, specifically, the discontinuous first-order transition to the mutation-driven pandemic phase. This latter prediction, we note, is quite unique in the epidemic spreading arena, which is almost always characterized by continuous transitions. All this, together with the *added spice* of the hysteresis under SIS, constitute what we view, as our main novelty: *offering a comprehensive, analytically derived, characterization of the pandemic phase-diagram under pathogen evolution.*

We hope the Referee will agree, that even if some aspects of this phase-diagram can be intuitively reasoned (see our response to **Comment 2** below), it still represents a rather fundamental advance. Quite often, such modelling frameworks that encapsulate, within few model assumptions, the potential complex transition patterns that we observe (*e.g.*, in **Figs. 2,3** and in **Supplementary Fig. 1**) emerge as fertile breeding grounds for future analyses.

2. Comment

In particular, it is shown that the epidemic may spread even if the initial R_0 is smaller than one, if the typical time of the mutations is shorter than the typical time of the extinction of the epidemics. In this perspective, I think that the result is not surprising, therefore, maybe it is not suitable to be published in Nature communication due to its limited impact.

Response

This comment echoes the sentiment of the Referee already in round 1, and stands in sharp contrast with the assessment of their peers (*e.g.*, *really interesting and important results*, Referee 2). We, of course, agree that the *conceptual idea* that rapid enough mutations can overcome the pandemic extinction seems rather intuitive. In fact, in our grey box, discussing our timescale analysis, we explain just that, stating that the pandemic transition is driven by the fact that the critical mutation is reached before the pathogen is eliminated, *i.e.* $\tau_c < \tau_1$. However, we disagree that this renders our findings to be of *limited impact*.

First, because, as we explain below, this is a very narrow interpretation of our results, which, in reality, go way beyond the trivial statement above. Second - in our view, the relevant criterion is not whether the results are *surprising*, but rather, whether they are *insightful*. We hope to convince the Referee below of both merits.

Insight. Consider the comment's summary of our paper that rapid enough mutations can lead to a pathogen revival. Indeed, on its own - a *not surprising* statement. The crucial point is, that this statement still leaves an array of unanswered questions, which cannot be answered without our methodical scientific analysis. For example, how rapid is *rapid enough*? Is there anything as *too rapid*? And when we are in the right zone of pathogen revival, how does this revival unfold - is it a gradual changeover, a continuous phase-transition or an abrupt jump? What precisely are the boundaries of this revived state? What is the fundamental modeling framework for the process, and what are its relevant parameters?

These questions cannot be answered by common wisdom alone, and require systematic analysis. First, defining the fundamental *model* – which has to be detailed enough to be scientifically meaningful, and yet simple enough to enable analytical treatment. Then – analyzing this model to obtain both *qualitative and quantitative results*. Finally, looking back at our results and gaining *retrospective insight* on the generalizable intuition behind them.

This retrospection is, indeed, for *some* (not *all* – see below) of our model’s prediction – quite intuitive. However, retrospection, by its nature, rarely results in something *surprising*. What it does provide – is *insight*. Typically, a deeper understanding on the aspects of our model’s predictions that extend beyond its specific assumptions, and hence generalize to more complex (and hence more realistic) dynamics.

Before we examine all the above components in *our* analysis, let us first exemplify them in, what is now considered, a canonical result in epidemic spreading:

The SIS spreading dynamics

Consider the classic SIS model and its well-known healthy vs. pandemic phases. One can trivialize this result, stating that it merely shows that *the more contagious is the disease the more prevalence it gains*. This is of course, *not surprising*, and can be guessed even without a model. Yet, it is an unfair presentation of the actual observation of the SIS phase-diagram, which is – indeed – deeper:

Model. The SIS compartments offer a fundamental modeling framework - realistic enough to offer testable predictions, and yet simple enough to allow for an analytical solution.

Quantitative and qualitative predictions. The analytical solution goes way beyond the trivial statement above. It identifies the relevant control parameter R_0 , then finds that as R_0 is increased prevalence, indeed, rises, however, in a non-trivial fashion – via a second-order phase-transition. Finally, the analysis predicts the precise critical transition point at $R_0 = 1$.

Retrospective insight. Once we identify the critical transition point, we can return to our original intuition and gain deeper insight. Indeed, it is true that the greater is R_0 the more prevalent is the disease, yet there is something unique about the specific point of $R_0 = 1$. It captures a state where, on average, an infected individual transmits the pathogen to precisely one other susceptible peer. This, we emphasize, is rather expected (*i.e. not surprising*): if secondary transmissions are below one, the prevalence decays with each infection cycle, whereas if they are above one, the infections proliferate with each instance of transmission. Still, the fact that it is *expected*, does not render it uninteresting or lacking insight. In fact, it offers crucial insight on the generalization of the model analysis, establishing the principle of the *reproduction number*, which indeed, extends to practically any disease spreading dynamics, well beyond the stylized SIS model. Hence, this observation continues to guide us also in our response to real-world pandemics, for which SIS or SIR are too simplistic.

The crucial point is that this quantitative prediction, identifying R_0 as the control parameter, and pointing to the unique role of $R_0 = 1$, while retrospectively intuitive,

could only be obtained through mathematical analysis.

Implications. Mapping the pandemic phase diagram, even for this simple case, continues to guide us in designing mitigation policies. The fact that the transition to the pandemic state is of critical nature, indicates that we should not simply aim to reduce R_0 , but that there is a *sacred* goal of pushing R_0 below unity. This is now part of common knowledge, but certainly not something that could have been *guessed*, absent the mathematical formulation of SIS or SIR.

Our analysis

Our analysis follows a very similar path. Before we expand on this issue, let us first clarify that we do not claim that our work is as fundamental and potentially impactful as the SIS or SIR models. This, indeed, only time can tell. Yet, we do stand our grounds that dismissing it as *not surprising* is tantamount to the above, clearly unfair, trivialization of the SIS model. Here is how we view our contribution in this light:

Model. We propose a basic modeling framework to track the interplay of the inter-host spreading dynamics with the intra-host evolutionary dynamics. These two processes, each characterized by their own parameters and timescales, are *not* independent. On the one hand, the spreading dynamics are affected by the pathogen's evolving recovery/infection rates. This feeds the evolutionary dynamics into the spreading patterns. On the other hand, the selection process of the evolutionary dynamics is driven by the duplication advantage of the fitter strains, which is governed by the inter-host spreading dynamics. All of this interdependence is captured by our minimalistic model, and we believe it is not at all easy to *guess* its precise outcomes.

Quantitative and qualitative predictions. Using a combination of analytical and numerical tools we identify the relevant control parameters, which include the classic R_0 , but now, also the observed inter-host mutation rate σ . The latter, we emphasize was *assumed* in the previous version of our paper, but now it is extracted from the microscopic intra-host dynamics (see our response to **Comment 3** below).

With these relevant parameters at hand, we proceed to derive the precise phase-diagram, observing two separate transitions: the first, from the infection-free to the mutation-driven phase (what the Referee deems *not surprising*), and the second from mutation-driven to volatile (which, we guess, the Referee agrees, is less trivial). For both transitions we obtain their precise boundaries, and their transition dynamics, *i.e.* first/second order. This characterization of the phase-diagram, we argue, goes beyond simply stating that rapid enough mutations may break through. Rather it characterizes *quantitatively* how rapid is *rapid enough*, how much is *too rapid*, and how the actual transition unfolds. It therefore answers all the questions we listed above that naturally follow from the comment's summary of our contribution.

Agreeably, no special surprises arise in our analysis, as all results are, indeed, explainable. Yet, at the same time, this list of predictions is far from trivial – and, we believe – rather insightful.

Retrospective insight. Once the phase-diagram is constructed we can revisit our original

intuition and better ground it, by identifying the precise timescales that drive each of the observed transitions. These are identified as the time to reach the critical mutation (τ_1), the time to reach pathogen extinction (τ_c) and the time for the selection process to lock-in fitter strains (τ_2). It is precisely this insight that helps generalize the relevance of our predictions beyond the simple SIS/SIR frameworks. Indeed, regardless of how complex the actual disease cycle (*e.g.*, Covid) or how convoluted the intra-host evolutionary dynamics, the three competing timescales of Eqs. (14)–(17) will continue to govern the observed spreading dynamics. Hence, analyzing our simple, analytically tractable model, offers insight that naturally generalizes to potentially more complex scenarios.

Implications. Finally, the deeper understanding offered by our analysis leads to specific mitigation guidelines, that are once again, generalizable to other evolving pandemic scenarios. Let us focus on one example, which we actually think *is* surprising. In **Fig. 4** and its related analysis, we indicate the importance of early response. In contrast to the classic pandemic dynamics, that have no *memory*, pushing R_0 below the threshold at any point, ensures to place the system at the safe zone, *i.e.* the infection-less phase. Here, however, as time progresses, and more infections accumulate, the phase boundary itself shifts towards smaller R_0 . This is because the transition point depends on the prevalence $\eta(t)$. Hence, the same response that may have worked early on, when implemented too late may catch the pathogen already in a different state, for which our mitigation is no longer sufficient.

To be clear, there are many well-known reasons to responde early. But this notion, that the phase-diagram itself changes with time, and hence late response catches the pathogen at an already different phase - that, to the best of our knowledge, is quite novel.

This track – *model, prediction, retrospection, implications* - is precisely what distinguishes common wisdom from systematic mathematical modeling. It allows us to first, answer quantitative questions, then, refer back to our original reasoning, and gain additional insight, here on the timescales driving the competing spreading/evolutionary dynamics, and finally derive non-trivial implications.

Revisions

We have now further tightened the narrative of our paper, specifically in the **Introduction** and **Discussion** sections, to make sure these (*surprising and not surprising*) insights shine through.

3. Comment

Moreover, before publication even on a different journal, the author should answer the following important points:

- The paper has been deeply modified in order to take into account a much more complicated model of mutations dynamics. With these changes the presentation in my opinion turns out to be a bit convoluted, while the previous version was much more clear. In particular the authors introduce intra-host mutations, and it seems that if these mutations act independently on the intra-host and inter host-fitness the epidemic evolution turns out to be

described by the diffusive process considered in the first version while if there are correlations among intra and inter host fitness a drift emerges in the diffusive dynamics. If this is a correct view of the new model this should be more clearly explained; in particular in the new version it is not clear how the epidemic parameters alpha and beta are varied due to the fitness mutation and which are the characteristic time scales. In general, I think that maybe it is not so interesting to devote a lot of work to discuss the evolution in the mutation space if the overall effect is only a diffusion of a fitness which is a multiplicative factor of the epidemic parameters.

Response

This comment raises several issues, from the motivation behind our detailed intra-host modeling framework, to the details of its implementation. Let us address these issues one by one.

Modeling the intra-host mutations. This is a point that we, as authors, have also been grappling with. On the one hand, seeking to keep the story as streamlined as possible, we agree that delving into the details of the intra-host dynamics adds additional *weight* to the paper. Yet, on the other hand, we do not wish to jump over important details, specifically – we do not want to dilute too much of the actual machinery of the evolutionary dynamics. How then do we balance this *tug-of-war* between detail and conciseness?

After reading the reports of round 1, and after consulting with several experts in the evolutionary dynamics community, it became quite clear that our original balance, in which we simply *assume* a random walk in the inter-host fitness space, was too simplistic. Aiming for a broad interdisciplinary audience, we understood that it is key to link our analysis to the existing knowledge and understanding of intra/inter-host dynamics.

At its core, it is important to understand, that our paper is precisely designed to bridge between these two modeling frameworks:

- **Inter-host spread** follows the well-established models of network epidemiology, which can be solved via continuum differential equations (ODE).
- **Intra-host evolution** follows the existing models of evolutionary dynamics, which are discrete, stochastic and require probabilistic methods.

When they both unfold simultaneously they become deeply intertwined: the intra-host evolution impacts the spreading parameters, and the inter-host spreading dynamics drives the selection of fitter strains. This represents a complex interplay in which both modeling frameworks feed into each other.

But how do we connect, mathematically, these two distinct processes, when each requires its distinct mathematical toolbox?

Our solution to this coupling challenge is by mapping the intra-host evolutionary dynamics into the effective Gaussian random walk, which can be introduced quite smoothly into the continuum ODE equations of the SIS/SIR spread. And this is precisely the crucial point: this represents our *mathematical solution* to decoding this elusive interplay, but cannot be simply *assumed* as a replacement for the intra-host evolution

model itself. Indeed, from an evolutionary dynamics point of view, what we have learned from round 1, is that it is not at all clear that such mapping is, indeed, valid, and hence it cannot be assumed, but rather must be supported, starting from the fundamental microscopic description of the in-host dynamic, and leading to our simplified Gaussian random walk.

This is precisely the goal of **Fig. 1** and its related discussion in the section *Evolving pathogens and network contagion*.

Therefore, we view our paper's storyline is as follows:

- **Question:** What are the dynamics that emerge when inter-host spread is coupled with intra-host evolution?
- **Challenge:** The two processes behave like *oil and water* – they do not couple well mathematically, and therefore it is difficult to obtain analytical insight.
- **Solution:** Reducing the intra-host dynamics into an effective random walk enables us to analyze this interplay via timescale analysis.
- **Results:** The outcome is the complete pandemic phase-diagram of evolving pathogens.

With this narrative in mind, both models – intra and inter-host – are of equal weight, none is more central than the other. Hence, they each deserves their full treatment, before we begin introducing approximations and analytical simplifications. Of course, network scientists may feel that the evolutionary part is cumbersome, and evolution experts may wish to reduce the discussion on the SIR modeling details. These two sentiments were, indeed, clearly observed throughout our discussion in the reports.

With this in mind, we arrived at the conclusion that the most balanced approach is to describe both modeling frameworks in a way that is compatible with the views of their respective research community: *epidemic spreading as it is viewed by network scientists, evolutionary dynamics as they are viewed by evolution researchers*.

Compartmentalizing. To keep the flow of the paper, however, we did make sure to properly separate the two descriptions, allowing all readers to focus on the aspects of our modeling that *they* find most relevant. Therefore, the inter/intra-host model descriptions are clearly separated, each within its own subsection. As a result, in the current presentation if the reader wishes to directly skip to Eq. (8), *i.e.* skim through the evolutionary details, they will still be able to follow the story in full, only without a complete view of the intra-host dynamics.

To further strengthen the conciseness of the presentation we added a discussion in the **Introduction** that summarizes the coupling challenge and our proposed solution, along the lines of the above discussion. This now allows the interdisciplinary reader to follow our storyline in a flexible manner, from its fully detailed description, which the Referee experienced as perhaps *convoluted*, to its high-level summary, which cuts directly to our simplified solution.

Generalization. Along the process, we realized that there is, in fact, an added benefit to including the detailed modeling description of the intra-host processes. Indeed, it offers

an understanding of the microscopic origins of the observed inter-host dynamics, that in turn allow us to naturally generalize beyond the simple evolutionary random walk. This, we think, is already alluded to in the comment, when it mentions the correlated inter-intra-host fitness, which yields a biased random walk. It also motivated some of the discussion in round 1, which specifically raised these issues of alternative evolutionary paths besides the – *then assumed* – Gaussian random walk. Finally, it provides a useful *language* by which to discuss other, more general processes, as we do, for example, in our newly added discussion of the recurring infections. There we consider a non-Gaussian intra-host fitness space that gave rise to our sigmoidal ϕ_μ . Having a general language by which to describe the intra-host dynamics, made this adaptation quite natural, whereas in the previous version, it would have to be externally imposed into the modeling.

Implementation. We now refer to the second question in the comment, on the way in which we update the spreading parameters α and β . Upon infection, an individual i is injected with a specific strain μ , with parameters α_μ, β_μ . This strain duplicates within the host at a rate determined by its intra-host fitness φ_μ , and, in each duplication instance, transitions with probability p to one of its adjacent strains $\mu \pm 1$. After ρ replication cycles we arrive at i 's intra-host population with a distribution of strains given by $Z_\mu(i)$. During transmission, a random strain μ' , with spreading parameters $\alpha_{\mu'}, \beta_{\mu'}$, is extracted from $Z_\mu(i)$ and begins duplicating within the new host j .

The dynamics of this internal process directly determine the observed mutation rate σ . To understand this, consider the typical evolutionary step between transmissions, *i.e.* from α_μ, β_μ to $\alpha_{\mu'}, \beta_{\mu'}$. This step size is determined by:

- i. **Replication stability p .** The greater is p to more variability within each host and hence the larger is the typical inter-host evolutionary step, increasing σ .
- ii. **Intra-host fitness landscape variability σ_φ .** In case the landscape is diverse, *i.e.* large variance σ_φ , the intra-host replication will condense quite narrowly around the highest φ_μ strains, whose replication rate is significantly higher than the rest. This will result in low $Z_\mu(i)$ variability, and hence a smaller σ .
- iii. **Inter-host fitness landscape variability σ_ψ .** If the spreading fitness of the strains is broadly distributed, then the randomly selected strain from $Z_\mu(i)$ is likely to differ significantly from parent strain of the original infection. This translates to a larger σ .
- iv. **Replications per cycle ρ .** The more replications, the higher is the accumulated intra-host variability and hence the more rapid is the inter-host evolution rate σ .

In **Supplementary Section 5.2** we discuss the above in detail and measure precisely the impact of (i) – (iv) on σ . As we explain there, the observed effect of p, ρ and σ_φ is bounded, reaching an asymptotic *plateau* in their impact on σ . The remaining parameter σ_ψ , however, directly affects σ via an approximately linear relationship. Hence, this parameter is the main driver of the observed inter-host evolutionary dynamics.

Added benefit of our microscopic modeling. This analysis further demonstrates the added value in our current detailed evolutionary dynamics modeling, discussed above. Indeed, our *relevant parameter* σ , which was originally externally introduced into the

dynamics, represents an abstract observable, whose actual ingredients are hidden. With our current intra-host dynamics, we can link this parameter to the internal hidden processes, identifying how it emerges from the evolutionary timescales and statistics. This helps more firmly ground σ in the well-established language of evolutionary dynamics, connecting it to the frequently used parameters of the field.

4. Comment

In the supplementary section 1 the authors obtain the effect of fitness fluctuations on the boundary of the active phase in the phase diagram. In the calculation the fitness ψ multiplies $R_0 = \langle k \rangle \beta_0 / \alpha_0$ i.e. it seems that the fitness multiplies β_0 (notice that α_0 is present also in a different point of $\exp(-at)$ and it is not varied under fitness changes). Therefore, I expect that the expression for the boundaries of the active phase holds in the model where the fitness is coupled to the infection rate Eq. (3.1 supl) while the author use the expression (1.14 supl) to study the case where fitness varies the recovery rate α .

Response

Thanks – good catch. Indeed, our analytical derivation is presented in the context of β -evolution, not α -evolution, as we erroneously wrote in the Supplementary Information. We have now corrected this. Our choice to focus the mathematical analysis on β mutations is driven by simplicity: the resulting calculations are easier and more tractable. Under α mutations, the integral in Supplementary Eq. (1.6) becomes more cumbersome. Still, despite the analytical complication, our numerical results clearly indicate that both types of evolutionary processes, α or β , can be well-approximated by our analytical solution. It is, after all, an approximate analysis, and the minor differences in the derivation are, apparently, of no significant impact. Hence, we chose the strategy to present a tractable and non-cumbersome solution under β mutations, and then proceed to show numerically that our results are insensitive to this specific assumption, and cover also α mutations (**Supplementary Section 1.3**).

To be on the safe side, we now explicitly explain this point, regarding our analytical vs. numerical support, in the main text.

5. Comment

In the previous version, it was clear that the volatile phase emerges as consequence of the boundary ψ_{max} in the fitness combined with a large fluctuation of the mutations. Here, it seems that it is generated by large fluctuations only which should be quite surprising.

Response

The Referee is correct. We cannot observe the volatile phase under an unbounded fitness. The difference is that now, the upper bound on ψ is no longer an externally induced cap on the fitness. Rather it emerges naturally from our modeling of the intra-host dynamics. Indeed, in this modeling framework, which, we remind here – represents the common view of intra-host evolution, the system transitions via the rate matrix $M_{\mu\nu}$ between a

discrete and predefined set of variable strains. Therefore, there exists a naturally imposed boundary on the fitness, which, in turn, ensures the transition to the volatile phase.

Revisions

Our Eq. (16) now includes ψ_{\max} explicitly. It is therefore unambiguous that if ψ is unbounded, *i.e.* $\psi_{\max} \rightarrow \infty$ the volatile phase is averted.

We were sorry to see that the Referee did not recommend publication in *Nature Communications*, and hope our final clarifications here will convince them of the merit and potential insight in our results. Assessing insightfulness and impact is by its nature *subjective*, and therefore, it is quite expected that we may have a different view from that of the Referee. In contrast, when it comes to the more technical and *objective* side, we found that the Referee's comments on our mathematical analysis, in both rounds, were truly important and helpful. We therefore wish to thank the Referee for their deep reading and feedback – these are not at all taken for granted.

Reviewer #3

1. Comment

Overall, the authors did a very good job in revising and streamlining the manuscript, which is now much improved and clearer. Particularly, paragraph 1 “Evolving pathogens and network contagion” is appreciated. I have a few comments that the authors should address before publication.

Response

Thank you, for pushing us to include these improvements, and helping us bring our paper on par with the current thinking of the field.

2. Comment

Figure annotations: In the figures it may help if you just, instead of using the mathematical notations, write what the variables are, e.g. ‘average number infected’? or ‘fraction infected’ instead of $\eta(t)$, intra-host fitness $\phi(I,0)$.

Response

Of course. Good idea.

3. Comment

Variables: With regards to $\eta(t)$ it is unclear what it means: In the Introduction both ‘number infected individuals’ and ‘prevalence’ are stated. I think you mean ‘fraction of the population infected’.....same with $r(t)$ actually. In Eq (14): Introduce N . I guess you mean total population, but it is unclear. I suggest the authors go through the manuscript rigorously to introduce all variables in concise language.

Response

Rereading our text, we can now see how these confusions might arise. This is, as in round 1, again, a consequence of the inter-disciplinarity of the paper. Some notations and terminology are common in one community but ill-defined for others. We have now made a through scan of the text to smooth out these ambiguities.

4. Comment

Also, I think eq. (14) should be ‘greater than’ instead of ‘greater equal then’. (there is still a considerable probability that the pathogen vanishes for $\eta(\tau) = 1$, i.e. starting from one infected individual..... Explanation: Note that you derive the critical values by analyzing the corresponding ODE system, which approximates the evolution of the [expected number] of infected individuals. The underlying stochastic system is however bimodal (a number of trajectories will go extinct, while others grow). Hence $d/dt E[\text{Infected}] = 0$ ($= d/dt \xi$ in your manuscript) can also relate to the case where as many trajectories go extinct as there are

infected individuals in those trajectories that grow. E.g. check eq. (1.12) in the suppl. Text.

Response

We corrected Eq. (14).

We also fully agree with the distinction between the stochastic and deterministic versions of the SIR model, which is indeed relevant for our derivation. As is almost always the case, our mathematical analysis incorporates several approximations to achieve tractability. Among these is taking the deterministic ODE-based solution of the spreading dynamics, and confronting it with the probabilistic analysis of the fitness dynamics. The approximate nature of this coupling is now clearly stated in the text. Of course, considering how well it predicts the observed numerical results, we find this approximation to be rather effective.

In our view, the truly important discussion following every approximation is to understand what intricate features of the real system it may potentially overlook. And this point, we agree, touches precisely on the Referee's distinction. Our phase boundary, estimated from the condition that $\mathcal{J}(t) \approx 1$, captures two limits: in case $\mathcal{J}(t) \ll 1$ then it is safe to assume that practically all trajectories end in pathogen extinction. In the Referee's terminology this is captured by a bimodal distribution with the zero mode having almost all the probability mass. Conversely, if $\mathcal{J}(t) \gg 1$ the zero mode almost vanishes, and the majority of trajectories lead to pathogen proliferation.

In between these two extremes, under $\mathcal{J}(t) \approx 1$ the bimodality is, indeed, significant, having a finite probability to either grow or decay. This is precisely what we observe around the phase boundary: some realizations achieve a critical mutations and proliferate, others miss the opportunity and decay. Exactly following the Referee's prediction.

We interpret this to mean that if we are deep enough to the left or to the right of the critical point – the stochasticity in the spreading dynamics becomes unimportant, and hence can, in practice, be neglected. This is precisely the point of our approximation, which uses the *deterministic* ODE-based SIR modeling alongside the stochastic random walk of the mutation dynamics. However, while this is valid far away (right *or* left) from criticality, as we enter the vicinity of the critical point, suddenly this stochasticity becomes dominant, and the bimodality described in the comment, indeed, weighs in. Consequently, the outcome of the spread – growth or decay – is now driven by this bimodal distribution, and hence we expect to observe the aforementioned divergent trajectories.

Not surprisingly, in **Fig. 3b** of the main text we observe just that. Setting the system *at* our predicted critical point, we find that different realizations lead to different outcomes – proliferation (green) or extinction (blue). Hence, we can see this bimodality play out in practice. We now use this example to deepen the discussion on this stochastic bimodal effect. As we drift away, even slightly, to the left/right of criticality, the system rapidly selects its mode: infection-free or mutation-driven.

5. Comment

Eq. (15): do you actually mean ‘approximately’, instead of the used ‘tilde’ sign (\sim) which rather refers to ‘sampled from’.

Response

Good point. The \sim sign has several meanings. In this context it means *scales as*, or *grows like*. Such relationship captures equality up to multiplicative constants, or higher order corrections. For example, the force of gravity follows $F = Gm_1m_2/r^2$, which is commonly expressed as $F \sim 1/r^2$, i.e. it *scales as* an inverse square or distance. To avoid this ambiguity, we now explain precisely in the text what \sim means in Eqs. (15) and (16).

6. Comment

The COVID example models a scenario, as if the virus would gradually accumulate mutations as it spreads from individual to individual; This is actually not the driving force in COVID evolution (see limitations)

Response

Thanks. See our response below to **Comment 8**.

7. Comment

Include a limitations (or ‘limitations and outlook’) section.

Response

Done.

8. Comment

Limitations:

i. The COVID example models a scenario, as if the virus evolves by gradually accumulating mutations, which it does not (I may be wrong here and it would be great if the authors depicted the maximum inter-host (?reinfection?) fitness in the population for Fig. 5 b& d). There are COVID-relevant factors not considered here:

a. Variability (rather bimodality) in the number of replication events before onwards transmission.

- E.g. infection of a subpopulation of immune-compromised individuals which drive the emergence of highly evolved strains, due to long durations of viral shedding (= a large number of intra-patient replication). There are reports of individuals shedding infection-competent virus for > 6 month. Here is an example: PMID: 35120605. These individuals may be more common in regions with e.g. HIV-prevalence (= immunocompromised), like e.g. sub-Saharan Africa (HIV-prevalence up to 10%).*

- In the “normal” case however, SARS-CoV-2 hardly changes between transmission (there are < 1 substitutions on average in the entire SARS-CoV-2 genome for cases that are linked via*

transmission, i.e. PMID: 35062291,34181711, 34248221, 34387545, 33688063). The consequence would be that $\sigma \rightarrow 0$. The weakness of the utilized model is that σ has to be quite large in the model to reproduce a multi-wave, mutation-driven pandemic.

In summary: In a small fraction of individuals, there may be a large random walk in the inter-host fitness landscape, while for the majority this walk is quite small. Or more general: heterogeneity in pathogenesis can possibly lead to a mutation-driven contagion dynamic.

b. Zoonotic reservoirs beyond the control of a policy maker

Response

This is a very insightful comment that touches on potential variability in the individual host-disease cycle. Specifically, if we understand correctly, the Referee argues that in COVID-19, most of the variability between strains is generated within a small fraction of individuals whose intra-host replication extends for extremely long periods, having $\rho \rightarrow \infty$. Hence, most hosts have $\sigma \rightarrow 0$, with no significant mutation between infection and secondary transmission, while few have $\sigma > 0$, responsible for the observed mutations.

Reading this comment, we were not certain about the preferred course of action. On the one hand, it is straightforward to introduce this host variability into our simulations, it is, indeed, no more than a technical complication. On the other hand, we were unsure how well this supports the narrative of our paper. After all, the focus of our paper is not to guide specific COVID-19 policies, but rather to introduce our analysis framework, which we wish to keep as clean and focused as possible, without redundant technical *baggage*.

With this in mind, we designed our original presentation through, what we think is an insightful and, mainly, didactic path:

- i. **Modeling.** Present a basic first principles model of evolving pathogen spreading dynamics.
- ii. **Analysis.** Present an approximate analytical approach that helps predict the model's emergent phase-diagram.
- iii. **Broader insight.** Demonstrate a non-trivial (and timely) application, where both the spreading dynamics and the evolutionary dynamics are more complex, i.e. beyond simple SIR contagion and Gaussian fitness spaces.

Adding too much COVID-specific details to item (iii), we fear, may lose the focus of, what, in our view, is a currently concise and *clean* storyline. Hence, the host-variability, while clearly more realistic, and interesting on its own right, adds a component that, we feel, is external to our current exposition.

Moreover, considering the way in which we currently produce our observed phase-diagrams, in which each *pixel* is an aggregation of many realizations, this host-variability will have little impact on our results. It will, therefore, be a *complication* to the model, that lacks justification, as it will, in practice, play an insignificant role in our presented results. In simple terms – the reader may be confused by the motivation to add this detail to our analysis.

To be clear, we are quite certain that such bimodality in σ may have many implications –

an average medium σ it quite different from a majority with $\sigma \rightarrow 0$ alongside a minority with high σ . For example, if the long-term spreaders are rare, one may observe significant differences across realizations, depending on whether the spread caught up to one of these individuals or not. There are likely many other statistical observations that will be affected by this bimodality. However, we are not pursuing any of these statistical variations here, as they are not within the focus or scope of our present analysis. They would, indeed, be an undesired digression. It would, therefore, in our view, seem unnatural and, perhaps, redundant to include this detail in our modeling.

With these considerations in mind, we have decided to take the middle-ground: keep our modeling of the reinfection scenario clean and focused, but at the same time acknowledge and discuss this unique aspect of COVID-19, that is smoothed out in our analysis. Hence, we now added this discussion, including the relevant references, and suggest it as a potential expansion of our present exposition that should be studied on its own right.

Zoonotic reservoirs. Clearly these are relevant, but, we are quite certain that the Referee will agree - they are not part of our proposed modeling framework. Adding them would truly be foreign to our current paper's storyline, and contribute little to its main message.

9. Comment

ii. The mutation-fitness model (Gaussian) is of course oversimplified. In particular, the long tail towards increasing fitness values is quite unrealistic. Please state these limitations and discuss possibilities to improve the model.

Response

We thank the Referee for this valuable comment. Undoubtedly, setting the fitness landscapes $\varphi_\mu(i)$ and ψ_μ to be Gaussian is a simplification, chosen for conciseness and tractability. In this, we follow Einstein's known directive: *a model should be as simple as possible, but no simpler than that*. Namely, we seek the most concise presentation that still has sufficient detail to predict the relevant phenomena.

It is, of course, straightforward to adapt our analysis to any choice of fitness distributions, as the present Gaussian form is not at all an essential or intrinsic component of our modeling framework. Still, aiming to predict and observe the mutation-pandemic phase diagrams, we find that this simplification is quite effective, helping us streamline the analysis.

We now discuss this choice, and its more realistic alternatives on location. We also return to this point in the **Discussion Section**, where, as the Referee suggests, we explain the limitations and potential generalizations of our model. We emphasize again, that our paper proposes a modeling *framework*, not a specific system. This framework, as we now discuss in our revised submission, is quite flexible, and can accept different types of evolutionary parameters, including non-Gaussian fitness landscapes. This flexibility was clearly demonstrated in our analysis of the recurring infections, where we analyzed a fundamentally different landscape, in which fitness follows a sigmoidal function (ϕ_μ).

10. Comment

Minor:

Figure 2 and text on page 5/6: maybe rearrange the panels or the text such that the order of figure panels that you discuss is 2a/b, then c/d, then e/f, instead of a/b, e/f and c/d as it is currently the case.

Response

Done.

We wish to thank the Referee for pushing us to improve our paper, fundamentally in round 1, and with sharp attention to the remaining details in round 2. Together, both rounds led us to a much-improved presentation – first at the macro-level, getting the story and analysis right (round 1), then at the micro-level, refining all remaining details that were still lacking (round 2).

REVIEWER COMMENTS

Reviewer #1 (Remarks to the Author):

The paper has been corrected and improved according the suggestions of the referee and now it could be published.

However, in my opinion, the impact of the paper as already explained does not meet the high standard criteria of Nature Communication and it should be published in a different journal.

Reviewer #3 (Remarks to the Author):

I have read the revised manuscript and all reviewer comments and responses. I can also relate to reviewer #1's criticism, which I think is a result of the authors trying the sell the manuscript in the broadest possible context, which is not necessary and advantageous in my view.

I agree with reviewer #1 that:

"If mutations are too slow, the pathogen prevalence decays prior to the appearance of a critical mutation."

is trivial. But I do not think that the sentence above is the main result of the paper.

Suggestion:

It would help to slightly rewrite the abstract and possibly change the title as well.

In my view, the innovative aspect of the work is the coupling of intra-host and intra-host evolutionary dynamics. I.e. the selection forces that shape intra-host evolution do not necessarily lead to more transmissible strains (both are independent). All subsequent results stem from this basic modelling assumption.

Personally, I find this modelling assumption appealing, as it may apply to many (but not ALL) pathogens. However, this modelling assumption is rarely considered in related work. Hence, the scope/modelling assumptions should probably be made clearer, as it would help the reader to place and interpret the results more accurately, overall strengthening the paper. It would also point more clearly at the utility,

as well as the limitation of this study: I.e. When the intra- and inter-host selection forces are independent; the results hold; if they are coupled, clearly there would be different outcomes (e.g. no volatile zone).

Suggestion title: "Epidemic spreading under mutually independent intra- and inter-host pathogen evolution"

In the abstract emphasize that: "...for many pathogens, different selection forces shape intra- and inter-individual evolution. We show that in the extreme case when these two forces are mutually independent,"

Minor:

Comment 5:

If you compare eq. (10) with eq. (15) there is clearly inconsistent use of notation (the 'tilde' sign). Mathematically, what you mean by

"Therefore σ_c in (15) is only predicted upto a multiplicative constant of order unity, as signified by the \sim sign, i.e. σ_c scales as, not equals to the expression on the r.h.s.."

could very simply be written as \propto in LaTeX, or as $O(\dots)$. For the sake of clarity, please do not use the same math symbol with two very different meanings in mind. It is just unnecessarily confusing.

Reviewer #3

1. Comment

I have read the revised manuscript and all reviewer comments and responses. I can also relate to reviewer #1's criticism, which I think is a result of the authors trying to sell the manuscript in the broadest possible context, which is not necessary and advantageous in my view.

I agree with reviewer #1 that: "If mutations are too slow, the pathogen prevalence decays prior to the appearance of a critical mutation" is trivial. But I do not think that the sentence above is the main result of the paper.

Suggestion: It would help to slightly rewrite the abstract and possibly change the title as well. In my view, the innovative aspect of the work is the coupling of intra-host and inter-host evolutionary dynamics. I.e. the selection forces that shape intra-host evolution do not necessarily lead to more transmissible strains (both are independent). All subsequent results stem from this basic modelling assumption. Personally, I find this modelling assumption appealing, as it may apply to many (but not ALL) pathogens. However, this modelling assumption is rarely considered in related work. Hence, the scope/modelling assumptions should probably be made clearer, as it would help the reader to place and interpret the results more accurately, overall strengthening the paper. It would also point more clearly at the utility, as well as the limitation of this study: I.e. When the intra- and inter-host selection forces are independent; the results hold; if they are coupled, clearly there would be different outcomes (e.g. no volatile zone).

Suggestion title: "Epidemic spreading under mutually independent intra- and inter-host pathogen evolution"

In the abstract emphasize that: "...for many pathogens, different selection forces shape intra- and inter-individual evolution. We show that in the extreme case when these two forces are mutually independent, ..."

Response

Thank you for continuing to offer helpful suggestions on our paper. It has now been through quite a journey and we believe, that with every step, we are further elucidating it and refining its presentation. We agree with the Referee's summary of our main novelty, and with their response to Referee 1's critique. We also agree that it is better to make this context clear already at the title/abstract of the paper. Hence, we have now revised both precisely as the comment suggests.

2. Comment

Minor:

Comment 5: If you compare eq. (10) with eq. (15) there is clearly inconsistent use of notation (the 'tilde' sign). Mathematically, what you mean by "Therefore σc in (15) is only predicted

upto a multiplicative constant of order unity, as signified by the \sim sign, i.e. σc scales as, not equals to the expression on the r.h.s.." could very simply be written as \propto in LaTeX, or as $O(\dots)$ ". For the sake of clarity, please do not use the same math symbol with two very different meanings in mind. It is just unnecessarily confusing.

Response

Corrected. In both Eq. (15), mentioned in the comment, and (16), which was not mentioned but had similar notation. We have also edited the derivation in the Supplementary Information accordingly.